# CLASS-IMBALANCED GRAPH LEARNING WITHOUT CLASS REBALANCING

## ABSTRACT

Class imbalance is prevalent in real-world node classification tasks and poses great challenges for graph machine-learning models. Most existing studies are rooted in a *class-rebalancing* (CR) perspective and aim to address class imbalance with class-wise reweighting or resampling. In this work, we approach the root cause of class-imbalance bias from an orthogonal topological paradigm. Specifically, we theoretically reveal and empirically observe two fundamental phenomena in the underlying graph topology that can greatly exacerbate the predictive bias stemming from class imbalance. In light of these findings, we devise a lightweight topological augmentation framework called ToBE to mitigate the class-imbalance bias *without class rebalancing*. Being orthogonal to CR, the proposed ToBE is a model-agnostic and efficient solution that can be seamlessly combined with and further boost existing CR techniques. Systematic experiments on real-world imbalanced graph learning tasks show that ToBE can deliver up to 46.27% performance gain and up to 72.74% bias reduction over existing techniques. Code is available at https://anonymous.4open.science/r/ToBE/.

## 1 INTRODUCTION

Node classification stands as one of the most fundamental tasks in graph machine learning, holding significant relevance in various real-world applications (Akoglu et al., 2015; Tang & Liu, 2010). Graph Neural Networks (GNNs) have demonstrated great success in tackling related tasks due to their robust representation learning capabilities (Song et al., 2022b). However, real-world graphs are often inherently class-imbalanced, i.e., the sizes of unique classes vary significantly, and a few majority classes have overwhelming numbers in the training set. In *Class-Imbalanced Graph Learning* (CIGL), GNNs are prone to suffer from severe performance degradation on minority class nodes (Shi et al., 2020; Zhao et al., 2021b; Park et al., 2022). This results in a pronounced predictive bias characterized by a large performance disparity between the majority and minority classes.

Traditional imbalance-handling techniques rely on *class rebalancing (CR)* such as class reweighting and resampling (Chawla et al., 2002; Cui et al., 2019), which works well for non-graph data. Recent studies propose more graph-specific CR strategies tailored for CIGL, e.g., neighborhood-aware reweighting (Li et al., 2022; Huang et al., 2022) and oversampling (Zhao et al., 2021b; Park et al., 2022). Nonetheless, these works are restricted to the class-rebalancing paradigm. Parallel to class imbalance, another emerging line of research studies topology imbalance, i.e., "the asymmetric topological properties of the labeled nodes" (Chen et al., 2021). Inspired by this, we theoretically identify a fundamental cause of class-imbalance bias in terms of topological differences between minority and majority classes (Theorems 1 & 2). Our findings reveal an unexplored avenue in CIGL: mitigating class-imbalance bias *without traditional class rebalancing* but instead, with purely topological manipulation. Following this novel perspective, we devise a lightweight practical solution for CIGL that can be seamlessly combined with and further boost existing CR techniques.

Specifically, we theoretically show and empirically observe two fundamental topological phenomena that impose great challenges in CIGL: **(i)** *ambivalent message-passing* (AMP), i.e., high ratio of non-self-class neighbors in the node receptive field, and **(ii)** *distant message-passing* (DMP), i.e., poor connectivity with self-class labeled nodes. Intuitively, AMP leads to a higher influx of noisy information from other classes, and DMP leads to poor reception of effective supervision signals from the same class. Both result in lower signal-to-noise ratios and thus induce higher classification errors. Our theoretical finding further reveals that the minority class is inherently more susceptible

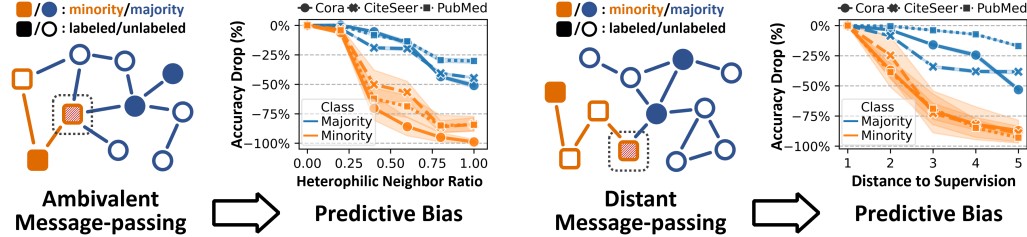

(a) **L**: concept of AMP. **R**: relative performance loss with respect to the non-self-class neighbor ratio.

(b) **L**: concept of DMP. **R**: relative performance loss w.r.t. distance to the nearest same-class labeled node.

Figure 1: Concepts of *ambivalent message-passing* (AMP) and *distant message-passing* (DMP) and their impact in real-world imbalanced node classification tasks (Park et al., 2022). Both factors lead to a substantial increase in prediction errors, and further, a larger performance disparity/bias (i.e., the gap between the blue and orange curves) between the majority and minority classes.

to both AMP and DMP, which leads to a more pronounced predictive bias. Such bias induced by the graph topology escalates as the level of class imbalance increases. Fig. 1 visually illustrates the concepts of AMP and DMP, highlighting their distinct impacts on the predictive performance of majority and minority classes.

These findings raise the question: how to tame the errors and biases induced by graph topology in CIGL? To answer this, we devise TOBE (Topological Balanced augmEntation), a model-agnostic and efficient technique to handle the topological challenges in class-imbalanced node classification. Guided by the theoretical and empirical findings, TOBE dynamically locates and rectifies nodes critically influenced by AMP and DMP during learning, thereby effectively reducing the errors and biases in CIGL. Being orthogonal to class rebalancing, our solution is able to work hand-in-hand with existing techniques based on reweighting (Japkowicz & Stephen, 2002; Chen et al., 2021) and resampling (Chawla et al., 2002; Zhao et al., 2021b; Park et al., 2022) and further boost their performance. Systematic experiments on real-world CIGL tasks show that TOBE delivers significant performance boost (up to 46.27%) and bias reduction (up to 72.74%) over various CIGL baselines with diverse GNN architectures.

Our contributions: **(i) Novel Perspective.** We demonstrate the feasibility of taming class-imbalance bias *without* class rebalancing, which provides a new avenue that is orthogonal to the predominant class-rebalancing practice in CIGL. **(ii) Theoretical Insights.** We theoretically show the differences in graph topology between minority and majority classes and investigate the effect of these differences in shaping predictive bias, shedding light on future CIGL research. **(iii) Practical Solution.** We devise a lightweight topological augmentation technique TOBE for class-imbalanced node classification. As a *model-agnostic*, *effective*, and *efficient* solution, it can be seamlessly combined with and further boost existing class-rebalancing techniques. **(iv) Empirical Study.** We empirically validate our theoretical findings and methodological designs. Systematic experiments and analysis across a diverse range of real-world tasks and GNN architectures show that TOBE consistently demonstrates superior performance in both *promoting classification* and *mitigating predictive bias*.

## 2 CLASS IMBALANCE AND LOCAL TOPOLOGY

In this section, we delve into the impact of graph topology on the predictive bias in class-imbalanced node classification. We theoretically unveil that compared to the majority class, the minority class is inherently more susceptible to both Ambivalent Message Passing (AMP) and Distant Message Passing (DMP). This significantly worsens minority-class performance and leads to a more pronounced predictive bias stemming from class imbalance. After that, we present an empirical analysis to validate our theoretical findings, and to provide insights on how to mitigate the bias induced by AMP and DMP in practice. Detailed proofs can be found in Appendix A.

**Theoretical analysis on local topology.** Consider a graph $\mathcal{G} : (\mathcal{V}, \mathcal{E})$ generated from a stochastic block model (Holland et al., 1983) SBM$(n, p, q)$, where $n$ is the total number of nodes, $p$ and $q$ are the intra- and inter-class node connection probability. To facilitate analysis, we call node $u$ *homo-connected* to node $v$ if there is a path $[u, v_1, ..., v_k, v]$ where $v_1, ..., v_k, v$ are of the same class, and let

$\mathcal{H}(u, k)$ denote the set of $k$-hop homo-connected neighbors of $u$. For binary node classification, we denote the number of nodes of class $i$ as $n_i$ ($n_1 + n_2 = n$); without loss of generality, let class 1/2 be the minority/majority class (thus $n_1 \ll n_2$). We denote class $i$'s node set as $\mathcal{V}_i$ and labeled node set as $\mathcal{V}_i^{\text{L}}$ ($\subset \mathcal{V}_i$). For asymptotic analysis, we adopt conventional assumptions: $n_1 \cdot p = \beta + \mathcal{O}\left(\frac{1}{n}\right)$ (i.e., $\beta$ is the average intra-class node degree of class 1); $p/q = \mathcal{O}(1)$ (Decelle et al., 2011).

We now give formal definitions of AMP and DMP. For a node $u$ from class $i$, we define its **(i)** *k-hop AMP coefficient* $\alpha^k(u) \in [0, \infty)$ as the ratio of the expected number of non-self-class nodes to self-class nodes in its $k$-hop neighborhood $\mathcal{H}(u, k)$, i.e., $\alpha^k(u) := \frac{|\{v | v \notin \mathcal{V}_i, v \in \mathcal{H}(u,k)\}|}{|\{v | v \in \mathcal{V}_i, v \in \mathcal{H}(u,k)\}|}$; **(ii)** *k-hop DMP coefficient* $\delta^k(u) \in \{0, 1\}$ as the indicator of whether all labeled nodes in its $k$-hop neighborhood are NON-self-class, i.e., $\delta^k(u) := \mathbb{1}(L_i^k(u) = 0, \Sigma_j L_j^k(u) > 0)$, where $L_j^k(u) = |\{v | v \in \mathcal{V}_j^{\text{L}}, v \in \mathcal{H}(u, k)\}|$. For an intuitive example, the target node (marked by the dashed box) in Fig. 1(a) has $\alpha^1(u) = 3/1, \delta^1(u) = 0$ and node in Fig. 1(b) has $\alpha^1(u) = 1/1, \delta^1(u) = 1$. Further, to characterize the level of AMP/DMP for different *class*, for class $i$ we define $\alpha_i^k := \frac{\mathbb{E}_{u \in \mathcal{V}_i}[|\{v | v \notin \mathcal{V}_i, v \in \mathcal{H}(u,k)\}|]}{\mathbb{E}_{u \in \mathcal{V}_i}[|\{v | v \in \mathcal{V}_i, v \in \mathcal{H}(u,k)\}|]}$ and $\delta_i^k := \mathbb{P}(\delta^k(u) = 1)$, where $u$ is a node of class $i$. Intuitively, a higher $\alpha_i$ or $\delta_i$ indicates that class $i$ is more susceptible to AMP or DMP. Building on these metrics, we analyze the disparities in $\alpha$ and $\delta$ between minority and majority classes, thereby providing insights into how the *underlying graph topology induces additional class-imbalance bias*.

We provide a $k$-hop analysis here. To facilitate drawing conclusions, we define imbalance ratio $\rho := n_2/n_1$. The larger the $\rho$ is, the more imbalanced the dataset is. Then for AMP, we have the following Theorem 1.

**Theorem 1** (AMP-sourced bias). *For a large $n$, the ratio of AMP coefficients $\alpha$ for the minority class to the majority class grows polynomially with the imbalance ratio $\rho$ and exponentially with $k$:*

$$\frac{\alpha_1^k}{\alpha_2^k} = \left(\rho \cdot \frac{\sum_{t=1}^{k}(\rho\beta)^{t-1}}{\sum_{t=1}^{k}\beta^{t-1}}\right)^2 + \mathcal{O}\left(\frac{1}{n}\right). \tag{1}$$

Theorem 1 shows that the same-class neighbor proportion of minority-class nodes is significantly smaller than that of majority-class nodes, i.e., the minority class is more susceptible to AMP. As the imbalance ratio $\rho$ increases, this issue becomes even more pronounced and introduces a higher bias into the learning process. Moving on to DMP, we have the following Theorem 2.

**Theorem 2** (DMP-sourced bias). *Let $r_i^{\text{L}} := \frac{|\mathcal{V}_i^{\text{L}}|}{|\mathcal{V}_i|}$ denote the label rate of class $i$. For a large $n$, the ratio of DMP coefficients $\delta$ of the minority class over the majority class grows exponentially with $\rho$:*

$$\frac{\delta_1^k}{\delta_2^k} \approx \frac{1 - r_1^{\text{L}}}{1 - r_2^{\text{L}}} e^{(\rho-1)\beta} + \mathcal{O}\left(\frac{1}{n}\right). \tag{2}$$

Similarly, the result shows that the minority class exhibits a significantly higher susceptibility to DMP than the majority class. Theorem 2 also has several interesting implications: **(i)** *The imbalance ratio greatly affects the bias induced by DMP*, as $\delta_1^k/\delta_2^k$ grows exponentially with $\rho$. **(ii)** *Labeling more minority-class nodes can mitigate, but hardly solve the problem.* Enlarging the minority-class label rate $r_1^{\text{L}}$ can linearly shrink $\delta_1^k/\delta_2^k$, but it can hardly eliminate the bias induced by DMP (i.e., to have $\delta_1^k \leq \delta_2^k$) as $e^{(\rho-1)\beta}$ is usually very large in practice. Take the *Cora* dataset (Sen et al., 2008) as an example (let class 1/class 2 denote the smallest/largest class): eliminating the DMP bias requires the minority-class label rate $r_1^{\text{L}} \geq 1 - \frac{1 - r_2^{\text{L}}}{e^{(\rho-1)\beta}} > 1 - 5.05 \times 10^{-8}$, which is practically infeasible.

**A closer look at AMP & DMP in practice.** Our theoretical findings show that both AMP and DMP affect the minority and majority classes differently, and the difference is primarily determined by the imbalance ratio $\rho$. However, directly manipulating $\rho$ is tricky in practice as it requires sampling new nodes and edges from an unknown underlying graph generation model, or at least, simulating the process by oversampling. To verify the theoretical results, and to provide more insights on how to mitigate the bias brought about by AMP and DMP in practice, we conduct a fine-grained empirical analysis on a real-world task[1]. Results are detailed in Fig. 2.

Starting from Fig. 2(a), we can first observe that the minority class 1 has a larger proportion of nodes with high $\alpha$ or $\delta$ than the majority class 2. This naturally leads to a significantly higher average $\alpha$

---

[1]Results obtained by training a GCN on the *PubMed* dataset (minority class 0 vs. the rest, $\rho = 3.81$).

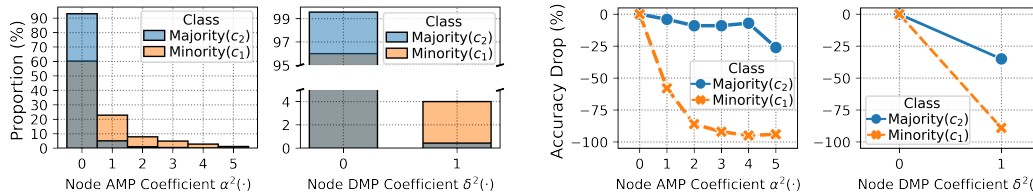

(a) Distribution of node AMP/DMP coefficients.

(b) Impact of AMP/DMP on predictive performance.

Figure 2: Node-level distribution of AMP and DMP coefficients and their impact on learning.

and $\delta$ for class 1 than for class 2 (specifically, $\alpha_1/\alpha_2 = 1.357/0.179$, $\delta_1/\delta_2 = 0.040/0.004$), which is consistent with our theoretical findings. Further, Fig. 2(b) shows that both AMP and DMP significantly reduce the prediction accuracy, especially for the minority class. This can be explained from a graph signal denoising perspective (Nt & Maehara, 2019): AMP introduces additional noise from dissimilar nodes, and DMP leads to less efficient label propagation/denoising, thus their impact is particularly significant on minority classes that are more susceptible to noise (Johnson & Khoshgoftaar, 2019) due to poor representation in the feature space. Taken together, we note an intriguing fact that the impact of AMP/DMP is concentrated on a small fraction of "critical" minority nodes with large $\alpha$ or $\delta$ (e.g., the $\alpha \geq 1$ / $\delta = 1$ part in Fig. 2(a)). In other words, one can surrogate the tricky manipulation of $\rho$ and directly mitigate the impact of AMP/DMP by *locating and rectifying a small number of critical nodes*, and this exactly motivates our subsequent studies.

## 3  HANDLING CLASS IMBALANCE VIA TOPOLOGICAL AUGMENTATION

Armed with the findings from Section 2, we now discuss how to devise a practical strategy to mitigate the error and bias induced by graph topology in CIGL. Earlier analyses have shown that this can be achieved by identifying and rectifying the critical nodes that are highly influenced by AMP/DMP. This naturally poses two challenging questions: **(i)** How can critical nodes be located as the direct calculation of $\alpha/\delta$ using ground-truth labels is not possible? **(ii)** Subsequently, how can critical nodes be rectified and minimize the negative impact caused by AMP and DMP?

In answering the above questions, we devise a *lightweight* framework TOBE (Topological Balanced augmEntation) for handling the topology-sourced errors and biases in CIGL. Specifically, for locating the misclassified nodes, ToBA leverages model-based prediction uncertainty to assess the risk of potential misclassification caused by AMP/DMP for each node (§ 3.1). Then to rectify a misclassified node, we estimate a posterior likelihood of each node being in each class (§ 3.2) and dynamically augment the misclassified node's topological context based on our risk scores and posterior likelihoods (§ 3.3) thereby mitigating the impact of AMP and DMP. An overview of the proposed TOBE framework is shown in Fig. 3.

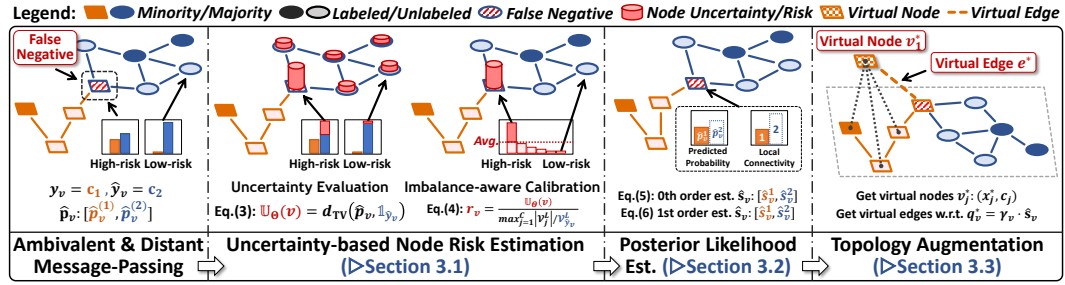

Figure 3: The proposed TOBE framework, best viewed in color.

### 3.1  NODE MISCLASSIFICATION RISK ESTIMATION

We now elaborate on the technical details of TOBE. As discussed earlier, our first step is to locate the critical nodes that are highly influenced by AMP/DMP. Given the unavailability of ground-truth labels, direct computation of AMP/DMP coefficient is infeasible in practice. Fortunately, recent studies have shown that conflicting or lack of information from the neighborhood can disturb GNN learning and the associated graph-denoising process for affected nodes (Nt & Maehara, 2019; Wu et al., 2019; Ma et al., 2021). This further yields high vacuity or dissonance uncertainty (Stadler

et al., 2021; Zhao et al., 2020) in the prediction. This motivates us to exploit the *model prediction uncertainty* to estimate nodes' risk of being misclassified due to AMP/DMP.

**Uncertainty quantification.** While there exist many techniques for uncertainty quantification (e.g., Bayesian-based (Zhang et al., 2019; Hasanzadeh et al., 2020), Jackknife sampling (Kang et al., 2022a)), they often either have to modify the model architecture, and/or impose additional computational overhead. In this study, we aim to streamline the design of ToBE for optimal efficiency and adaptability. To this end, we employ an efficient and highly effective approach to uncertainty quantification. Formally, let $C$ be the number of classes. For a node $v$, consider model $F(\cdot; \Theta)$'s predicted probability vector $\hat{\boldsymbol{p}}_v = F(\boldsymbol{A}, \boldsymbol{X}; \Theta)_v$, i.e., $\hat{p}_v^{(j)} = \mathbb{P}(y_v = j | \boldsymbol{A}, \boldsymbol{X}, \Theta)$. Let $\hat{y}_v$ be the predicted label. We measure the uncertainty score $\mathbb{U}_\Theta(v)$ by the total variation (TV) distance:

$$\mathbb{U}_\Theta(v) := d_{\mathrm{TV}}(\hat{\boldsymbol{p}}_v, \mathbb{1}_{\hat{y}_v}) = \tfrac{1}{2} \sum_{j=1}^C |\hat{\boldsymbol{p}}_v^{(j)} - \mathbb{1}_{\hat{y}_v}^{(j)}| \in [0, 1]. \tag{3}$$

Intuitively, a node has higher uncertainty if the model is less confident about its current prediction. We remark that this metric can be naturally replaced by other uncertainty measures (e.g., information entropy or more complex ones) with additional computation cost, yet the impact on performance is marginal. Please refer to the ablation study provided in Appendix D.1 for more details.

**Imbalance-calibrated misclassification risk.** Due to the lack of training instances, minority classes generally exhibit higher uncertainty. Therefore, using $\mathbb{U}_\Theta(\cdot)$ directly as the risk score would treat most minority-class nodes as high-risk, which is contrary to our intention of rectifying the false negatives (i.e., minority nodes wrongly predicted as majority-class) that cause bias in CIGL. To cope with this, we propose *imbalance-aware calibration* for risk scores. For each class $i$, let $\hat{\mathcal{V}}_i := \{u \in \mathcal{V} | \hat{y}_u = i\}$ and $\hat{\mathcal{V}}_i^{\mathrm{L}} := \{u \in \mathcal{V}^{\mathrm{L}} | y_u = i\}$. For node $v$ with predicted label $\hat{y}_v$, we define its risk $r_v$ as:

$$r_v := \frac{\mathbb{U}_\Theta(v)}{\max_{j=1}^C |\mathcal{V}_j^{\mathrm{L}}| / |\mathcal{V}_{\hat{y}_v}^{\mathrm{L}}|} \in [0, 1]. \tag{4}$$

Intuitively speaking, Eq. (4) calibrates $v$'s prediction uncertainty by a *label imbalance score* $\max_{j=1}^C |\mathcal{V}_j^{\mathrm{L}}| / |\mathcal{V}_{\hat{y}_v}^{\mathrm{L}}|$. Minority classes with smaller labeled sets $\mathcal{V}_i^{\mathrm{L}}$ will be discounted more.

**Empirical validation.** We validate the effectiveness of the proposed node risk assessment method, as shown in Fig. 4. The results indicate that our approach can accurately estimate node misclassification risk across various real-world CIGL tasks while enjoying computational efficiency.

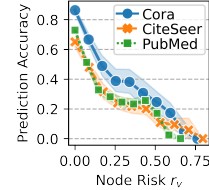

Figure 4: The negative correlation between the estimated node risk (x-axis) and the prediction accuracy (y-axis). We apply 10 sliding windows to compute the mean and deviation of the accuracy.

## 3.2 Posterior Likelihood Estimation

With the estimated risk scores of being affected by AMP/DMP, we move to the next question: how to rectify high-risk nodes with topological augmentation? As high-risk nodes are prone to misclassification, their true labels are more likely to be among the non-predicted classes $j \neq \hat{y}_v$. This motivates us to investigate schemes to harness information from these non-predicted classes. Since uniformly drawing from all classes probably introduces noise to learning, we propose to estimate the *posterior likelihood* $\hat{s}_v^{(j)}$ that a high-risk node $v$ belongs to each class $j$ after observing the current predictions. To estimate $\hat{s}_v^{(j)}$, we introduce a *zeroth-order* scheme and a *first-order* scheme with $\mathcal{O}(|\mathcal{V}|C)$ and $\mathcal{O}(|\mathcal{E}|C)$ time complexity, respectively. Please refer to §4 for a practical complexity analysis. We do not employ higher-order schemes due to the $\mathcal{O}(|\mathcal{V}|^{k-1}|\mathcal{E}|C)$ time complexity of the $k$th-order scheme.

**Zeroth-order estimation.** A natural approach is to utilize the predicted probabilities $\hat{p}_v^{(j)}$. As we have shown, the predicted label $\hat{y}_v$ of a high-risk node $v$ is very likely to be wrong. Thus, we define the posterior likelihood $\hat{s}_v^{(j)}$ as the conditional probability given that the class is not $\hat{y}_v$, i.e.,

$$\hat{s}_v^{(j)} := \mathbb{P}_{y \sim \hat{\boldsymbol{p}}_v}[y = j | y \neq \hat{y}_v] = \begin{cases} \hat{p}_v^{(j)} / (1 - \hat{p}_v^{(\hat{y}_v)}), & \text{if } j \neq \hat{y}_v, \\ 0, & \text{if } j = \hat{y}_v. \end{cases} \tag{5}$$

Intuitively, the posterior likelihoods $\hat{s}_v^{(j)}$ are consistent with the predicted probabilities $\hat{p}_v^{(j)}$ except for the wrongly predicted label $j = \hat{y}_v$. This can be computed efficiently on GPU in matrix form.

**First-order estimation via random walk.** We further explore the *local topology* for $\hat{s}_v^{(j)}$ estimation. Since neighboring nodes on a homophily graph tend to share labels, we consider a 1-step random walk starting from node $v$. Let $\mathcal{N}(v)$ be the neighboring node set of $v$, and let $v' \sim \mathcal{N}(v)$ denote the ending node of the random walk. We define $\hat{s}_v^{(j)}$ as the conditional probability that $v'$ is predicted as class $j$ given that $v'$ is not predicted as class $\hat{y}_v$, i.e.,

$$\hat{s}_v^{(j)} := \mathbb{P}_{v' \sim \mathcal{N}(v)}[\hat{y}_{v'} = j | \hat{y}_{v'} \neq \hat{y}_v] = \begin{cases} \frac{|\{v' \in \mathcal{N}(v) | \hat{y}_{v'} = j\}|}{|\mathcal{N}(v)| - |\{v' \in \mathcal{N}(v) | \hat{y}_{v'} = \hat{y}_v\}|}, & \text{if } j \neq \hat{y}_v, \\ 0, & \text{if } j = \hat{y}_v. \end{cases} \quad (6)$$

Intuitively, $\hat{s}_v^{(j)}$ is proportional to the label frequency among adjacent nodes. Different from the zeroth-order scheme, this scheme relies on both node-level predictions and local connectivity patterns. The computation can be done via sparse matrix operation with $\mathcal{O}(|\mathcal{E}|C)$ time complexity. As a remark, although this scheme can extend to $k$-step random walks, we do not employ them due to the $\mathcal{O}(|\mathcal{V}|^{k-1}|\mathcal{E}|C)$ complexity of exact computation and the high variance of stochastic computation.

**Empirical validation.** Figure 5 compares the two schemes in practice. Results show that all high-risk ($r_v > 0$) minority nodes are misclassified, and both schemes can effectively find alternatives with significantly higher chances to be the ground truth class for high-risk nodes[2].

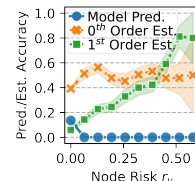

Figure 5: The minority-class accuracy of model prediction $\hat{y}_v = F(\boldsymbol{A}, \boldsymbol{X}; \Theta)$, and max-likelihood-based candidate selection $\hat{y}_v^s = \mathrm{argmax}(\hat{\boldsymbol{s}}_v)$, on PubMed dataset.

### 3.3 VIRTUAL TOPOLOGY AUGMENTATION

Finally, we discuss how to mitigate AMP and DMP via topology augmentation using our node risk scores and the posterior likelihoods. The general idea is to augment the local topology of high-risk nodes so as to integrate information from nodes that share similar patterns (mitigate AMP), even if they are not closely adjacent to each other in the graph topology (mitigate DMP), thus achieving less-biased CIGL. A straightforward way is to connect high-risk nodes to nodes from high-likelihood classes in the original graph. However, this can be problematic in practice as a massive number of possible edges could be generated, greatly disturbing the original topology structure.

To achieve efficient augmentation without disrupting the graph topology, we create virtual nodes (one per class) as "shortcuts" connecting to high-risk nodes according to posterior likelihoods. These shortcuts aggregate and pass class information to high-risk nodes from nodes that exhibit similar patterns (even if they are distant in the original graph), thus mitigating both AMP and DMP.

Formally, for each class $j$, we build a virtual node $v_j^*$ with feature $\boldsymbol{x}_{v_j^*} := \sum_{v \in \hat{\mathcal{V}}_j} \boldsymbol{x}_v / |\hat{\mathcal{V}}_j|$ and label $y_{v_j^*} := j$, and compute the average risk $\bar{r}_j := \sum_{v \in \hat{\mathcal{V}}_j} r_v / |\hat{\mathcal{V}}_j|$. Then for each node $v$, we connect a virtual edge between $v$ and virtual node $v_j^*$ with probability proportional to the posterior likelihood $\hat{s}_v^{(j)}$. However, if the connection probability is exactly $\hat{s}_v^{(j)}$, there will be many unnecessary virtual edges for low-risk nodes. Hence, we introduce a discount factor $\gamma_v$ based on risk scores and connect the virtual edge with probability $q_v^{(j)} := \gamma_v \hat{s}_v^{(j)}$. To design the optimal $\gamma_v$, we propose to solve the following constrained quadratic program:

$$\min_{\boldsymbol{\gamma} \geq \boldsymbol{0}} \left( - \sum_{v \in \mathcal{V}} (r_v - \bar{r}_{\hat{y}_v}) \gamma_v + \frac{1}{2} \|\boldsymbol{\gamma}\|_2^2 \right), \quad (7)$$

where the first term encourages virtual edges for high-risk nodes, and the second term is to minimize the number of virtual edges. The closed-form solution is $\gamma_v = \max(r_v - \bar{r}_{\hat{y}_v}, 0)$ (Antoniadis & Fan, 2001), which avoids virtual edges for low-risk nodes as we desire.

**Final Remarks.** Our algorithm scales linearly with the number of nodes/edges with 0th/1st-order estimation (has $\mathcal{O}(|\mathcal{V}|C)/\mathcal{O}(|\mathcal{E}|C)$ complexity), and can be executed in parallel on matrix form. This makes ToBE highly efficient and allows dynamically graph augmentation in each training step. A detailed complexity analysis is provided in the next section (see Table 3), and we discuss the further speedup in D.2. The procedure of ToBE is summarized in Algorithm 1 in Appendix B.

---

[2]Note that Fig. 5 is just an illustrative example using $\arg \max(\hat{\boldsymbol{s}}_v)$. In practice, we consider the whole $\hat{\boldsymbol{s}}_v$ when sampling virtual edges, as described in Section 3.3.

Table 1: Performance of ToBE in real-world IGL tasks. We report metrics for both classification performance (Balanced Accuracy, Macro-F1) and bias (PerfStd), as well as the absolute gain after applying ToBE. Due to space limitation, we omit the error bar and only report the results of the best IGL technique for most settings, full results can be found in Appendix E. For each baseline's three settings, i.e., Base, + ToBE$_0$, + ToBE$_1$, we mark the best/second-best results in **bold**/underlined.

| Dataset (IR=10) | | Cora | | | CiteSeer | | | PubMed | | |
|---|---|---|---|---|---|---|---|---|---|---|
| **Metric: BAcc.↑** | | Base | + ToBE$_0$ | + ToBE$_1$ | Base | + ToBE$_0$ | + ToBE$_1$ | Base | + ToBE$_0$ | + ToBE$_1$ |
| GCN | Vanilla | 61.56 | 65.54$_{+3.98}$ | **69.80**$_{+8.24}$ | 37.62 | 52.65$_{+15.03}$ | **55.37**$_{+17.75}$ | 64.23 | **68.62**$_{+4.39}$ | 67.57$_{+3.34}$ |
| | Reweight | 67.65 | 70.97$_{+3.32}$ | **72.14**$_{+4.49}$ | 42.49 | 57.91$_{+15.41}$ | **58.36**$_{+15.87}$ | 71.20 | **74.19**$_{+2.98}$ | 73.37$_{+2.16}$ |
| | ReNode | 66.60 | 71.37$_{+4.77}$ | **71.84**$_{+5.25}$ | 42.57 | 57.47$_{+14.90}$ | **59.28**$_{+16.70}$ | 71.52 | **73.20**$_{+1.68}$ | 72.53$_{+1.02}$ |
| | Resample | 59.48 | 72.51$_{+13.03}$ | **74.24**$_{+14.76}$ | 39.15 | 57.90$_{+18.75}$ | **58.78**$_{+19.63}$ | 64.97 | 72.53$_{+7.56}$ | **72.87**$_{+7.90}$ |
| | SMOTE | 58.27 | 72.16$_{+13.89}$ | **73.89**$_{+15.62}$ | 39.27 | 60.06$_{+20.79}$ | **61.97**$_{+22.71}$ | 64.41 | **73.17**$_{+8.77}$ | 73.13$_{+8.73}$ |
| | GSMOTE | 67.99 | 68.52$_{+0.53}$ | **71.55**$_{+3.56}$ | 45.05 | **57.68**$_{+12.63}$ | 57.65$_{+12.60}$ | 73.99 | 73.09$_{-0.90}$ | **76.57**$_{+2.58}$ |
| | GENS | 70.12 | 72.22$_{+2.10}$ | **72.58**$_{+2.46}$ | 56.01 | 60.60$_{+4.59}$ | **62.67**$_{+6.65}$ | 73.66 | 76.11$_{+2.45}$ | **76.91**$_{+3.25}$ |
| | Best | 70.12 | 72.51$_{+2.39}$ | **74.24**$_{+4.11}$ | 56.01 | 60.60$_{+4.59}$ | **62.67**$_{+6.65}$ | 73.99 | 76.11$_{+2.12}$ | **76.91**$_{+2.92}$ |
| GAT | Best | 69.76 | 72.14$_{+2.38}$ | **73.29**$_{+3.53}$ | 51.50 | 60.95$_{+9.44}$ | **63.49**$_{+11.99}$ | 73.13 | 75.55$_{+2.42}$ | **75.65**$_{+2.52}$ |
| SAGE | Best | 68.84 | 71.31$_{+2.47}$ | **73.02**$_{+4.18}$ | 52.57 | 64.36$_{+11.78}$ | **66.35**$_{+13.77}$ | 71.55 | 75.89$_{+4.34}$ | **77.38**$_{+5.83}$ |
| APPNP | Best | 73.74 | **75.02**$_{+1.28}$ | 73.78$_{+0.05}$ | 50.88 | **66.62**$_{+15.74}$ | 65.57$_{+14.69}$ | 72.76 | 73.37$_{+0.62}$ | **74.90**$_{+2.14}$ |
| GPRGNN | Best | 73.38 | 74.01$_{+0.63}$ | **74.89**$_{+1.51}$ | 54.66 | **64.16**$_{+9.51}$ | 63.89$_{+9.23}$ | 73.56 | 75.69$_{+2.13}$ | **77.49**$_{+3.93}$ |
| **Metric: Macro-F1↑** | | Base | + ToBE$_0$ | + ToBE$_1$ | Base | + ToBE$_0$ | + ToBE$_1$ | Base | + ToBE$_0$ | + ToBE$_1$ |
| GCN | Best | 69.96 | 71.62$_{+1.66}$ | **72.82**$_{+2.86}$ | 54.45 | 59.89$_{+5.43}$ | **62.46**$_{+8.01}$ | 71.28 | 75.77$_{+4.49}$ | **76.86**$_{+5.58}$ |
| GAT | Best | 69.96 | 70.87$_{+0.91}$ | **72.31**$_{+2.35}$ | 48.34 | 60.04$_{+11.70}$ | **62.55**$_{+14.21}$ | 71.78 | **75.13**$_{+3.35}$ | 74.96$_{+3.17}$ |
| SAGE | Best | 68.23 | 70.40$_{+2.17}$ | **71.71**$_{+3.48}$ | 51.05 | 63.87$_{+12.82}$ | **65.91**$_{+14.86}$ | 70.06 | 75.33$_{+5.27}$ | **76.92**$_{+6.86}$ |
| APPNP | Best | 73.67 | **73.67**$_{+0.00}$ | 73.22$_{-0.45}$ | 45.25 | **66.18**$_{+20.93}$ | 65.20$_{+19.95}$ | 70.65 | 72.55$_{+1.91}$ | **74.61**$_{+3.96}$ |
| GPRGNN | Best | 73.08 | 72.89$_{-0.18}$ | **73.54**$_{+0.47}$ | 50.34 | **63.59**$_{+13.25}$ | 63.12$_{+12.78}$ | 71.45 | 75.47$_{+4.03}$ | **77.62**$_{+6.17}$ |
| **Metric: PerfStd↓** | | Base | + ToBE$_0$ | + ToBE$_1$ | Base | + ToBE$_0$ | + ToBE$_1$ | Base | + ToBE$_0$ | + ToBE$_1$ |
| GCN | Best | 20.04 | **14.43**$_{-5.61}$ | 15.25$_{-4.79}$ | 16.95 | **13.82**$_{-3.13}$ | 13.93$_{-3.02}$ | 11.93 | **3.35**$_{-8.58}$ | 5.15$_{-6.78}$ |
| GAT | Best | 20.08 | **15.05**$_{-5.03}$ | 17.32$_{-2.76}$ | 25.21 | **10.68**$_{-14.53}$ | 13.24$_{-11.97}$ | 10.29 | **3.01**$_{-7.28}$ | 4.58$_{-5.71}$ |
| SAGE | Best | 19.81 | **13.32**$_{-6.49}$ | 14.87$_{-4.94}$ | 19.76 | 13.17$_{-6.59}$ | **12.78**$_{-6.98}$ | 11.76 | **3.35**$_{-8.41}$ | 4.09$_{-7.68}$ |
| APPNP | Best | 18.09 | **16.87**$_{-1.22}$ | 18.46$_{+0.37}$ | 25.95 | **14.91**$_{-11.04}$ | 19.19$_{-6.75}$ | 14.49 | 8.04$_{-6.45}$ | **3.95**$_{-10.54}$ |
| GPRGNN | Best | 18.84 | **16.78**$_{-2.06}$ | 17.89$_{-0.95}$ | 24.14 | **19.84**$_{-4.30}$ | 22.83$_{-1.31}$ | 14.40 | 9.75$_{-4.65}$ | **6.61**$_{-7.79}$ |

*ToBE$_0$/ToBE$_1$: ToBE with $0^{\text{th}}$/$1^{\text{st}}$-order posterior likelihood estimation. We report the average score of 5 independent runs to eliminate randomness.

## 4 EXPERIMENTS

We carry out systematic experiments and analysis to validate ToBE in the following aspects: **(i) Effectiveness** in both *promoting imbalanced node classification* and *mitigating the prediction bias* between different classes. **(ii) Versatility** in cooperating with and further boosting various CIGL techniques and GNN backbones. **(iii) Robustness** to extreme class imbalance. **(iv) Efficiency** in real-world applications. We refer the reader to Appendix for reproducibility details (§C), ablation study, limitations and more discussions of ToBE (§D), and full empirical results (§E).

**Experiment Setup.** We validate ToBE on five benchmark datasets for semi-supervised node classification, including the *Cora, CiteSeer, PubMed* from Plantoid graphs (Sen et al., 2008), and larger-scale *CS, Physics* from co-author networks (Shchur et al., 2018) with high-dimensional features. Following the same setting as prior studies (Park et al., 2022; Song et al., 2022a; Zhao et al., 2021b), we select half of the classes as minority. The imbalance ratio $\rho = n_{max}/n_{min} \geq 1$ is the ratio between the size of the largest class to the smallest class, i.e., more imbalance ⇔ higher IR. Detailed data statistics and class distributions can be found in Appendix C.1. We test ToBE with **six** CIGL techniques (Park et al., 2022; Chen et al., 2021; Zhao et al., 2021b; Chawla et al., 2002; Japkowicz & Stephen, 2002) and **five** popular GNN backbones (Chien et al., 2020; Gasteiger et al., 2018; Veličković et al., 2018; Hamilton et al., 2017; Welling & Kipf, 2016) under all possible combinations to fully validate ToBE's effectiveness and versatility in practice. Note that although there are other techniques available for CIGL (Hong et al., 2021; Kang et al., 2019; Shi et al., 2020), previous studies (Park et al., 2022; Song et al., 2022a) have shown they are generally outperformed by the baselines we use. Detailed settings can be found in Appendix C.2. To ensure a comprehensive evaluation, we employ **three** metrics to assess both the classification performance (Balanced Accuracy, Macro-F1) and the model predictive bias (PerfStd, i.e., the standard deviation of accuracy scores across all classes). Lower PerfStd indicates smaller performance gap between all majority and minority classes, and thus smaller predictive bias. See details in Appendix C.3. For clarity, we use ↑/↓ to denote larger/smaller is better for each metric.

**On the effectiveness and versatility of ToBE (Table 1).** We report the main results in Table 1. In **all** 630 setting combinations (3 datasets×5 backbones×7 baselines×2 ToBE variants×3 metrics), ToBE achieves significant and consistent performance improvements over other CIGL tech-

Table 2: Performance of TOBE under varying types and levels of class imbalance. We report the best balanced accuracy score here due to space limitation, full results can be found in Appendix E.

| Dataset | Cora | | CiteSeer | | PubMed | | CS | | Physics | |
|---|---|---|---|---|---|---|---|---|---|---|
| **Step IR** | 10 | 20 | 10 | 20 | 10 | 20 | 10 | 20 | 10 | 20 |
| Base | 61.6 | 52.7 | 37.6 | 34.2 | 64.2 | 60.8 | 75.4 | 65.3 | 80.1 | 67.7 |
| + TOBE | 69.8$_{+8.2}$ | 71.3$_{+18.5}$ | 55.4$_{+17.7}$ | 51.3$_{+17.1}$ | 68.6$_{+4.4}$ | 63.3$_{+2.5}$ | 82.6$_{+7.2}$ | 79.9$_{+14.5}$ | 87.6$_{+7.5}$ | 88.0$_{+20.2}$ |
| BestIGL | 70.1$_{+8.6}$ | 66.5$_{+13.8}$ | 56.0$_{+18.4}$ | 47.2$_{+13.0}$ | 74.0$_{+9.8}$ | 71.1$_{+10.3}$ | 84.1$_{+8.7}$ | 81.3$_{+15.9}$ | 89.4$_{+9.3}$ | 85.7$_{+18.0}$ |
| + TOBE | **74.2**$_{+12.7}$ | **71.6**$_{+18.9}$ | **62.7**$_{+25.0}$ | **62.5**$_{+28.3}$ | **76.9**$_{+12.7}$ | **75.7**$_{+14.9}$ | **86.3**$_{+11.0}$ | **85.6**$_{+20.2}$ | **91.2**$_{+11.1}$ | **90.9**$_{+23.2}$ |
| **Natural IR** | 50 | 100 | 50 | 100 | 50 | 100 | 50 | 100 | 50 | 100 |
| Base | 58.1 | 61.8 | 44.9 | 44.7 | 52.0 | 51.1 | 73.8 | 71.4 | 76.0 | 77.7 |
| + TOBE | 69.1$_{+11.0}$ | 68.3$_{+6.5}$ | 58.4$_{+13.4}$ | 57.4$_{+12.7}$ | 55.6$_{+3.6}$ | 56.5$_{+5.3}$ | 82.1$_{+8.3}$ | 81.9$_{+10.5}$ | 86.9$_{+10.9}$ | 84.1$_{+6.4}$ |
| BestIGL | 71.0$_{+12.9}$ | 73.8$_{+12.0}$ | 56.3$_{+11.4}$ | 56.3$_{+11.6}$ | 72.7$_{+20.7}$ | 72.8$_{+21.7}$ | 81.2$_{+7.4}$ | 81.4$_{+10.0}$ | 85.8$_{+9.8}$ | 87.2$_{+9.5}$ |
| + TOBE | **73.1**$_{+15.0}$ | **76.9**$_{+15.1}$ | **62.1**$_{+17.1}$ | **61.3**$_{+16.6}$ | **75.8**$_{+23.8}$ | **75.9**$_{+24.8}$ | **85.0**$_{+11.2}$ | **84.5**$_{+13.2}$ | **88.6**$_{+12.6}$ | **89.7**$_{+12.0}$ |

*Base+TOBE: best TOBE variant score w/o other IGL methods; **BestIGL**: best IGL baseline w/o TOBE; **BestIGL+TOBE**: best IGL baseline w/ TOBE;

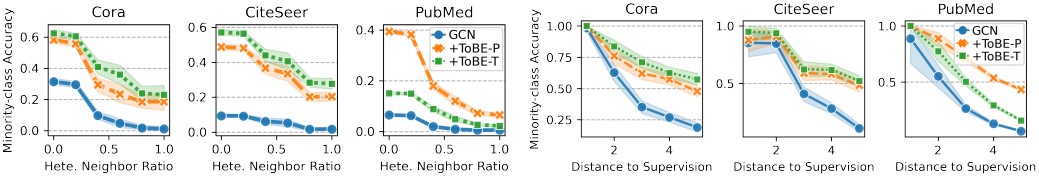

(a) TOBE mitigates AMP in multiple CIGL tasks.     (b) TOBE mitigates DMP in multiple CIGL tasks.

Figure 6: TOBE alleviates the bias from both AMP and DMP via topological augmentation.

niques, which also yields new state-of-the-art performance. In addition to the superior performance in boosting accuracy and macro-F1, TOBE also greatly reduces the model predictive bias. Specifically, we notice that: **(1)** On the basis of the SOTA CIGL baseline, TOBE can handle the topological challenges in CIGL and further boost its performance by a large margin (e.g., for all CIGL techniques with GCN, TOBE brings up to 26.81%/57.82%/13.71% improvement in balanced accuracy on Cora/CiteSeer/PubMed datasets). **(2)** In addition to better classification performance, by mitigating the AMP and DMP, TOBE also greatly reduces the predictive bias in CIGL, with up to 32.75%/57.62%/71.88% reduction in performance deviation on Cora/CiteSeer/PubMed. **(3)** $\text{TOBE}_1$ generally demonstrates better classification performance, while $\text{TOBE}_0$ performs better in terms of reducing performance deviation. Detailed results can be found in Appendix E.

**On the robustness of TOBE (Table 2).** We further extend the results in Table 1 and test TOBE's robustness to varying types and levels of imbalance, as reported in Table 2. In this experiment, we extend the step imbalance ratio from 10 (used in Table 1) to 20 to test TOBE under even more challenging class imbalance scenarios. In addition, we consider the natural (long-tail) class imbalance (Park et al., 2022) that is commonly observed in real-world graphs with IR of 50 and 100. Datasets from (Shchur et al., 2018) (*CS, Physics*) are also included to test TOBE on large-scale tasks. Results show that: **(1)** TOBE is robust to extreme class imbalance, and it consistently boosts the best performance by a significant margin under varying types and levels of imbalance. **(2)** The performance drop from increasing IR is significantly lowered by TOBE, i.e., applying TOBE improves model's robustness to extreme class imbalance. **(3)** TOBE's advantage is even more prominent under higher class imbalance, e.g., on Cora with step IR, the performance gain of applying TOBE on Base is 13.4%/35.2% with IR = 10/20, and similar patterns can be observed in other settings.

**On mitigating AMP and DMP (Fig. 6).** We further design experiments to verify to what extent TOBE can effectively handle the topological challenges identified in this paper, i.e., ambivalent and distant message-passing. Specifically, we investigate whether TOBE can improve the prediction accuracy of minority class nodes that are highly influenced by AMP/DMP, i.e., with high heterophilic neighbor ratio/long distance to supervision. Results are shown in Fig. 6 (5 independent runs with GCN classifier, IR=10). As can be observed, TOBE effectively alleviates the negative impact of AMP and DMP and helps node classifiers to achieve better performance in minority classes.

**Complexity analysis (Table 3).** TOBE introduces $C$ (the number of class) virtual nodes with $\mathcal{O}(n)$ edges. Because of the long-tail distribution of node uncertainty and the discount factor used to solve Eq. (7), only a small portion of nodes have positive risks with relatively few (empirically around 1-3%) virtual edges introduced. For computation, $\text{TOBE}_0/\text{TOBE}_1$ has $\mathcal{O}(|\mathcal{V}|C)/\mathcal{O}(|\mathcal{E}|C)$ space

complexity. Since all the operations can be executed in parallel in matrix form, $\text{ToBE}_0/\text{ToBE}_1$ has $\mathcal{O}(\frac{|\mathcal{V}|C}{D})/\mathcal{O}(\frac{|\mathcal{E}|C}{D})$ time complexity, where $D$ is the number of available computational units. Note that $D$ is usually large for modern GPUs, thus the computation of ToBE is highly efficient in practice. Table 3 reports the ratio of virtual nodes/edges to the original graph introduced and the running time of ToBE on different datasets. We discuss how to further speed up ToBE in practice in Appendix D.2.

Table 3: Efficiency results of $\text{ToBE}_0/\text{ToBE}_1$.

| Dataset | $\Delta$ Nodes (%) | $\Delta$ Edges (%) | $\Delta$ Time (ms) |
|---------|--------------------|--------------------|--------------------|
| Cora | 0.258% | 2.842%/1.509% | 4.50/4.65ms |
| CiteSeer | 0.180% | 3.715%/1.081% | 4.72/4.97ms |
| PubMed | 0.015% | 3.175%/1.464% | 6.23/6.64ms |
| CS | 0.082% | 1.395%/1.053% | 16.97/18.61ms |
| Physics | 0.014% | 0.797%/0.527% | 30.68/31.91ms |

\* Results obtained on an NVIDIA® Tesla V100 32GB GPU.

## 5 RELATED WORKS

**Imbalanced graph learning.** Class imbalance is ubiquitous in many machine-learning tasks and has been extensively studied (He & Garcia, 2008; Krawczyk, 2016). However, most of the existing works focus on i.i.d. scenarios, which may not be tailored to the unique characteristics of graph data. To handle imbalanced graph learning, several techniques have been proposed in recent studies (e.g., by adversarial training (Shi et al., 2020; Qu et al., 2021), designing new GNN architectures (Wang et al., 2020; Liu et al., 2021) or loss functions (Song et al., 2022a)), we review the most closely related model-agnostic CR methods here. One of the early works GraphSMOTE (Zhao et al., 2021b) adopts SMOTE (Chawla et al., 2002) oversampling in the node embedding space to synthesize minority nodes and complements the topology with a learnable edge predictor. A more recent work GraphENS (Park et al., 2022) synthesizes the ego network through saliency-based ego network mixing to handle the neighbor-overfitting problem. Most studies are rooted in a *class-rebalancing* perspective and address the imbalance by node/class-wise reweighting or resampling.

**Topology-imbalance in graphs.** Another recent line of research focuses on the topology imbalance issue, firstly discussed in ReNode (Chen et al., 2021). They found that "the unequal structure role of labeled nodes" can cause influence conflict, and design a method to re-weight the labeled nodes based on a conflict detection measure. Other works further discussed how to better address the issue via position-aware structure learning (Sun et al., 2022), and handle topology-imbalance in fake news detection (Gao et al., 2022) and bankruptcy prediction (Liu et al., 2023). These studies discussed concepts related to "influence conflict/insufficient", which motivated us to investigate AMP/DMP in this work. It is worth mentioning that this line of research is not tailored for handling class-imbalanced graph learning. In this work, we present the first principled study to approach the source of the class-imbalance bias from graph topology, and show that the performance of existing topology-imbalance algorithm (Chen et al., 2021) can be significantly boosted by our solution.

**GNN with heterophilic/long-distance propagation.** Numerous studies exist in the literature concerning learning from heterophilic graphs (Zheng et al., 2022; Xu et al., 2023) and employing multi-hop propagation in learning (Gasteiger et al., 2018; Zhao et al., 2021a). In particular, heterophilic GNNs often combine intermediate representations to derive more refined structure-aware features (Zhu et al., 2020). The GPRGNN (Chien et al., 2020) takes a step further by introducing learnable weights to adaptively combine representations from each layer. Meanwhile, in multi-hop propagation, APPNP (Gasteiger et al., 2018) stands as a representative technique that leverages personalized PageRank for extracting information from a broader neighborhood. These works focus on addressing global graph heterophily and long-distance propagation by modifying the GNN architecture or aggregation operators, and are not specifically designed to address class imbalance. Empirical results show that our method can also significantly boost the performance of such GNNs (Chien et al., 2020; Gasteiger et al., 2018) in various CIGL tasks.

## 6 CONCLUSION

In this paper, we study the class-imbalanced node classification problem from an under-explored topological perspective. We theoretically reveal and empirically observe that two fundamental phenomena in graph topology, i.e., ambivalent and distant message-passing, can greatly exacerbate the predictive bias stemming from class imbalance. In light of this, we propose ToBE to identify the nodes that are critically influenced by such challenges and rectify their learning by dynamic topological augmentation. ToBE is a swift and model-agnostic framework that can seamlessly complement other CIGL techniques, augmenting their performance and mitigating predictive bias. Systematic experiments validate ToBE's superior effectiveness, versatility, robustness, and efficiency across various CIGL tasks. We hope this work provides a new perspective for handling class imbalance and illuminates promising avenues for future research in imbalanced graph learning.

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

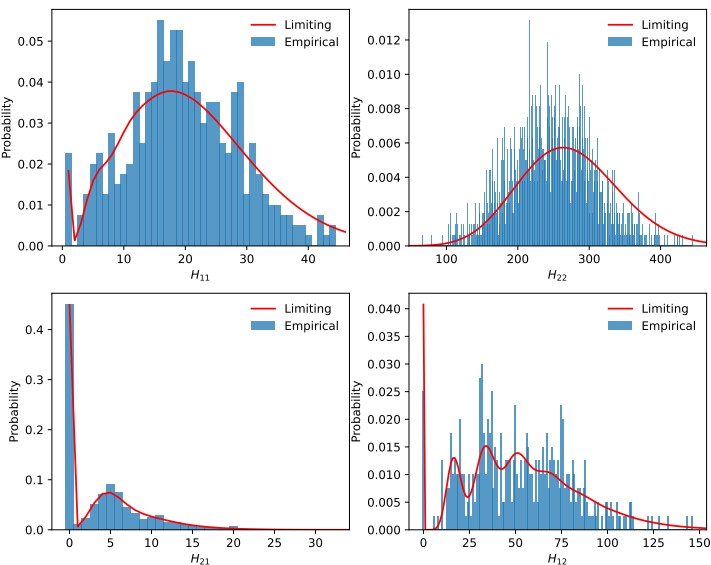

Figure 7: Distributions of $H_{ij}^k$. Simulated with $n = 2000$, $\rho = 4$, $p = 0.01$, $q = 0.002$, $k = 2$.

## A    PROOFS OF THEORETICAL RESULTS

Define random variables $H_{ij}^k := |\mathcal{V}_j \cap \mathcal{H}(u, k)|$ denote the number of class-$j$ $k$-hop homo-connected neighbors of a node $u \in \mathcal{V}_i$. Note that the results of both Theorems 1 & 2 depend only on the distributions of $H_{ij}^k$. Thus, we will first derive the limiting distributions of $H_{ij}^k$ as a technical lemma, and then give the proofs of Theorems 1 & 2.

### A.1    LIMITING DISTRIBUTIONS OF $H_{ij}^k$

To count the number of homo-connected neighbors, consider the breadth-first search (BFS) tree rooted at node $u \in \mathcal{V}_1$. By enumerating the numbers of $1, \ldots, k$-hop homo-connected neighbors in the BFS tree respectively, we can calculate the exact joint distribution of $(H_{11}^k, H_{12}^k)$:

$$
\mathbb{P}\{H_{11}^k = s, H_{12}^k = s'\} = \sum_{\substack{a_1 + \cdots + a_k = s \\ b_1 + \cdots + b_k = s'}} \binom{n_1 - 1}{a_1, \ldots, a_k, n_1 - 1 - s} \binom{n_2}{b_1, \ldots, b_k, n_2 - s'}
$$

$$
p^{a_1} \left( \prod_{t=2}^k (1-p)^{a_t(1 + a_1 + \cdots + a_{t-2})} \big(1 - (1-p)^{a_{t-1}}\big)^{a_t} \right) (1-p)^{(n_1 - 1 - s)(1 + s - a_k)}
$$

$$
q^{b_1} \left( \prod_{t=2}^k (1-q)^{b_t} (1-p)^{b_t(b_1 + \cdots + b_{t-2})} \big(1 - (1-p)^{b_{t-1}}\big)^{b_t} \right) (1-q)^{n_2 - s'} (1-p)^{(n_2 - s')(s' - b_k)}.
$$

Thus, $H_{11}^k$ and $H_{12}^k$ are independent, and their marginal distributions are:

$$\mathbb{P}\{H_{11}^k = s\} = \sum_{a_1 + \cdots + a_k = s} \binom{n_1 - 1}{a_1, \ldots, a_k, n_1 - 1 - s}$$

$$p^{a_1} \left( \prod_{t=2}^{k} (1-p)^{a_t(1+a_1+\cdots+a_{t-2})} (1 - (1-p)^{a_{t-1}})^{a_t} \right) (1-p)^{(n_1-1-s)(1+s-a_k)},$$

$$\mathbb{P}\{H_{12}^k = s\} = \sum_{b_1 + \cdots + b_k = s} \binom{n_2}{b_1, \ldots, b_k, n_2 - s}$$

$$q^{b_1} \left( \prod_{t=2}^{k} (1-q)^{b_t} (1-p)^{b_t(b_1+\cdots+b_{t-2})} (1 - (1-p)^{b_{t-1}})^{b_t} \right) (1-q)^{n_2-s} (1-p)^{(n_2-s)(s-b_k)}.$$

Now consider $n \to \infty$. Let

$$\beta_{11} := \lim_{n \to \infty} n_1 \cdot p = \beta,$$

$$\beta_{22} := \lim_{n \to \infty} n_2 \cdot p = \rho\beta,$$

$$\beta_{21} := \lim_{n \to \infty} n_1 \cdot q = \beta\frac{q}{p},$$

$$\beta_{12} := \lim_{n \to \infty} n_2 \cdot q = \rho\beta\frac{q}{p}.$$

Then, the limiting distributions of $H_{11}^k$ and $H_{12}^k$ are:

$$\mathbb{P}\{H_{11}^k = s\} \sim \sum_{a_1 + \cdots + a_k = s} \frac{n_1^s}{a_1! \cdots a_k!} p^{a_1} \left( \prod_{t=2}^{k} (a_{t-1}p)^{a_t} \right) (1-p)^{n_1(1+s-a_k)}$$

$$\to e^{-\beta_{11}} \sum_{a_1 + \cdots + a_k = s} \frac{(\beta_{11}e^{-\beta_{11}})^{a_1}}{a_1!} \left( \prod_{t=2}^{k-1} \frac{(a_{t-1}\beta_{11}e^{-\beta_{11}})^{a_t}}{a_t!} \right) \frac{(a_{k-1}\beta_{11})^{a_k}}{a_k!},$$

$$\mathbb{P}\{H_{12}^k = s\} \sim \sum_{b_1 + \cdots + b_k = s} \frac{n_2^s}{b_1! \cdots b_k!} q^{b_1} \left( \prod_{t=2}^{k} (b_{t-1}p)^{b_t} \right) (1-q)^{n_2} (1-p)^{n_2(s-b_k)}$$

$$\to e^{-\beta_{12}} \sum_{b_1 + \cdots + b_k = s} \frac{(\beta_{12}e^{-\beta_{22}})^{b_1}}{b_1!} \left( \prod_{t=2}^{k-1} \frac{(b_{t-1}\beta_{22}e^{-\beta_{22}})^{b_t}}{b_t!} \right) \frac{(b_{k-1}\beta_{22})^{b_k}}{b_k!}.$$

Figure 7 shows that the limiting distributions are indeed a good approximation for finite $n$.

### A.2 PROOF OF THEOREM 1

Note that for any $t' = 1, \ldots, k$,

$$e^{-\beta_{11}} \sum_{a_1=0}^{\infty} \cdots \sum_{a_k=0}^{\infty} a_{t'} \cdot \frac{(\beta_{11}e^{-\beta_{11}})^{a_1}}{a_1!} \left( \prod_{t=2}^{k-1} \frac{(a_{t-1}\beta_{11}e^{-\beta_{11}})^{a_t}}{a_t!} \right) \frac{(a_{k-1}\beta_{11})^{a_k}}{a_k!} = \beta_{11}^{t'}.$$

Thus,

$$
\lim_{n\to\infty} \mathbb{E}[H_{11}^k] = \sum_{s=0}^{\infty} s \cdot \lim_{n\to\infty} \mathbb{P}\{H_{11}^k = s\}
$$

$$
= \sum_{s=0}^{\infty} s \cdot \mathrm{e}^{-\beta_{11}} \sum_{a_1+\cdots+a_k=s} \frac{(\beta_{11}\mathrm{e}^{-\beta_{11}})^{a_1}}{a_1!} \left( \prod_{t=2}^{k-1} \frac{(a_{t-1}\beta_{11}\mathrm{e}^{-\beta_{11}})^{a_t}}{a_t!} \right) \frac{(a_{k-1}\beta_{11})^{a_k}}{a_k!}
$$

$$
= \sum_{s=0}^{\infty} \mathrm{e}^{-\beta_{11}} \sum_{a_1+\cdots+a_k=s} s \cdot \frac{(\beta_{11}\mathrm{e}^{-\beta_{11}})^{a_1}}{a_1!} \left( \prod_{t=2}^{k-1} \frac{(a_{t-1}\beta_{11}\mathrm{e}^{-\beta_{11}})^{a_t}}{a_t!} \right) \frac{(a_{k-1}\beta_{11})^{a_k}}{a_k!}
$$

$$
= \mathrm{e}^{-\beta_{11}} \sum_{a_1=0}^{\infty} \cdots \sum_{a_k=0}^{\infty} (a_1 + \cdots + a_k) \cdot \frac{(\beta_{11}\mathrm{e}^{-\beta_{11}})^{a_1}}{a_1!} \left( \prod_{t=2}^{k-1} \frac{(a_{t-1}\beta_{11}\mathrm{e}^{-\beta_{11}})^{a_t}}{a_t!} \right) \frac{(a_{k-1}\beta_{11})^{a_k}}{a_k!}
$$

$$
= \sum_{t'=1}^{k} \mathrm{e}^{-\beta_{11}} \sum_{a_1=0}^{\infty} \cdots \sum_{a_k=0}^{\infty} a_{t'} \cdot \frac{(\beta_{11}\mathrm{e}^{-\beta_{11}})^{a_1}}{a_1!} \left( \prod_{t=2}^{k-1} \frac{(a_{t-1}\beta_{11}\mathrm{e}^{-\beta_{11}})^{a_t}}{a_t!} \right) \frac{(a_{k-1}\beta_{11})^{a_k}}{a_k!}
$$

$$
= \sum_{t'=1}^{k} \beta_{11}^{t'}.
$$

Similarly,

$$
\lim_{n\to\infty} \mathbb{E}[H_{22}^k] = \sum_{t=1}^{k} \beta_{22}^t,
$$

$$
\lim_{n\to\infty} \mathbb{E}[H_{12}^k] = \sum_{t=1}^{k} \beta_{12}\beta_{22}^{t-1},
$$

$$
\lim_{n\to\infty} \mathbb{E}[H_{21}^k] = \sum_{t=1}^{k} \beta_{21}\beta_{11}^{t-1}.
$$

It follows that

$$
\lim_{n\to\infty} \frac{\alpha_1^k}{\alpha_2^k} = \lim_{n\to\infty} \frac{\mathbb{E}[H_{12}^k]/\mathbb{E}[H_{11}^k]}{\mathbb{E}[H_{21}^k]/\mathbb{E}[H_{22}^k]}
$$

$$
= \frac{\sum_{t=1}^{k} \beta_{12}\beta_{22}^{t-1} / \sum_{t=1}^{k} \beta_{11}^t}{\sum_{t=1}^{k} \beta_{21}\beta_{11}^{t-1} / \sum_{t=1}^{k} \beta_{22}^t}
$$

$$
= \frac{\beta_{12}\beta_{22}}{\beta_{21}\beta_{11}} \cdot \frac{\left( \sum_{t=1}^{k} \beta_{22}^{t-1} \right)^2}{\left( \sum_{t=1}^{k} \beta_{11}^{t-1} \right)^2}
$$

$$
= \left( \rho \cdot \frac{\sum_{t=1}^{k} (\rho\beta)^{t-1}}{\sum_{t=1}^{k} \beta^{t-1}} \right)^2. \qquad \square
$$

### A.3 Proof of Theorem 2

For $k = 2$, note the identity:

$$
\sum_{s=0}^{\infty} \sum_{a=0}^{s} \frac{\lambda^a (\mu a)^{s-a}}{a!(s-a)!} = \sum_{a=0}^{\infty} \frac{\lambda^a}{a!} \sum_{b=0}^{\infty} \frac{(\mu a)^b}{b!} = \sum_{a=0}^{\infty} \frac{\lambda^a \mathrm{e}^{\mu a}}{a!} = \mathrm{e}^{\lambda \mathrm{e}^{\mu}}.
$$

It follows that (with $\lambda = (1 - r_1^{\mathrm{L}})\beta_{11}\mathrm{e}^{-\beta_{11}}$ and $\mu = (1 - r_1^{\mathrm{L}})\beta_{11}$)

$$
\begin{aligned}
\lim_{n\to\infty} \mathbb{E}[(1 - r_1^{\mathrm{L}})^{H_{11}^2}] &= \sum_{s=0}^{\infty}(1 - r_1^{\mathrm{L}})^s \cdot \lim_{n\to\infty}\mathbb{P}\{H_{11}^2 = s\} \\
&= \sum_{s=0}^{\infty}(1 - r_1^{\mathrm{L}})^s \cdot \mathrm{e}^{-\beta_{11}}\sum_{a=0}^{s}\frac{(\beta_{11}\mathrm{e}^{-\beta_{11}})^a(\beta_{11}a)^{s-a}}{a!(s-a)!} \\
&= \mathrm{e}^{-\beta_{11}}\sum_{s=0}^{\infty}\sum_{a=0}^{s}\frac{((1 - r_1^{\mathrm{L}})\beta_{11}\mathrm{e}^{-\beta_{11}})^a((1 - r_1^{\mathrm{L}})\beta_{11}a)^{s-a}}{a!(s-a)!} \\
&= \mathrm{e}^{-\beta_{11}}\mathrm{e}^{(1 - r_1^{\mathrm{L}})\beta_{11}\mathrm{e}^{-\beta_{11}}\mathrm{e}^{(1 - r_1^{\mathrm{L}})\beta_{11}}} \\
&= \mathrm{e}^{-(1 - (1 - r_1^{\mathrm{L}})\mathrm{e}^{-r_1^{\mathrm{L}}\beta_{11}})\beta_{11}} \\
&\approx \mathrm{e}^{-\beta_{11}}.
\end{aligned}
$$

For $k = 3$, similarly,

$$
\lim_{n\to\infty}\mathbb{E}[(1 - r_1^{\mathrm{L}})^{H_{11}^3}] = \mathrm{e}^{-\left(1 - (1 - r_1^{\mathrm{L}})\beta_{11}\mathrm{e}^{-\left(1 - (1 - r_1^{\mathrm{L}})\beta_{11}\mathrm{e}^{-r_1^{\mathrm{L}}\beta_{11}}\right)\beta_{11}}\right)\beta_{11}} \approx \mathrm{e}^{-\beta_{11}}.
$$

In general, the result for $k$ has $k$ nested exponentiations, but we still have:

$$
\lim_{n\to\infty}\mathbb{E}[(1 - r_1^{\mathrm{L}})^{H_{11}^k}] \approx \mathrm{e}^{-\beta_{11}}.
$$

Similarly,

$$
\lim_{n\to\infty}\mathbb{E}[(1 - r_2^{\mathrm{L}})^{H_{12}^k}] \approx \mathrm{e}^{-\beta_{12}},
$$
$$
\lim_{n\to\infty}\mathbb{E}[(1 - r_2^{\mathrm{L}})^{H_{22}^k}] \approx \mathrm{e}^{-\beta_{22}},
$$
$$
\lim_{n\to\infty}\mathbb{E}[(1 - r_1^{\mathrm{L}})^{H_{21}^k}] \approx \mathrm{e}^{-\beta_{21}}.
$$

By the law of total probability and the independence of $H_{i1}^k$ and $H_{i2}^k$,

$$
\begin{aligned}
\frac{\delta_1^k}{\delta_2^k} &= \frac{\mathbb{E}[(1 - r_1^{\mathrm{L}})^{H_{11}^k+1}(1 - (1 - r_2^{\mathrm{L}})^{H_{12}^k})]}{\mathbb{E}[(1 - r_2^{\mathrm{L}})^{H_{22}^k+1}(1 - (1 - r_1^{\mathrm{L}})^{H_{21}^k})]} \\
&= \frac{(1 - r_1^{\mathrm{L}})\mathbb{E}[(1 - r_1^{\mathrm{L}})^{H_{11}^k}](1 - \mathbb{E}[(1 - r_2^{\mathrm{L}})^{H_{12}^k}])}{(1 - r_2^{\mathrm{L}})\mathbb{E}[(1 - r_2^{\mathrm{L}})^{H_{22}^k}](1 - \mathbb{E}[(1 - r_1^{\mathrm{L}})^{H_{21}^k}])}.
\end{aligned}
$$

It follows that

$$
\begin{aligned}
\lim_{n\to\infty}\frac{\delta_1^k}{\delta_2^k} &\approx \frac{(1 - r_1^{\mathrm{L}})\mathrm{e}^{-\beta_{11}}(1 - \mathrm{e}^{-\beta_{12}})}{(1 - r_2^{\mathrm{L}})\mathrm{e}^{-\beta_{22}}(1 - \mathrm{e}^{-\beta_{21}})} \\
&\approx \frac{1 - r_1^{\mathrm{L}}}{1 - r_2^{\mathrm{L}}}\mathrm{e}^{\beta_{22}-\beta_{11}} = \frac{1 - r_1^{\mathrm{L}}}{1 - r_2^{\mathrm{L}}}\mathrm{e}^{(\rho-1)\beta}. \qquad \square
\end{aligned}
$$

# B  THE TOBE ALGORITHM

---

**Algorithm 1** TOBE: Topological Balanced Augmentation

---

**Require:** $\mathcal{G} : \{\boldsymbol{A}, \boldsymbol{X}\}$ with imbalanced labeled set $\mathcal{V}^{\mathrm{L}}$;

1: **Initialize:** node classifier $F(\cdot; \Theta)$;
2: **while** not converged **do**
3:      $\hat{\boldsymbol{P}} \leftarrow F(\boldsymbol{A}, \boldsymbol{X}; \Theta)$;
4:      $\hat{\boldsymbol{y}} \leftarrow \mathrm{argmax}_{axis=1}(\hat{\boldsymbol{P}})$;
5:      $\boldsymbol{r} \leftarrow \texttt{NodeRiskEstimation}(\hat{\boldsymbol{P}}, \hat{\boldsymbol{y}})$;             $\triangleright$ Eq. (3) - (4)
6:      $\hat{\boldsymbol{S}} \leftarrow \texttt{PosteriorEstimation}(\boldsymbol{A}, \hat{\boldsymbol{P}}, \hat{\boldsymbol{y}})$;         $\triangleright$ Eq. (5) - (6)
7:      **for** class $j = 1$ to $C$ **do**
8:          $\boldsymbol{x}_{v_j^*} \leftarrow \sum_{v \in \hat{\mathcal{V}}_j} \boldsymbol{x}_v / |\hat{\mathcal{V}}_j|$
9:          $y_{v_j^*} \leftarrow j$
10:        $v_j^* : (\boldsymbol{x}_{v_j^*}, y_{v_j^*})$            $\triangleright$ Synthesize virtual node $v_j^*$ for each class $j$
11:      $\mathcal{V}^* \leftarrow \{v_j^* | 1 \leq j \leq C\}$               $\triangleright$ Get virtual node set $\mathcal{V}^*$
12:      $\boldsymbol{Q}^* \leftarrow \hat{\boldsymbol{S}} \odot \gamma$            $\triangleright$ Get virtual link probabilities $\boldsymbol{Q}^*$ by Eq. (7)
13:      $\mathcal{E}^* \sim \boldsymbol{Q}^*$;                $\triangleright$ Sample virtual edges $\mathcal{E}^*$ w.r.t $\boldsymbol{Q}^*$
14:      Derive $\boldsymbol{X}^*, \boldsymbol{A}^*$ from $\mathcal{V} \cup \mathcal{V}^*, \mathcal{E} \cup \mathcal{E}^*$;
15:      Update $\Theta$ with augmented graph $\mathcal{G}^* : \{\boldsymbol{A}^*, \boldsymbol{X}^*\}$;
16: **return** a balanced classifier $F(\boldsymbol{A}, \boldsymbol{X}; \Theta)$;

---

# C  REPRODUCIBILITY

In this section, we describe the detailed experimental settings including (**§C.1**) data statistics, (**§C.2**) baseline settings, and (**§C.3**) evaluation protocols. The source code for implementing and evaluating TOBE and all the CIGL baseline methods will be released after the paper is published.

## C.1  DATA STATISTICS

As previously described, we adopt 5 benchmark graph datasets: the Cora, CiteSeer, and PubMed citation networks (Sen et al., 2008), and the CS and Physics coauthor networks (Shchur et al., 2018) to test TOBE on large graphs with more nodes and high-dimensional features. All datasets are publicly available[3]. Table 4 summarizes the dataset statistics.

Table 4: Statistics of datasets.

| Dataset | #nodes | #edges | #features | #classes |
|---------|--------|--------|-----------|----------|
| Cora | 2,708 | 10,556 | 1,433 | 7 |
| CiteSeer | 3,327 | 9,104 | 3,703 | 6 |
| PubMed | 19,717 | 88,648 | 500 | 3 |
| CS | 18,333 | 163,788 | 6,805 | 15 |
| Physics | 34,493 | 495,924 | 8,415 | 5 |

We follow previous works (Zhao et al., 2021b; Park et al., 2022; Song et al., 2022a) to construct and adjust the class imbalanced node classification tasks. For step imbalance, we select half of the classes ($\lfloor m/2 \rfloor$) as minority classes and the rest as majority classes. We follow the public split (Sen et al., 2008) for semi-supervised node classification where each class has 20 training nodes, then randomly remove minority class training nodes until the given imbalance ratio (IR) is met. The imbalance ratio is defined as IR $= \frac{\text{\#majority training nodes}}{\text{\#minority training nodes}} \in [1, \infty)$, i.e., more imbalanced data has higher IR. For natural imbalance, we simulate the long-tail class imbalance present in real-world data by utilizing a power-law distribution. Specifically, for a given IR, the largest head class have $n_{\text{head}} = $ IR training nodes, and the smallest tail class have 1 training node. The number of training

---

[3]https://pytorch-geometric.readthedocs.io/en/latest/modules/datasets.html.

nodes of the $k$-th class is determined by $n_k = \lfloor n_{\text{head}}^{\lambda_k} \rfloor, \lambda_k = \frac{m-k}{m-1}$. We set the IR (largest class to smallest class) to 50/100 to test ToBE's robustness under natural and extreme class imbalance. We show the training data distribution under step and natural imbalance in Fig. 8.

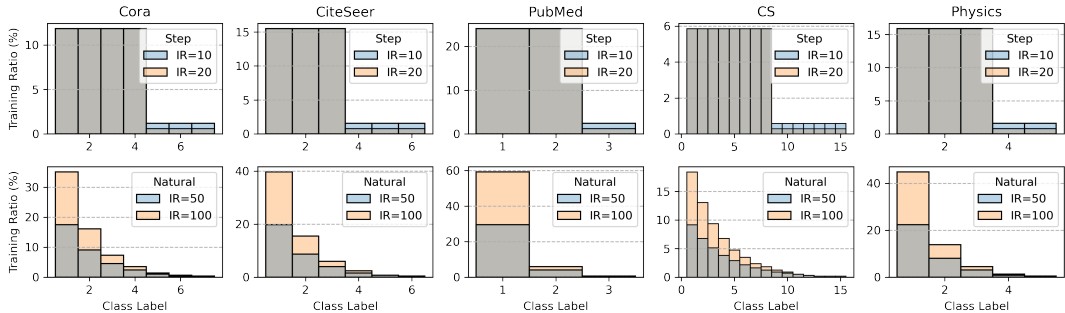

Figure 8: Class distribution of training datasets under step and natural imbalance.

## C.2 BASELINE SETTINGS

To fully validate ToBE's performance and compatibility with existing CIGL techniques and GNN backbones, we include six baseline methods with five popular GNN backbones in our experiments, and combine ToBE with them under all possible combinations. The included CIGL baselines can be generally divided into two categories: reweighting-based (i.e., Reweight (Japkowicz & Stephen, 2002), ReNode (Chen et al., 2021)) and augmentation-based (i.e., Oversampling (Japkowicz & Stephen, 2002), SMOTE (Chawla et al., 2002), GraphSMOTE (Zhao et al., 2021b), and GraphENS (Park et al., 2022)).

- Reweight (Japkowicz & Stephen, 2002) assigns minority classes with higher misclassification costs (i.e., weights in the loss function) by the inverse of the class frequency in the training set.

- ReNode (Chen et al., 2021) measures the influence conflict of training nodes, and perform instance-wise node reweighting to alleviate the topology imbalance.

- Oversample (Japkowicz & Stephen, 2002) augments minority classes with additional synthetic nodes by replication-base oversampling.

- SMOTE (Chawla et al., 2002) synthesizes minority nodes by 1) randomly selecting a seed node, 2) finding its $k$-nearest neighbors in the feature space, and 3) performing linear interpolation between the seed and one of its $k$-nearest neighbors.

- GraphSMOTE (Zhao et al., 2021b) extends SMOTE (Chawla et al., 2002) to graph-structured data by 1) performing SMOTE in the low-dimensional embedding space of GNN and 2) utilizing a learnable edge predictor to generate better topology connections for synthetic nodes.

- GraphENS (Park et al., 2022) directly synthesize the whole ego network (node with its 1-hop neighbors) for minority classes by similarity-based ego network combining and saliency-based node mixing to prevent neighbor memorization.

We use the public implementations[456] of the baseline methods for a fair comparison. For ReNode (Chen et al., 2021), we use its transductive version and search hyperparameters among the lower bound of cosine annealing $w_{min} \in \{0.25, 0.5, 0.75\}$ and upper bound of the cosine annealing $w_{max} \in \{1.25, 1.5, 1.75\}$ following the original paper. We set the teleport probability of PageRank $\alpha = 0.15$ as given by the default setting in the released implementation. As Oversample (Cui et al., 2019) and SMOTE (Chawla et al., 2002) were not proposed to handle graph data, we adopt their enhanced versions provided by GraphSMOTE (Zhao et al., 2021b), which also duplicate the edges from the seed nodes to the synthesized nodes in order to connect them to the

---

[4] https://github.com/victorchen96/renode
[5] https://github.com/TianxiangZhao/GraphSmote
[6] https://github.com/JoonHyung-Park/GraphENS

graph. For GraphSMOTE (Zhao et al., 2021b), we use the version that predicts edges with binary values as it performs better than the variant with continuous edge predictions in many datasets. For GraphENS (Park et al., 2022), we follow the settings in the paper: $\text{Beta}(2, 2)$ distribution is used to sample $\lambda$, the feature masking hyperparameter $k$ and temperature $\tau$ are tuned among $\{1, 5, 10\}$ and $\{1, 2\}$, and the number of warmup epochs is set to 5.

Since TOBE only manipulates the graph data and remains independent of the model architecture, it seamlessly integrates with the aforementioned CIGL techniques. During each training epoch, TOBE enhances the original graph $\mathcal{G}$ using the current model $f$ and yields the augmented graph $\mathcal{G}^*$. Subsequently, other CIGL methods operate on the augmented graph $\mathcal{G}^*$, specifically:

- Loss function engineering methods (Reweight and ReNode) perform loss computation and backpropagation based on $\mathcal{G}^*$.
- Data augmentation methods (Resampling, SMOTE, GSMOTE, GENS) carry out additional class-balancing operations on $\mathcal{G}^*$, generating new minority nodes based on its structure.

We use `pytorch` (Paszke et al., 2019) and `torch_geometric` (Fey & Lenssen, 2019) to implement all five GNN backbones used in this paper, i.e., GCN (Welling & Kipf, 2016), GAT (Veličković et al., 2018), GraphSAGE (Hamilton et al., 2017), APPNP (Gasteiger et al., 2018), and GPRGNN (Chien et al., 2020). Most of our settings are aligned with prevailing works (Park et al., 2022; Chen et al., 2021; Song et al., 2022a) to obtain fair and comparable results. Specifically, we implement all GNNs' convolution layer with ReLU activation and dropout (Srivastava et al., 2014) with a dropping rate of 0.5 before the last layer. For GAT, we set the number of attention heads to 4. For APPNP and GPRGNN, we follow the best setting in the original paper and use 2 `APPNP`/`GPR_prop` convolution layers with 64 hidden units. Note that GraphENS's official implementation requires modifying the graph convolution for resampling and thus cannot be directly combined with APPNP and GPRGNN. The teleport probability = 0.1 and the number of power iteration steps K = 10. We search for the best architecture for other backbones from #layers $l \in \{1, 2, 3\}$ and hidden dimension $d \in \{64, 128, 256\}$ based on the average of validation accuracy and F1 score. We train each GNN for 2,000 epochs using Adam optimizer (Kingma & Ba, 2014) with an initial learning rate of 0.01. To achieve better convergence, we follow (Park et al., 2022) to use 5e-4 weight decay and adopt the `ReduceLROnPlateau` scheduler in Pytorch, which reduces the learning rate by half if the validation loss does not improve for 100 epochs.

### C.3 EVALUATION PROTOCOL

To evaluate the predictive performance on class-imbalanced data, we use two balanced metrics, i.e., balanced accuracy (*BAcc.*) and macro-averaged F1 score (*Macro-F1*). They compute accuracy/F1-score for each class independently and use the unweighted average mean as the final score, i.e., *BAcc.* $= \frac{1}{m} \sum_{i=1}^{m} Acc(c_i)$, *Macro-F1* $= \frac{1}{m} \sum_{i=1}^{m} F1(c_i)$. Additionally, we use performance standard deviation (PerfStd) to evaluate the level of model predictive bias. Formally, let $Acc(c_i)$ be the classification accuracy of class $c_i$, the PerfStd is defined as the standard deviation of the accuracy scores of all classes, i.e., $\sqrt{\frac{1}{m} \sum_{i=1}^{m} (Acc(c_i) - BAcc.)^2}$. All the experiments are conducted on a Linux server with Intel® Xeon® Gold 6240R CPU and NVIDIA® Tesla V100 32GB GPU.

## D EXTENDED DISCUSSIONS

In this section, we present an ablation study (**§D.1**) validate the effectiveness and efficiency of the key modules, then we discuss how to further speed up TOBE in practice (**§D.2**); how TOBE alleviates overfitting (**§D.3**); the inductive capability of TOBE (**§D.4**); remarks on combining TOBE with multi-step message-passing GNNs (**§D.5**); further comparison with other baselines (**§D.6**); how to choose between $\text{TOBE}_0$ and $\text{TOBE}_1$ in practice (**§D.7**); and finally, the limitation and future works (**§D.8**).

### D.1 ABLATION STUDY

We present an ablation study to validate the effectiveness and efficiency of the key modules in TOBE. Specifically, for node risk estimation, we compare our total-variation-distance-based uncertainty with (i) the naïve random assignment that drawn uncertainty score from a uniform distribution

$U(0, 1)$ and (ii) the information entropy $\mathbf{H}(Y) = -\sum_{y \in \mathcal{Y}} p(y) \log_2(p(y))$. We substitute the original uncertainty metric with these aforementioned methods in $\text{ToBE}_0$, and assess their impact on performance as well as the computational time required for uncertainty estimation. It is worth noting that in practical implementation, the computation is parallelized on GPU (assuming sufficient GPU memory). Therefore, the computational time of a given uncertainty measure remains consistent across the three datasets we employed (Cora, CiteSeer, PubMed with IR=10). The detailed results are presented in Table 5, revealing that: (i) Randomly assigned uncertainty scores significantly impede the performance of ToBE, resulting in a large drop in both balanced accuracy and Marco-f1. (ii) In comparison to our approach, employing information entropy as the node uncertainty score necessitates $\sim$2.3x computation time, yet the influence on performance remains marginal.

Table 5: Ablation study on node risk estimation of ToBE.

| Uncertainty | Cora | | CiteSeer | | PubMed | | Computation Time(ms) |
|---|---|---|---|---|---|---|---|
| | BAcc | Macro-F1 | BAcc | Macro-F1 | BAcc | Macro-F1 | |
| Random | 61.64±1.89 | 59.44±1.71 | 46.59±2.29 | 44.37±3.23 | 61.60±1.69 | 58.13±1.81 | 0.0249 |
| Information Entropy | 65.18±1.68 | 63.11±1.91 | 51.87±2.96 | 50.36±3.43 | 67.72±1.27 | 67.19±1.57 | 0.1257 |
| TVDistance (ours) | 65.54±1.25 | 63.28±1.07 | 52.65±1.08 | 51.55±1.28 | 68.62±0.77 | 67.16±1.53 | 0.0543 |

Further, we conduct an ablation study for our posterior likelihood estimation strategy by comparing our $0^{\text{th}}$-order ($\text{ToBE}_0$) and $1^{\text{st}}$-order ($\text{ToBE}_1$) likelihood estimation methods with the random method that assigns (unnormalized) node-class likelihood by drawing from a uniform distribution $U(0, 1)$. Results are shown in Table 6. We can observe that the random method significantly worsens the predictive performance on all CIGL tasks. Altogether, the ablation study results confirm the effectiveness and efficiency of the design of ToBE, showcasing its ability to deliver strong performance with minimal computational overhead.

Table 6: Ablation study on posterior likelihood estimation of ToBE.

| Estimation | Cora | | CiteSeer | | PubMed | | Computation Time(ms) |
|---|---|---|---|---|---|---|---|
| | BAcc | Macro-F1 | BAcc | Macro-F1 | BAcc | Macro-F1 | |
| Random | 63.85±2.17 | 61.94±2.68 | 46.51±3.27 | 41.70±4.33 | 64.32±1.23 | 53.58±2.48 | 0.0883 |
| 0th Order ($\text{ToBE}_0$) | 65.54±1.25 | 63.28±1.07 | 52.65±1.08 | 51.55±1.28 | 68.62±0.77 | 67.16±1.53 | 0.1251 |
| 1st Order ($\text{ToBE}_1$) | 69.80±1.30 | 68.68±1.49 | 55.37±1.39 | 54.94±1.44 | 67.57±3.22 | 64.40±3.68 | 0.3030 |

## D.2 ON THE FURTHER SPEEDUP OF ToBE

As stated in the paper, thanks to its simple and efficient design, ToBE can be integrated into the GNN training process to perform dynamic topology augmentation based on the training state. By default, we run ToBE in every iteration of GNN training, i.e., the granularity of applying ToBE is 1, as described in Alg. 1. However, we note that in practice, this granularity can be increased to further reduce the cost of applying ToBE. This operation can result in a significant linear speedup ratio: setting the granularity to $N$ reduces the computational overhead of ToBE to $1/N$ of the original (i.e., $N$x speedup ratio), with minor performance degradation. This could be helpful for scaling ToBE to large-scale graphs in practice. In this section, we design experiments to validate the influence of different ToBE granularity (i.e., the number of iterations per each use of ToBE) in real-world CIGL tasks. We set the granularity to 1/5/10/50/100 and test the performance of $\text{ToBE}_T$ with a vanilla GCN classifier on the Cora/CiteSeer/PubMed dataset with an imbalance ratio of 10. Fig. 9 shows the empirical results from 10 independent runs. The red horizontal line in each subfigure represents the baseline (vanilla GCN) performance. It can be observed that setting a larger ToBE granularity is an effective way to further speed up ToBE in practice. The performance drop of adopting this trick is relatively minor, especially considering the significant linear speedup ratio it brings. The predictive performance boost brought by ToBE is still significant even with a large granularity at 100 (i.e., with 100x ToBE speedup).

## D.3 ToBE ALLEVIATES NODE AND NEIGHBOR OVERFITTING

Prior studies have demonstrated that the minority class is prone to overfitting due to limited training samples (Johnson & Khoshgoftaar, 2019), and well-designed oversampling methods can address this

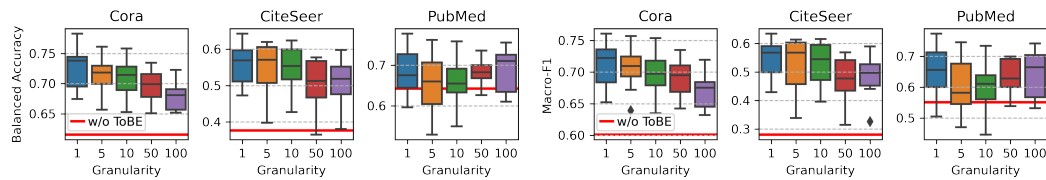

(a) Influence of TOBE granularity on BAcc.  (b) Influence of TOBE granularity on Macro-F1.

Figure 9: Influence of the TOBE granularity (i.e., the number of iterations per each use of TOBE). Note that this brings a linear speedup ratio in practice, e.g., granularity = 100 ⇔ 100x speedup.

Table 7: The node replacing and neighbor replacing experiment results.

| Dataset | Cora | | | CiteSeer | | | PubMed | | |
|---|---|---|---|---|---|---|---|---|---|
| Setting | Base | +TOBE$_0$ | +TOBE$_1$ | Base | +TOBE$_0$ | +TOBE$_1$ | Base | +TOBE$_0$ | +TOBE$_1$ |
| Training Acc | 1.000 | 1.000 | 1.000 | 1.000 | 1.000 | 1.000 | 1.000 | 1.000 | 1.000 |
| w/ Node Rep. | 0.403±0.203 | 0.755±0.207 | 0.839±0.160 | 0.600±0.203 | 0.970±0.070 | 0.858±0.243 | 0.547±0.367 | 0.934±0.172 | 0.926±0.181 |
| w/ Neighbor Rep. | 0.762±0.195 | 0.924±0.110 | 0.917±0.110 | 0.604±0.239 | 0.981±0.055 | 0.842±0.197 | 0.449±0.354 | 0.915±0.202 | 0.906±0.211 |

by synthesizing diverse minority samples (Gosain & Sardana, 2017). While TOBE doesn't directly synthesize samples, it dynamically augments the topological structure, enhancing the diversity of neighbors for minority class nodes. Consequently, TOBE effectively mitigates node and neighbor memorization issues (Park et al., 2022) (i.e., overfitting on the graph training data) induced by static message-passing. To empirically demonstrate TOBE's ability to prevent overfitting on the training nodes, we conduct further experiments with node-replacing and neighbor-replacing:

- Node replacing: We replace the features of minority class training (seen) nodes with features from unseen nodes of the same class. This test assesses the model's vulnerability to node feature overfitting.
- Neighbor replacing: In graph data, message-passing-based GNNs may overfit the neighborhood of minority class nodes (neighbor memorization [4]). To address this, we retained the features of minority training nodes but replaced their neighbors with unseen nodes. This test evaluates the model's susceptibility to neighbor set overfitting.

Following (Park et al., 2022), we adopt GraphSAGE (Hamilton et al., 2017) as the GNN backbone and test its accuracy drop on the training set under different settings. We do the same test after adding TOBE$_0$ and TOBE$_1$ to see whether TOBE can help prevent overfitting (i.e., has smaller accuracy drop under node/neighbor replacing). The results are shown in Table 7. We can observe that While GNN can attain 100% accuracy on the training set, its predictive performance on training nodes significantly drops after node/neighbor replacing, indicating substantial overfitting to features and neighbor sets. However, the same GNN trained with TOBE augmentation has significantly smaller performance degradation, showcasing the effectiveness of TOBE in preventing overfitting.

## D.4 THE INDUCTIVE CAPABILITY OF TOBE

Table 8: The inductive capability of TOBE.

| Metric & Method | | PPI-1 (IR=2.054) | PPI-2 (IR=3.051) | PPI-3 (IR=3.835) | PPI-4 (IR=3.010) | PPI-5 (IR=6.963) |
|---|---|---|---|---|---|---|
| BAcc.↑ | GCN | 73.52±1.26 | 72.06±3.81 | 69.79±2.82 | 74.61±3.16 | 71.70±3.85 |
| | GCN+TOBE$_0$ | **74.89**±1.64 (+1.36) | 75.47±2.34 (+3.41) | 70.17±3.20 (+0.38) | **77.64**±2.76 (+3.03) | 74.98±3.98 (+3.28) |
| | GCN+TOBE$_1$ | 73.97±1.44 (+0.44) | **76.56**±2.82 (+4.49) | **73.43**±3.14 (+3.63) | 76.34±3.13 (+1.73) | **75.12**±3.74 (+3.42) |
| F1↑ | GCN | 73.85±0.34 | 74.30±3.22 | 71.84±1.64 | 75.33±1.77 | 73.18±1.92 |
| | GCN+TOBE$_0$ | **75.23**±0.80 (+1.39) | 77.07±1.07 (+2.77) | 72.73±2.24 (+0.89) | **76.85**±1.34 (+1.51) | **76.33**±1.49 (+3.15) |
| | GCN+TOBE$_1$ | 74.86±0.91 (+1.01) | **77.58**±1.54 (+3.28) | **74.90**±1.62 (+3.06) | 76.48±1.40 (+1.15) | 76.28±1.76 (+3.11) |

TOBE improves the ability of Graph Neural Networks (GNNs) to learn more effective representations for minority classes by addressing misclassifications induced by AMP/DMP. While we have previously validated the superior performance of TOBE in transductive node classification, it is important to note that the enhanced representation facilitated by TOBE also aids GNNs in generalizing more effectively to unseen graphs. In this section, we present additional experiments and analyses to confirm the inductive learning capabilities of TOBE.

We evaluate the inductive capabilities of TOBE on protein-protein interaction networks (PPI) (Zitnik & Leskovec, 2017). The objective is to predict protein functions for unseen proteins (graphs). The PPI dataset encompasses multiple distinct protein function targets, resulting in *binary classification tasks with varying class imbalance ratios*. This configuration serves as a good testbed to assess TOBE's effectiveness in mitigating class imbalance specifically on unseen graphs. We select the initial 5 protein functions as the prediction targets. Each task involves training on the first 5 graphs, and testing the model on the remaining 15 graphs. Following the same procedure in reporting the main results, we conduct 5 independent runs to eliminate randomness.

The results are presented in Table 8. It is evident that TOBE contributes to enhanced generalization of GNN to unseen graphs, particularly in tasks with high class imbalance ratios (e.g., on PPI-5 with IR=6.963, TOBE brings consistent 3+% improvement in both metrics). We attribute this improvement to TOBE's capacity to rectify misclassifications within minority classes, thereby facilitating GNN in acquiring more effective representations for these minority classes.

## D.5 REMARKS ON COMBINING TOBE WITH MULTI-STEP MESSAGE-PASSING GNNS

Table 9: Case study on the propagation steps $k$ of multi-step message-passing GNNs.

| Metric | BAcc. | | | Macro-F1 | | |
|---|---|---|---|---|---|---|
| | Model: APPNP | | | | | |
| Setting | Base | +TOBE$_0$ | +TOBE$_1$ | Base | +TOBE$_0$ | +TOBE$_1$ |
| K=10 | **73.74**±1.12 | 75.02±1.54 | 72.15±0.76 | **73.67**±0.98 | 73.67±1.18 | 69.79±0.72 |
| K=5 | 72.64±1.00 | **75.43**±1.46 | 73.35±1.06 | 72.86±1.06 | 74.22±1.35 | 72.66±0.76 |
| K=3 | 71.82±0.86 | 74.43±1.30 | **74.73**±1.06 | 71.80±0.84 | **75.50**±1.07 | **74.10**±1.00 |
| K=1 | 64.69±1.57 | 72.62±1.18 | 73.51±1.24 | 64.34±2.07 | 73.64±1.14 | 73.33±1.30 |
| | Model: GPRGNN | | | | | |
| Setting | Base | +TOBE$_0$ | +TOBE$_1$ | Base | +TOBE$_0$ | +TOBE$_1$ |
| K=10 | **73.38**±0.67 | 73.71±0.57 | 73.93±1.60 | **73.08**±0.66 | 71.52±0.50 | 71.72±1.51 |
| K=5 | 72.89±1.21 | 73.72±0.93 | 74.32±1.17 | 72.74±1.15 | 72.93±0.71 | 72.12±1.02 |
| K=3 | 71.77±0.63 | **74.35**±0.92 | **74.55**±1.30 | 71.82±0.66 | **73.25**±0.60 | **73.98**±1.24 |
| K=1 | 63.47±1.11 | 71.10±0.78 | 72.33±0.90 | 62.85±1.42 | 68.80±0.75 | 70.02±0.79 |

In this section, we present remarks on how to better integrate TOBE with multi-step message-passing GNNs, particularly APPNP and GPRGNN. We observe that directly combining ToBE with APPNP/GPRGNN may lead to over-smoothing issues. The underlying reason is that ToBE introduces dynamic virtual nodes and edges to the graph and thus increase its connectivity. Additionally, both APPNP and GPRGNN involve multi-step message-passing based on Personalized PageRank (PPR), which can potentially exacerbate over-smoothing on graphs with enhanced connectivity. Therefore, we should use a smaller number of message-passing steps when incorporating with TOBE.

As described in Appendix C.2, following the original papers' settings, we set the number of propagation steps $K$ for APPNP/GPRGNN to 10. However, when combined with ToBE, we should reduce $K$ to alleviate over-smoothing. We tested $K = 10/5/3/1$ with both APPNP and GPRGNN (results are shown in Table 9) and observe that smaller $K$ helps TOBE to achieve better performance. Generally, the vanilla APPNP and GPRGNN achieves the best performance at $k = 10$. When combining with ToBE, the best performance is usually achieved at $K = 3$. Thus we suggest to set $k$ to around 3 when incorporating TOBE to boost multi-step message-passing GNNs.

## D.6 FURTHER COMPARISON WITH OTHER BASELINES

Given that our experiments require a comprehensive combination of existing CIGL techniques and GNN backbones to test ToBE, we chose model-independent and efficient CIGL methods for a comprehensive evaluation in the main results. In this section, we compare TOBE with other representative CIGL techniques that are either model-dependent or with design constraints making them

Table 10: Comparison between TOBE and additional representative CIGL baselines.

| Metric | BAcc. | | | Macro-F1 | | | PerfStd | | |
|---|---|---|---|---|---|---|---|---|---|
| Dataset | Cora | CiteSeer | PubMed | Cora | CiteSeer | PubMed | Cora | CiteSeer | PubMed |
| GCN | 0.619 | 0.376 | 0.642 | 0.607 | 0.281 | 0.551 | 0.275 | 0.299 | 0.347 |
| BNE | 0.651 | 0.515 | 0.654 | 0.674 | 0.497 | 0.613 | 0.204 | 0.155 | 0.245 |
| GCN+TAM | 0.620 | 0.381 | 0.671 | 0.610 | 0.290 | 0.607 | 0.296 | 0.324 | 0.308 |
| GCN+GraphMixup | 0.641 | 0.508 | OOM | 0.627 | 0.473 | OOM | 0.245 | 0.252 | OOM |
| GCN+TOBE$_0$ | 0.655 | 0.527 | **0.686** | 0.633 | 0.516 | **0.672** | 0.213 | **0.139** | **0.092** |
| GCN+TOBE$_1$ | **0.698** | **0.554** | 0.676 | **0.687** | **0.550** | 0.644 | **0.185** | **0.139** | 0.218 |

incompatible with TOBE. Specifically, we include BNE (Zhu et al., 2023), GraphMixup (Wu et al., 2022), and TAM (Song et al., 2022a). BNE preprocesses graph data to derive independent features and employs an MLP for prediction, making it incompatible with TOBE and existing GNNs. Graph-Mixup and TAM rely on invariant graph topology. Further, the code implementation for Graph-Mixup does not support the addition of virtual nodes to the original graph.

To ensure a fair comparison, we implemented these methods in a unified experimental framework, employing the same data splitting strategy as in our paper and reporting the averages over five independent runs (i.e., ensuring comparability with the results reported in our paper). The results for Imbalance Ratio=10/20 are reported in Table 10. We can observe that TOBE continues to demonstrate a notable advantage compared to these methods. We also note that GraphMixup encounters out-of-memory (OOM) issues due to the requirement of computing dense adjacency matrix on the PubMed dataset with a Tesla V100 32GB GPU.

### D.7 CHOOSING BETWEEN TOBE$_0$ AND TOBE$_1$

In this section, we summarize the strengths and limitations of TOBE$_0$ and TOBE$_1$, and give suggestions for choosing between them in practice. In short, we recommend using TOBE$_1$ to achieve better classification performance. But in case that computational resources are limited, TOBE$_0$ can serve as a more efficient alternative. The reasons are as follows:

**Performance.** We observe a noticeable performance gap between TOBE$_0$ and TOBE$_1$, wherein TOBE$_1$ consistently demonstrates superior classification performance due to its incorporation of local topological structure. Across the 15 scenarios outlined in Table 1 (the best scores for 3 datasets x 5 GNN backbones): (1) TOBE$_1$ outperforms TOBE$_0$ significantly in 12 out of 15 scenarios for BAcc/F1 scores, with an average F1 advantage of 1.692 in the 11 leading cases. (2) Conversely, TOBE$_0$ exhibits a less pronounced advantage in the instances where it outperforms TOBE$_1$, with an average advantage of 0.518 in the 4 leading scenarios[7].

**Efficiency.** On the other hand, it's worth noting that TOBE$_1$ generally incurs higher time and space complexity compared to TOBE$_0$. Specifically, TOBE$_0$ demonstrates linear complexity concerning the number of nodes, whereas TOBE$_1$ exhibits linear growth in complexity with the number of edges. Given that real-world graph data often features a significantly larger number of edges than nodes, TOBE$_0$ is usually the more efficient option (especially for densely connected graphs).

### D.8 LIMITATIONS AND FUTURE WORKS

One potential limitation of the proposed TOBE framework is its reliance on exploiting model prediction for risk and likelihood estimation. This strategy may not provide accurate estimation when the model itself exhibits extremely poor predictive performance. However, this rarely occurs in practice and can be prevented by more careful fine-tuning of parameters and model architectures. In addition to this, as described in Section 3, we adopt several fast measures to estimate node uncertainty, prediction risk, and posterior likelihood for the sake of efficiency. Other techniques for such purposes (e.g., deterministic (Liu et al., 2020; Zhao et al., 2020)/Bayesian (Zhang et al., 2019; Hasanzadeh et al., 2020)/Jackknife (Kang et al., 2022a) uncertainty estimation) can be easily integrated into the proposed TOBE framework, although the computational efficiency might be a major bottleneck.

---

[7]Despite TOBE$_1$ holding a relative performance edge, both TOBE$_0$ and TOBE$_1$ substantially enhance the performance of the *best-performing* CIGL baseline methods. Over the 15 scenarios, TOBE$_0$ yields an average improvement of 4.789/5.848 in the best BAcc/F1, while TOBE$_1$ brings an average improvement of 5.805/6.950.

Table 11: Balanced accuracy of combining TOBE with 6 IGL baselines × 5 GNN backbones.

| Dataset (IR=10) | | Cora | | | CiteSeer | | | PubMed | |
|---|---|---|---|---|---|---|---|---|---|
| Metric: BAcc.↑ | | Base | + TOBE$_0$ | + TOBE$_1$ | Base | + TOBE$_0$ | + TOBE$_1$ | Base | + TOBE$_0$ | + TOBE$_1$ |
| GCN | Vanilla | $61.56_{\pm1.24}$ | $65.54_{\pm1.25}$ | $69.80_{\pm1.30}$ | $37.62_{\pm1.61}$ | $52.65_{\pm1.08}$ | $55.37_{\pm1.39}$ | $64.23_{\pm1.55}$ | $68.62_{\pm0.77}$ | $67.57_{\pm3.22}$ |
| | Reweight | $67.65_{\pm0.64}$ | $70.97_{\pm1.28}$ | $72.14_{\pm0.72}$ | $42.49_{\pm2.66}$ | $57.91_{\pm0.98}$ | $58.36_{\pm1.09}$ | $71.20_{\pm2.33}$ | $74.19_{\pm1.12}$ | $73.37_{\pm0.96}$ |
| | ReNode | $66.60_{\pm1.33}$ | $71.37_{\pm0.62}$ | $71.84_{\pm1.25}$ | $42.57_{\pm1.05}$ | $57.47_{\pm0.62}$ | $59.28_{\pm0.59}$ | $71.52_{\pm2.16}$ | $73.20_{\pm0.71}$ | $72.53_{\pm1.62}$ |
| | Resample | $59.48_{\pm1.53}$ | $72.51_{\pm0.68}$ | $74.24_{\pm0.91}$ | $39.15_{\pm2.05}$ | $57.90_{\pm0.33}$ | $58.78_{\pm1.44}$ | $64.97_{\pm1.94}$ | $72.53_{\pm0.85}$ | $72.87_{\pm1.16}$ |
| | SMOTE | $58.27_{\pm1.05}$ | $72.16_{\pm0.53}$ | $73.89_{\pm1.06}$ | $39.27_{\pm1.90}$ | $60.06_{\pm0.81}$ | $61.97_{\pm1.19}$ | $64.41_{\pm1.95}$ | $73.17_{\pm0.84}$ | $73.13_{\pm0.77}$ |
| | GSMOTE | $67.99_{\pm1.37}$ | $68.52_{\pm0.81}$ | $71.55_{\pm0.50}$ | $45.05_{\pm1.95}$ | $57.68_{\pm1.03}$ | $57.65_{\pm1.18}$ | $73.99_{\pm0.88}$ | $73.09_{\pm1.30}$ | $76.57_{\pm0.42}$ |
| | GENS | $70.12_{\pm0.43}$ | $72.22_{\pm0.57}$ | $72.58_{\pm0.58}$ | $56.01_{\pm1.17}$ | $60.60_{\pm0.63}$ | $62.67_{\pm0.42}$ | $73.66_{\pm1.18}$ | $76.11_{\pm0.60}$ | $76.91_{\pm1.03}$ |
| | Best | 70.12 | 72.51 | **74.24** | 56.01 | 60.60 | 62.67 | 73.99 | 76.11 | 76.91 |
| GAT | Vanilla | $61.53_{\pm1.13}$ | $66.27_{\pm0.83}$ | $70.13_{\pm1.07}$ | $39.25_{\pm1.84}$ | $55.66_{\pm1.23}$ | $60.34_{\pm1.66}$ | $65.46_{\pm0.69}$ | $73.19_{\pm0.86}$ | $74.75_{\pm1.18}$ |
| | Reweight | $66.94_{\pm1.24}$ | $71.80_{\pm0.48}$ | $71.61_{\pm0.85}$ | $41.29_{\pm3.39}$ | $59.33_{\pm0.51}$ | $61.23_{\pm0.99}$ | $68.37_{\pm1.41}$ | $75.30_{\pm1.07}$ | $74.52_{\pm1.14}$ |
| | ReNode | $66.81_{\pm0.98}$ | $72.14_{\pm1.24}$ | $70.31_{\pm1.38}$ | $43.25_{\pm1.78}$ | $58.26_{\pm1.98}$ | $56.49_{\pm1.73}$ | $71.18_{\pm2.13}$ | $75.55_{\pm1.01}$ | $75.22_{\pm0.84}$ |
| | Resample | $57.76_{\pm1.73}$ | $71.90_{\pm0.88}$ | $73.29_{\pm1.08}$ | $35.97_{\pm1.42}$ | $60.10_{\pm1.26}$ | $60.33_{\pm0.75}$ | $65.14_{\pm0.86}$ | $73.27_{\pm0.61}$ | $73.89_{\pm0.40}$ |
| | SMOTE | $58.81_{\pm0.64}$ | $70.50_{\pm0.44}$ | $72.19_{\pm0.75}$ | $36.95_{\pm1.86}$ | $60.59_{\pm1.19}$ | $62.36_{\pm1.18}$ | $64.81_{\pm1.47}$ | $73.90_{\pm0.68}$ | $74.08_{\pm0.51}$ |
| | GSMOTE | $64.68_{\pm1.02}$ | $69.29_{\pm1.82}$ | $71.14_{\pm0.96}$ | $41.82_{\pm1.14}$ | $56.11_{\pm1.23}$ | $57.71_{\pm2.58}$ | $68.72_{\pm1.69}$ | $74.65_{\pm0.65}$ | $74.41_{\pm1.57}$ |
| | GENS | $69.76_{\pm0.45}$ | $70.63_{\pm0.40}$ | $71.02_{\pm1.22}$ | $51.50_{\pm2.21}$ | $60.95_{\pm1.51}$ | $63.49_{\pm0.75}$ | $73.13_{\pm1.18}$ | $74.34_{\pm0.35}$ | $75.65_{\pm0.82}$ |
| | Best | 69.76 | 72.14 | **73.29** | 51.50 | 60.95 | 63.49 | 73.13 | 75.55 | 75.65 |
| SAGE | Vanilla | $59.17_{\pm1.23}$ | $66.24_{\pm0.92}$ | $66.53_{\pm0.80}$ | $42.96_{\pm0.28}$ | $54.99_{\pm2.51}$ | $53.18_{\pm2.90}$ | $67.56_{\pm0.84}$ | $75.31_{\pm0.93}$ | $77.38_{\pm0.68}$ |
| | Reweight | $63.76_{\pm0.89}$ | $70.15_{\pm1.15}$ | $71.14_{\pm0.84}$ | $45.91_{\pm2.05}$ | $57.95_{\pm0.73}$ | $55.90_{\pm0.93}$ | $68.03_{\pm1.69}$ | $74.56_{\pm0.41}$ | $75.39_{\pm0.38}$ |
| | ReNode | $65.32_{\pm1.07}$ | $71.31_{\pm1.29}$ | $71.54_{\pm0.85}$ | $48.55_{\pm2.31}$ | $56.32_{\pm0.40}$ | $56.49_{\pm1.73}$ | $69.08_{\pm2.04}$ | $74.24_{\pm0.20}$ | $75.28_{\pm0.69}$ |
| | Resample | $57.77_{\pm1.35}$ | $71.24_{\pm1.08}$ | $73.01_{\pm1.02}$ | $39.37_{\pm1.40}$ | $61.41_{\pm1.11}$ | $61.93_{\pm1.40}$ | $69.22_{\pm1.28}$ | $74.91_{\pm1.19}$ | $75.80_{\pm0.39}$ |
| | SMOTE | $58.81_{\pm1.97}$ | $70.31_{\pm1.35}$ | $73.02_{\pm2.29}$ | $38.42_{\pm1.69}$ | $64.14_{\pm0.75}$ | $66.35_{\pm0.70}$ | $64.96_{\pm1.56}$ | $74.59_{\pm0.96}$ | $77.31_{\pm0.45}$ |
| | GSMOTE | $61.57_{\pm1.78}$ | $69.88_{\pm0.96}$ | $72.28_{\pm1.48}$ | $42.21_{\pm2.12}$ | $60.91_{\pm1.33}$ | $62.32_{\pm1.06}$ | $71.55_{\pm0.64}$ | $74.74_{\pm0.81}$ | $76.14_{\pm0.21}$ |
| | GENS | $68.84_{\pm0.41}$ | $69.78_{\pm1.18}$ | $71.92_{\pm0.71}$ | $52.57_{\pm1.78}$ | $64.36_{\pm0.68}$ | $63.84_{\pm0.68}$ | $71.38_{\pm0.99}$ | $75.89_{\pm1.17}$ | $76.46_{\pm1.29}$ |
| | Best | 68.84 | 71.31 | **73.02** | 52.57 | 64.36 | 66.35 | 71.55 | 75.89 | 77.38 |
| APPNP | Vanilla | $55.37_{\pm1.65}$ | $58.13_{\pm1.69}$ | $61.71_{\pm1.66}$ | $35.69_{\pm0.14}$ | $35.68_{\pm0.15}$ | $36.02_{\pm0.25}$ | $59.30_{\pm0.50}$ | $55.62_{\pm0.31}$ | $57.82_{\pm0.29}$ |
| | Reweight | $72.62_{\pm0.47}$ | $73.62_{\pm0.89}$ | $72.51_{\pm0.87}$ | $50.88_{\pm3.64}$ | $63.54_{\pm1.02}$ | $65.57_{\pm1.11}$ | $72.00_{\pm0.81}$ | $72.15_{\pm0.60}$ | $71.22_{\pm1.10}$ |
| | ReNode | $73.74_{\pm1.12}$ | $75.02_{\pm1.54}$ | $72.15_{\pm0.76}$ | $50.50_{\pm3.51}$ | $63.73_{\pm0.54}$ | $65.13_{\pm0.40}$ | $72.76_{\pm1.37}$ | $71.54_{\pm0.96}$ | $71.88_{\pm0.70}$ |
| | Resample | $65.78_{\pm1.72}$ | $73.14_{\pm0.94}$ | $73.57_{\pm0.92}$ | $40.79_{\pm1.87}$ | $66.54_{\pm0.49}$ | $59.51_{\pm4.16}$ | $67.74_{\pm1.94}$ | $72.25_{\pm0.81}$ | $74.41_{\pm0.95}$ |
| | SMOTE | $65.34_{\pm1.68}$ | $73.18_{\pm1.02}$ | $72.88_{\pm0.90}$ | $40.79_{\pm2.05}$ | $66.62_{\pm0.33}$ | $58.82_{\pm4.59}$ | $67.24_{\pm2.10}$ | $72.67_{\pm1.65}$ | $73.33_{\pm1.37}$ |
| | GSMOTE | $71.13_{\pm0.72}$ | $73.37_{\pm0.82}$ | $73.78_{\pm0.71}$ | $45.37_{\pm2.75}$ | $64.95_{\pm0.11}$ | $62.95_{\pm2.58}$ | $69.57_{\pm2.20}$ | $73.37_{\pm0.95}$ | $74.90_{\pm1.27}$ |
| | Best | 73.74 | **75.02** | 73.78 | 50.88 | 66.62 | 65.57 | 72.76 | 73.37 | 74.90 |
| GPRGNN | Vanilla | $67.97_{\pm0.51}$ | $71.99_{\pm1.14}$ | $73.38_{\pm1.18}$ | $42.31_{\pm2.16}$ | $55.85_{\pm0.89}$ | $58.82_{\pm1.91}$ | $67.04_{\pm1.82}$ | $57.92_{\pm0.45}$ | $77.49_{\pm1.15}$ |
| | Reweight | $72.15_{\pm0.57}$ | $72.90_{\pm1.33}$ | $73.22_{\pm0.55}$ | $53.22_{\pm2.89}$ | $59.78_{\pm0.76}$ | $61.00_{\pm1.82}$ | $73.35_{\pm1.07}$ | $75.22_{\pm1.02}$ | $76.86_{\pm0.76}$ |
| | ReNode | $73.38_{\pm0.67}$ | $73.71_{\pm0.57}$ | $73.93_{\pm1.60}$ | $54.66_{\pm2.82}$ | $59.69_{\pm0.73}$ | $60.34_{\pm1.31}$ | $73.56_{\pm0.98}$ | $75.69_{\pm1.10}$ | $76.25_{\pm0.67}$ |
| | Resample | $67.00_{\pm1.33}$ | $72.94_{\pm1.02}$ | $74.89_{\pm0.86}$ | $42.27_{\pm2.15}$ | $64.16_{\pm0.62}$ | $63.89_{\pm0.98}$ | $70.42_{\pm1.51}$ | $73.79_{\pm0.83}$ | $75.31_{\pm0.54}$ |
| | SMOTE | $66.99_{\pm1.33}$ | $74.01_{\pm1.51}$ | $74.41_{\pm1.05}$ | $40.97_{\pm2.02}$ | $63.88_{\pm0.55}$ | $62.60_{\pm1.72}$ | $70.29_{\pm1.47}$ | $73.89_{\pm0.69}$ | $75.48_{\pm1.02}$ |
| | GSMOTE | $70.94_{\pm0.57}$ | $73.63_{\pm1.25}$ | $74.02_{\pm0.90}$ | $48.01_{\pm3.28}$ | $63.03_{\pm0.92}$ | $61.68_{\pm0.86}$ | $71.51_{\pm1.91}$ | $72.16_{\pm0.58}$ | $74.77_{\pm0.83}$ |
| | Best | 73.38 | 74.01 | **74.89** | 54.66 | 64.16 | 63.89 | 73.56 | 75.69 | 77.49 |

How to exploit alternative uncertainty/risk measures while retaining computational efficiency is an interesting future direction.

Further, beyond the class imbalance in the node label distribution, graph data can also exhibit multi-facet skewness in other aspects. For instance, class imbalance may also exist in edge-level (e.g., in edge classification/prediction) and graph-level (e.g., in graph classification). Beyond the quantity imbalance among classes, skewness may also exists in the topological structure, such as degree imbalance (Kang et al., 2022b), and motif-level imbalance (Zhao et al., 2022). How to jointly consider the multi-facet node/edge/graph-level imbalance to benefit more graph learning tasks is an exciting yet challenging future direction.

Finally, some recent studies (Zhou et al., 2022; Du et al., 2021) highlight the possibility of extracting graph structures from i.i.d. tabular data and leveraging them to enhance representation learning. It's worth noting that the extracted graphs may also manifest class imbalance, potentially influencing classification performance. The exploration of TOBE's potential to transcend graph-centric challenges and find applications in non-graph tasks represents an exciting avenue for further investigation.

# E  ADDITIONAL RESULTS

Due to space limitation, we report the key results of our experiments in Table 1 and 2. We now provide complete results for all settings with the standard error of 5 independent runs. Specifically, Table 11 & 12 & 13 complement Table 1, and Table 14 complements Table 2. The results indicate that TOBE can consistently boost various CIGL baselines with all GNN backbones, performance metrics, as well as different types and levels of class imbalance, which aligns with our conclusions in the paper.

Table 12: Macro-F1 score of combining ToBE with 6 IGL baselines × 5 GNN backbones.

| | Dataset (IR=10) | Cora | | | CiteSeer | | | PubMed | | |
|---|---|---|---|---|---|---|---|---|---|---|
| | **Metric: Macro-F1↑** | Base | +ToBE$_0$ | +ToBE$_1$ | Base | +ToBE$_0$ | +ToBE$_1$ | Base | +ToBE$_0$ | +ToBE$_1$ |
| **GCN** | Vanilla | 60.10$_{\pm1.53}$ | 63.28$_{\pm1.07}$ | **68.68$_{\pm1.49}$** | 28.05$_{\pm2.53}$ | 51.55$_{\pm1.28}$ | **54.94$_{\pm1.44}$** | 55.09$_{\pm2.48}$ | **67.16$_{\pm1.53}$** | 64.40$_{\pm3.68}$ |
| | Reweight | 67.85$_{\pm0.62}$ | 69.41$_{\pm1.01}$ | **70.31$_{\pm0.82}$** | 36.59$_{\pm3.66}$ | 56.84$_{\pm1.06}$ | **57.54$_{\pm1.08}$** | 67.07$_{\pm3.42}$ | 72.94$_{\pm0.81}$ | **73.24$_{\pm0.90}$** |
| | ReNode | 66.66$_{\pm1.59}$ | 69.79$_{\pm0.79}$ | **70.59$_{\pm1.25}$** | 34.64$_{\pm1.54}$ | 56.69$_{\pm0.64}$ | **58.07$_{\pm0.77}$** | 67.86$_{\pm3.99}$ | **72.61$_{\pm0.41}$** | 72.25$_{\pm0.89}$ |
| | Resample | 57.34$_{\pm2.27}$ | 71.36$_{\pm0.39}$ | **72.82$_{\pm1.13}$** | 29.73$_{\pm2.77}$ | 57.17$_{\pm0.48}$ | **58.03$_{\pm1.42}$** | 56.74$_{\pm3.54}$ | 71.19$_{\pm0.83}$ | **73.13$_{\pm1.33}$** |
| | SMOTE | 55.65$_{\pm1.62}$ | 71.04$_{\pm0.16}$ | **72.82$_{\pm0.86}$** | 29.39$_{\pm2.81}$ | 59.53$_{\pm0.88}$ | **61.53$_{\pm1.24}$** | 56.14$_{\pm3.74}$ | 71.72$_{\pm0.60}$ | **72.83$_{\pm1.20}$** |
| | GSMOTE | 68.01$_{\pm1.67}$ | 67.60$_{\pm1.00}$ | **70.28$_{\pm0.48}$** | 40.07$_{\pm3.02}$ | **56.64$_{\pm1.09}$** | 56.25$_{\pm1.50}$ | 70.60$_{\pm1.17}$ | 72.95$_{\pm1.39}$ | **75.70$_{\pm0.35}$** |
| | GENS | 69.96$_{\pm0.29}$ | 71.62$_{\pm0.64}$ | **72.28$_{\pm0.65}$** | 54.45$_{\pm1.69}$ | 59.89$_{\pm0.68}$ | **62.46$_{\pm0.43}$** | 71.28$_{\pm1.84}$ | 75.77$_{\pm0.55}$ | **76.86$_{\pm0.93}$** |
| | Best | 69.96 | 71.62 | **72.82** | 54.45 | 59.89 | **62.46** | 71.28 | 75.77 | **76.86** |
| **GAT** | Vanilla | 60.71$_{\pm1.61}$ | 64.27$_{\pm0.95}$ | **68.93$_{\pm0.79}$** | 31.12$_{\pm3.15}$ | 54.71$_{\pm1.18}$ | **59.42$_{\pm1.55}$** | 57.32$_{\pm1.55}$ | 71.27$_{\pm1.11}$ | **74.03$_{\pm1.08}$** |
| | Reweight | 66.49$_{\pm1.34}$ | **69.84$_{\pm0.91}$** | 69.79$_{\pm0.77}$ | 34.94$_{\pm4.09}$ | 58.53$_{\pm0.68}$ | **60.28$_{\pm1.12}$** | 67.38$_{\pm3.22}$ | **75.13$_{\pm1.13}$** | 73.88$_{\pm1.38}$ |
| | ReNode | 67.27$_{\pm1.23}$ | **70.61$_{\pm0.83}$** | 68.24$_{\pm1.48}$ | 37.72$_{\pm2.61}$ | 57.64$_{\pm2.11}$ | **58.57$_{\pm0.75}$** | 67.38$_{\pm3.22}$ | 74.88$_{\pm0.99}$ | **74.96$_{\pm1.18}$** |
| | Resample | 55.36$_{\pm2.47}$ | 70.87$_{\pm0.94}$ | **72.31$_{\pm1.07}$** | 25.71$_{\pm1.97}$ | **59.77$_{\pm1.31}$** | 59.66$_{\pm0.95}$ | 57.24$_{\pm1.54}$ | 72.53$_{\pm0.66}$ | **73.09$_{\pm0.83}$** |
| | SMOTE | 57.49$_{\pm0.60}$ | 69.68$_{\pm0.66}$ | **71.74$_{\pm1.03}$** | 26.05$_{\pm2.30}$ | 59.83$_{\pm1.33}$ | **61.75$_{\pm1.30}$** | 55.66$_{\pm2.76}$ | **73.33$_{\pm1.00}$** | 73.30$_{\pm0.16}$ |
| | GSMOTE | 64.34$_{\pm1.69}$ | 68.23$_{\pm1.80}$ | **69.77$_{\pm1.08}$** | 35.07$_{\pm1.77}$ | 55.86$_{\pm1.10}$ | **56.25$_{\pm2.69}$** | 63.35$_{\pm2.97}$ | **74.23$_{\pm0.84}$** | 73.34$_{\pm2.06}$ |
| | GENS | 69.96$_{\pm0.62}$ | 69.83$_{\pm0.41}$ | **70.71$_{\pm1.16}$** | 48.34$_{\pm2.19}$ | 60.04$_{\pm1.85}$ | **62.55$_{\pm0.86}$** | 71.78$_{\pm1.19}$ | 72.69$_{\pm0.84}$ | **74.42$_{\pm1.12}$** |
| | Best | 69.96 | 70.87 | **72.31** | 48.34 | 60.04 | **62.55** | 71.78 | **75.13** | 74.96 |
| **SAGE** | Vanilla | 57.36$_{\pm1.77}$ | 64.90$_{\pm0.87}$ | **65.61$_{\pm0.97}$** | 36.07$_{\pm1.06}$ | **54.76$_{\pm2.47}$** | 51.86$_{\pm3.25}$ | 63.75$_{\pm1.24}$ | 74.35$_{\pm0.74}$ | **76.92$_{\pm0.63}$** |
| | Reweight | 63.72$_{\pm1.10}$ | 69.06$_{\pm0.90}$ | **69.59$_{\pm0.53}$** | 39.64$_{\pm2.57}$ | **57.17$_{\pm0.76}$** | 54.83$_{\pm0.74}$ | 62.83$_{\pm2.57}$ | 73.88$_{\pm0.40}$ | **75.42$_{\pm0.48}$** |
| | ReNode | 65.59$_{\pm1.44}$ | **69.99$_{\pm1.35}$** | 69.86$_{\pm1.27}$ | 44.20$_{\pm3.68}$ | 55.41$_{\pm0.48}$ | **55.78$_{\pm1.63}$** | 64.97$_{\pm3.00}$ | 74.33$_{\pm0.20}$ | **74.88$_{\pm0.53}$** |
| | Resample | 55.29$_{\pm2.12}$ | 70.40$_{\pm1.11}$ | **71.49$_{\pm0.79}$** | 30.14$_{\pm2.20}$ | 60.71$_{\pm1.25}$ | **61.29$_{\pm1.48}$** | 65.23$_{\pm2.26}$ | 74.28$_{\pm0.96}$ | **75.48$_{\pm0.44}$** |
| | SMOTE | 56.72$_{\pm2.69}$ | 69.42$_{\pm1.29}$ | **71.71$_{\pm1.94}$** | 29.22$_{\pm2.33}$ | 63.61$_{\pm0.87}$ | **65.91$_{\pm0.68}$** | 57.60$_{\pm3.22}$ | 72.98$_{\pm0.69}$ | **76.45$_{\pm0.77}$** |
| | GSMOTE | 59.44$_{\pm2.25}$ | 69.10$_{\pm0.95}$ | **71.30$_{\pm1.47}$** | 34.86$_{\pm3.46}$ | **60.53$_{\pm1.27}$** | 55.68$_{\pm0.31}$ | 64.36$_{\pm1.02}$ | 74.36$_{\pm1.02}$ | **75.68$_{\pm0.31}$** |
| | GENS | 68.23$_{\pm0.72}$ | 69.76$_{\pm0.95}$ | **71.11$_{\pm0.81}$** | 51.05$_{\pm2.03}$ | 63.87$_{\pm0.82}$ | 63.41$_{\pm0.57}$ | 70.06$_{\pm0.86}$ | 75.33$_{\pm1.46}$ | **76.01$_{\pm1.14}$** |
| | Best | 68.23 | 70.40 | **71.71** | 51.05 | 63.87 | **65.91** | 70.06 | 75.33 | **76.92** |
| **APPNP** | Vanilla | 50.39$_{\pm2.81}$ | 54.19$_{\pm2.58}$ | **59.99$_{\pm2.49}$** | 22.21$_{\pm0.13}$ | 22.54$_{\pm0.25}$ | **22.89$_{\pm0.22}$** | 44.50$_{\pm0.21}$ | **44.67$_{\pm0.07}$** | 44.59$_{\pm0.06}$ |
| | Reweight | 72.63$_{\pm0.53}$ | **72.71$_{\pm0.60}$** | 70.61$_{\pm0.65}$ | 45.25$_{\pm4.85}$ | 63.08$_{\pm1.03}$ | **65.20$_{\pm1.20}$** | 69.53$_{\pm1.14}$ | 72.24$_{\pm0.58}$ | **72.26$_{\pm0.80}$** |
| | ReNode | 73.67$_{\pm0.98}$ | **73.67$_{\pm1.18}$** | 69.79$_{\pm0.72}$ | 44.91$_{\pm4.99}$ | 62.97$_{\pm0.78}$ | **64.47$_{\pm0.40}$** | 70.65$_{\pm1.66}$ | **72.33$_{\pm0.90}$** | 72.18$_{\pm0.55}$ |
| | Resample | 65.20$_{\pm2.08}$ | 72.25$_{\pm0.82}$ | **72.72$_{\pm0.97}$** | 31.04$_{\pm2.76}$ | **66.06$_{\pm0.54}$** | 54.57$_{\pm6.08}$ | 62.42$_{\pm3.62}$ | 72.32$_{\pm0.93}$ | **74.27$_{\pm1.08}$** |
| | SMOTE | 64.70$_{\pm2.06}$ | **72.90$_{\pm0.83}$** | 72.31$_{\pm0.94}$ | 30.90$_{\pm2.86}$ | **66.18$_{\pm0.37}$** | 53.90$_{\pm6.26}$ | 61.83$_{\pm3.65}$ | 72.55$_{\pm1.61}$ | **73.87$_{\pm1.37}$** |
| | GSMOTE | 71.20$_{\pm0.67}$ | 73.02$_{\pm0.74}$ | **73.22$_{\pm0.92}$** | 37.90$_{\pm4.29}$ | **64.56$_{\pm0.18}$** | 60.41$_{\pm3.84}$ | 65.65$_{\pm3.06}$ | 72.54$_{\pm0.85}$ | **74.61$_{\pm1.36}$** |
| | Best | 73.67 | **73.67** | 73.22 | 45.25 | **66.18** | 65.20 | 70.65 | 72.55 | **74.61** |
| **GPRGNN** | Vanilla | 67.86$_{\pm0.79}$ | 70.80$_{\pm1.16}$ | **72.32$_{\pm1.18}$** | 35.00$_{\pm2.96}$ | 55.06$_{\pm0.89}$ | **56.31$_{\pm2.87}$** | 59.01$_{\pm3.62}$ | 50.12$_{\pm1.46}$ | **77.62$_{\pm1.04}$** |
| | Reweight | **71.66$_{\pm0.85}$** | 70.46$_{\pm0.98}$ | 71.24$_{\pm0.49}$ | 49.19$_{\pm3.61}$ | 59.11$_{\pm0.73}$ | **60.30$_{\pm2.04}$** | 71.18$_{\pm0.95}$ | 75.47$_{\pm0.90}$ | **77.01$_{\pm0.52}$** |
| | ReNode | **73.08$_{\pm0.66}$** | 71.52$_{\pm0.50}$ | 71.72$_{\pm1.51}$ | 50.34$_{\pm3.18}$ | **59.10$_{\pm0.75}$** | 58.94$_{\pm1.36}$ | 71.45$_{\pm1.19}$ | 75.08$_{\pm1.06}$ | **75.76$_{\pm0.84}$** |
| | Resample | 66.42$_{\pm1.65}$ | 71.70$_{\pm0.86}$ | **73.54$_{\pm0.83}$** | 32.60$_{\pm2.71}$ | **63.59$_{\pm0.65}$** | 63.12$_{\pm1.06}$ | 66.58$_{\pm2.08}$ | 73.66$_{\pm0.86}$ | **75.42$_{\pm0.35}$** |
| | SMOTE | 66.43$_{\pm1.74}$ | 72.89$_{\pm1.23}$ | **73.47$_{\pm1.12}$** | 31.38$_{\pm2.70}$ | **63.41$_{\pm0.55}$** | 61.23$_{\pm2.59}$ | 66.78$_{\pm1.97}$ | 73.98$_{\pm0.70}$ | **75.63$_{\pm0.91}$** |
| | GSMOTE | 70.87$_{\pm0.53}$ | 72.53$_{\pm0.85}$ | **73.12$_{\pm0.95}$** | 42.82$_{\pm4.52}$ | **62.09$_{\pm1.04}$** | 60.82$_{\pm0.88}$ | 67.93$_{\pm3.01}$ | 72.72$_{\pm0.72}$ | **74.66$_{\pm0.74}$** |
| | Best | 73.08 | 72.89 | **73.54** | 50.34 | **63.59** | 63.12 | 71.45 | 75.47 | **77.62** |

Table 13: Performance deviation of combining TOBE with 6 IGL baselines × 5 GNN backbones.

| Dataset (IR=10) | | Cora | | | CiteSeer | | | PubMed | |
|---|---|---|---|---|---|---|---|---|---|
| Metric: PerfStd↓ | | Base | + TOBE$_0$ | + TOBE$_1$ | Base | + TOBE$_0$ | + TOBE$_1$ | Base | + TOBE$_0$ | + TOBE$_1$ |
| GCN | Vanilla | $27.88_{\pm1.79}$ | $21.27_{\pm1.76}$ | $18.49_{\pm2.68}$ | $29.93_{\pm1.38}$ | $13.82_{\pm2.06}$ | $13.93_{\pm0.80}$ | $34.73_{\pm2.14}$ | $9.23_{\pm2.78}$ | $21.81_{\pm4.51}$ |
| | Reweight | $22.29_{\pm1.41}$ | $14.43_{\pm2.51}$ | $18.32_{\pm2.20}$ | $25.47_{\pm1.78}$ | $19.10_{\pm1.48}$ | $22.64_{\pm0.79}$ | $19.33_{\pm5.26}$ | $10.21_{\pm1.58}$ | $5.88_{\pm0.83}$ |
| | ReNode | $22.88_{\pm1.64}$ | $14.65_{\pm2.07}$ | $17.00_{\pm2.13}$ | $30.31_{\pm1.51}$ | $20.22_{\pm0.88}$ | $22.99_{\pm1.09}$ | $18.14_{\pm5.79}$ | $12.99_{\pm1.57}$ | $10.96_{\pm1.92}$ |
| | Resample | $31.57_{\pm1.85}$ | $15.13_{\pm2.14}$ | $15.25_{\pm2.79}$ | $31.00_{\pm1.32}$ | $16.30_{\pm1.89}$ | $20.79_{\pm0.43}$ | $30.90_{\pm5.67}$ | $11.63_{\pm3.20}$ | $7.82_{\pm0.80}$ |
| | SMOTE | $33.32_{\pm1.38}$ | $16.33_{\pm1.12}$ | $17.95_{\pm2.50}$ | $32.61_{\pm1.45}$ | $17.27_{\pm0.86}$ | $18.25_{\pm0.89}$ | $31.79_{\pm5.21}$ | $10.56_{\pm1.82}$ | $11.66_{\pm2.58}$ |
| | GSMOTE | $21.78_{\pm1.79}$ | $17.90_{\pm2.75}$ | $18.44_{\pm2.20}$ | $22.64_{\pm2.69}$ | $21.37_{\pm1.25}$ | $21.01_{\pm1.45}$ | $15.87_{\pm2.34}$ | $3.35_{\pm1.08}$ | $5.83_{\pm1.27}$ |
| | GENS | $20.04_{\pm1.12}$ | $16.98_{\pm3.02}$ | $18.02_{\pm2.23}$ | $16.95_{\pm2.64}$ | $14.94_{\pm0.75}$ | $15.54_{\pm0.60}$ | $11.93_{\pm3.46}$ | $5.95_{\pm1.85}$ | $5.15_{\pm0.80}$ |
| | Best | 20.04 | 14.43 | 15.25 | 16.95 | 13.82 | 13.93 | 11.93 | 3.35 | 5.15 |
| GAT | Vanilla | $27.38_{\pm1.71}$ | $19.23_{\pm0.80}$ | $17.97_{\pm2.65}$ | $28.32_{\pm2.07}$ | $15.62_{\pm0.77}$ | $15.90_{\pm0.95}$ | $30.94_{\pm1.27}$ | $10.77_{\pm2.04}$ | $8.51_{\pm2.48}$ |
| | Reweight | $22.90_{\pm1.67}$ | $16.44_{\pm2.71}$ | $17.32_{\pm2.83}$ | $30.27_{\pm1.26}$ | $18.64_{\pm1.30}$ | $20.83_{\pm1.06}$ | $24.92_{\pm1.58}$ | $3.01_{\pm0.96}$ | $5.44_{\pm1.33}$ |
| | ReNode | $23.13_{\pm1.54}$ | $15.05_{\pm1.59}$ | $18.96_{\pm1.65}$ | $25.21_{\pm1.85}$ | $20.48_{\pm0.82}$ | $20.51_{\pm0.49}$ | $18.15_{\pm4.37}$ | $4.77_{\pm1.22}$ | $6.17_{\pm0.42}$ |
| | Resample | $32.73_{\pm2.12}$ | $17.87_{\pm2.04}$ | $17.64_{\pm2.57}$ | $32.59_{\pm0.89}$ | $17.76_{\pm1.79}$ | $18.73_{\pm1.02}$ | $31.67_{\pm0.98}$ | $6.18_{\pm1.35}$ | $4.58_{\pm1.14}$ |
| | SMOTE | $31.17_{\pm0.69}$ | $18.40_{\pm1.03}$ | $18.26_{\pm1.87}$ | $33.32_{\pm0.88}$ | $10.68_{\pm0.68}$ | $13.24_{\pm1.12}$ | $32.79_{\pm2.13}$ | $8.14_{\pm2.00}$ | $7.56_{\pm1.05}$ |
| | GSMOTE | $24.84_{\pm1.60}$ | $15.48_{\pm2.08}$ | $18.23_{\pm2.00}$ | $26.74_{\pm1.53}$ | $18.34_{\pm1.54}$ | $19.76_{\pm0.42}$ | $24.50_{\pm2.74}$ | $5.12_{\pm1.38}$ | $8.54_{\pm2.03}$ |
| | GENS | $20.08_{\pm1.56}$ | $17.75_{\pm2.40}$ | $17.88_{\pm2.50}$ | $26.49_{\pm1.18}$ | $12.89_{\pm0.77}$ | $15.09_{\pm0.95}$ | $10.29_{\pm2.75}$ | $7.83_{\pm2.27}$ | $7.55_{\pm2.38}$ |
| | Best | 20.08 | 15.05 | 17.32 | 25.21 | 10.68 | 13.24 | 10.29 | 3.01 | 4.58 |
| SAGE | Vanilla | $29.94_{\pm1.75}$ | $18.62_{\pm2.13}$ | $19.49_{\pm1.67}$ | $26.75_{\pm1.58}$ | $14.56_{\pm1.16}$ | $18.13_{\pm1.32}$ | $21.09_{\pm3.43}$ | $10.96_{\pm1.99}$ | $4.09_{\pm1.17}$ |
| | Reweight | $25.61_{\pm1.60}$ | $15.24_{\pm2.66}$ | $17.54_{\pm2.45}$ | $29.95_{\pm1.83}$ | $19.05_{\pm1.60}$ | $22.94_{\pm0.49}$ | $25.47_{\pm3.49}$ | $3.35_{\pm0.72}$ | $8.09_{\pm0.19}$ |
| | ReNode | $24.12_{\pm1.73}$ | $13.32_{\pm3.03}$ | $15.45_{\pm2.41}$ | $22.41_{\pm4.31}$ | $22.20_{\pm0.97}$ | $22.75_{\pm0.87}$ | $22.92_{\pm4.36}$ | $7.63_{\pm1.23}$ | $5.77_{\pm1.55}$ |
| | Resample | $31.66_{\pm1.47}$ | $15.77_{\pm2.75}$ | $15.08_{\pm2.74}$ | $30.29_{\pm1.16}$ | $18.72_{\pm0.90}$ | $18.48_{\pm2.00}$ | $21.41_{\pm2.88}$ | $4.68_{\pm1.42}$ | $4.76_{\pm1.09}$ |
| | SMOTE | $30.86_{\pm2.64}$ | $17.30_{\pm2.09}$ | $14.87_{\pm3.23}$ | $32.07_{\pm1.00}$ | $13.17_{\pm1.33}$ | $12.78_{\pm0.43}$ | $31.62_{\pm2.86}$ | $13.88_{\pm1.44}$ | $11.63_{\pm2.28}$ |
| | GSMOTE | $27.71_{\pm1.86}$ | $17.28_{\pm2.25}$ | $16.10_{\pm2.94}$ | $28.77_{\pm2.61}$ | $18.69_{\pm0.76}$ | $18.05_{\pm1.21}$ | $20.10_{\pm0.90}$ | $5.37_{\pm1.21}$ | $4.64_{\pm1.61}$ |
| | GENS | $19.81_{\pm1.65}$ | $17.50_{\pm2.05}$ | $17.63_{\pm2.11}$ | $19.76_{\pm2.07}$ | $15.99_{\pm0.81}$ | $16.99_{\pm0.85}$ | $11.76_{\pm2.91}$ | $7.63_{\pm1.51}$ | $8.31_{\pm1.64}$ |
| | Best | 19.81 | 13.32 | 14.87 | 19.76 | 13.17 | 12.78 | 11.76 | 3.35 | 4.09 |
| APPNP | Vanilla | $38.32_{\pm1.94}$ | $35.50_{\pm2.10}$ | $32.18_{\pm1.68}$ | $36.82_{\pm0.10}$ | $36.67_{\pm0.25}$ | $36.83_{\pm0.36}$ | $42.13_{\pm0.27}$ | $40.45_{\pm0.11}$ | $41.34_{\pm0.16}$ |
| | Reweight | $19.83_{\pm1.46}$ | $17.33_{\pm2.88}$ | $18.46_{\pm2.42}$ | $26.19_{\pm2.93}$ | $20.96_{\pm0.58}$ | $20.83_{\pm1.06}$ | $16.96_{\pm2.84}$ | $8.04_{\pm1.94}$ | $9.13_{\pm1.05}$ |
| | ReNode | $18.09_{\pm2.52}$ | $16.87_{\pm2.95}$ | $19.47_{\pm2.00}$ | $25.95_{\pm3.66}$ | $22.09_{\pm1.43}$ | $20.42_{\pm1.57}$ | $14.49_{\pm3.81}$ | $10.25_{\pm1.93}$ | $3.95_{\pm1.02}$ |
| | Resample | $27.28_{\pm2.13}$ | $18.37_{\pm2.34}$ | $18.72_{\pm2.32}$ | $32.71_{\pm1.23}$ | $15.87_{\pm1.02}$ | $23.72_{\pm4.14}$ | $25.86_{\pm4.38}$ | $13.60_{\pm1.68}$ | $9.49_{\pm1.65}$ |
| | SMOTE | $27.86_{\pm1.78}$ | $18.61_{\pm2.35}$ | $19.42_{\pm1.81}$ | $33.26_{\pm1.10}$ | $14.91_{\pm0.93}$ | $22.90_{\pm4.49}$ | $26.37_{\pm4.47}$ | $13.37_{\pm1.97}$ | $8.70_{\pm2.29}$ |
| | GSMOTE | $20.98_{\pm1.45}$ | $18.19_{\pm2.59}$ | $18.55_{\pm2.24}$ | $29.39_{\pm2.20}$ | $16.49_{\pm1.12}$ | $19.76_{\pm3.44}$ | $22.32_{\pm4.21}$ | $11.53_{\pm3.00}$ | $10.69_{\pm2.27}$ |
| | Best | 18.09 | 16.87 | 18.46 | 25.95 | 14.91 | 19.19 | 14.49 | 8.04 | 3.95 |
| GPRGNN | Vanilla | $22.96_{\pm1.20}$ | $18.12_{\pm2.29}$ | $17.00_{\pm2.98}$ | $27.57_{\pm1.32}$ | $17.10_{\pm1.17}$ | $20.94_{\pm2.58}$ | $29.94_{\pm3.68}$ | $36.57_{\pm1.46}$ | $5.30_{\pm0.91}$ |
| | Reweight | $20.94_{\pm1.21}$ | $17.83_{\pm2.82}$ | $19.67_{\pm1.81}$ | $22.43_{\pm2.39}$ | $21.52_{\pm1.06}$ | $20.03_{\pm1.81}$ | $16.12_{\pm1.84}$ | $7.54_{\pm0.49}$ | $5.48_{\pm1.49}$ |
| | ReNode | $18.84_{\pm2.19}$ | $16.78_{\pm2.53}$ | $17.89_{\pm2.96}$ | $24.14_{\pm1.47}$ | $19.84_{\pm1.79}$ | $22.83_{\pm1.41}$ | $14.40_{\pm3.13}$ | $9.75_{\pm2.20}$ | $6.61_{\pm1.47}$ |
| | Resample | $25.62_{\pm1.80}$ | $19.23_{\pm2.26}$ | $17.61_{\pm2.77}$ | $33.08_{\pm0.66}$ | $17.04_{\pm0.78}$ | $15.98_{\pm0.93}$ | $22.59_{\pm2.75}$ | $7.62_{\pm2.50}$ | $7.76_{\pm0.85}$ |
| | SMOTE | $25.44_{\pm1.88}$ | $16.97_{\pm3.19}$ | $17.38_{\pm2.78}$ | $32.85_{\pm0.95}$ | $15.09_{\pm1.23}$ | $16.85_{\pm2.51}$ | $21.35_{\pm2.76}$ | $9.41_{\pm2.67}$ | $6.09_{\pm0.62}$ |
| | GSMOTE | $21.23_{\pm1.48}$ | $18.02_{\pm2.62}$ | $19.06_{\pm2.28}$ | $24.21_{\pm3.06}$ | $14.83_{\pm0.95}$ | $19.11_{\pm1.73}$ | $20.08_{\pm3.77}$ | $5.99_{\pm1.49}$ | $8.27_{\pm0.75}$ |
| | Best | 18.84 | 16.78 | 17.00 | 22.43 | 14.83 | 15.98 | 14.40 | 5.99 | 5.30 |

Table 14: Performance of TOBE under varying types and levels of class imbalance. For each setting, we report the relative gain over base and mark the best/second-best score in **bold**/underlined.

| Dataset | | Cora | | CiteSeer | | PubMed | | CS | | Physics | |
|---|---|---|---|---|---|---|---|---|---|---|---|
| **Step IR** | | 10 | 20 | 10 | 20 | 10 | 20 | 10 | 20 | 10 | 20 |
| BAcc↑ | Base | 61.6 | 52.7 | 37.6 | 34.2 | 64.2 | 60.8 | 75.4 | 65.3 | 80.1 | 67.7 |
| | + ToBE | $69.8_{+13.4\%}$ | $71.3_{+35.2\%}$ | $55.4_{+47.2\%}$ | $51.3_{+49.9\%}$ | $68.6_{+6.8\%}$ | $63.3_{+4.1\%}$ | $82.6_{+9.6\%}$ | $79.9_{+22.2\%}$ | $87.6_{+9.4\%}$ | $88.0_{+29.9\%}$ |
| | BestIGL | $70.1_{+13.9\%}$ | $66.5_{+26.2\%}$ | $56.0_{+48.9\%}$ | $47.2_{+38.0\%}$ | $74.0_{+15.2\%}$ | $71.1_{+17.0\%}$ | $84.1_{+11.6\%}$ | $81.3_{+24.4\%}$ | $89.4_{+11.6\%}$ | $85.7_{+26.6\%}$ |
| | + ToBE | $74.2_{+20.6\%}$ | $71.6_{+35.9\%}$ | $62.7_{+66.6\%}$ | $62.5_{+82.6\%}$ | $76.9_{+19.7\%}$ | $75.7_{+24.5\%}$ | $86.3_{+14.5\%}$ | $85.6_{+31.0\%}$ | $91.2_{+13.9\%}$ | $90.9_{+34.2\%}$ |
| Macro-F1↑ | Base | 60.1 | 47.0 | 28.1 | 21.9 | 55.1 | 46.4 | 72.7 | 59.2 | 80.7 | 64.7 |
| | + ToBE | $68.7_{+14.3\%}$ | $69.6_{+48.1\%}$ | $54.9_{+95.8\%}$ | $48.9_{+123.5\%}$ | $67.2_{+21.9\%}$ | $60.7_{+30.8\%}$ | $78.6_{+8.1\%}$ | $74.7_{+26.1\%}$ | $88.8_{+10.0\%}$ | $87.8_{+35.8\%}$ |
| | BestIGL | $70.0_{+16.4\%}$ | $66.2_{+40.9\%}$ | $54.5_{+94.1\%}$ | $45.0_{+105.6\%}$ | $71.3_{+29.4\%}$ | $68.9_{+48.3\%}$ | $83.9_{+15.3\%}$ | $80.9_{+36.7\%}$ | $89.5_{+10.9\%}$ | $86.2_{+33.2\%}$ |
| | + ToBE | $72.8_{+21.2\%}$ | $70.2_{+49.4\%}$ | $62.5_{+122.7\%}$ | $62.1_{+183.6\%}$ | $76.9_{+39.5\%}$ | $74.9_{+61.2\%}$ | $85.4_{+17.5\%}$ | $84.6_{+43.0\%}$ | $90.7_{+12.4\%}$ | $90.0_{+39.2\%}$ |
| PerfStd↓ | Base | 27.9 | 39.0 | 29.9 | 35.1 | 34.7 | 41.5 | 21.2 | 32.1 | 22.2 | 36.0 |
| | + ToBE | $21.3_{-23.7\%}$ | $24.4_{-37.5\%}$ | $13.9_{-53.5\%}$ | $16.7_{-52.5\%}$ | $21.8_{-37.2\%}$ | $29.1_{-29.9\%}$ | $17.4_{-18.2\%}$ | $22.9_{-28.8\%}$ | $11.5_{-48.3\%}$ | $25.6_{-29.0\%}$ |
| | BestIGL | $20.0_{-28.1\%}$ | $21.9_{-43.8\%}$ | $16.9_{-43.4\%}$ | $18.0_{-48.6\%}$ | $11.9_{-65.6\%}$ | $14.2_{-65.7\%}$ | $8.9_{-58.3\%}$ | $12.3_{-61.8\%}$ | $6.3_{-71.7\%}$ | $12.4_{-65.5\%}$ |
| | + ToBE | $15.2_{-45.3\%}$ | $17.5_{-55.2\%}$ | $13.9_{-53.5\%}$ | $16.7_{-52.5\%}$ | $5.1_{-85.2\%}$ | $4.6_{-89.0\%}$ | $7.9_{-62.7\%}$ | $10.1_{-68.5\%}$ | $6.6_{-70.2\%}$ | $6.9_{-80.8\%}$ |
| **Natural IR** | | 50 | 100 | 50 | 100 | 50 | 100 | 50 | 100 | 50 | 100 |
| BAcc↑ | Base | 58.1 | 61.8 | 44.9 | 44.7 | 52.0 | 51.1 | 73.8 | 71.4 | 76.0 | 77.7 |
| | + ToBE | $69.1_{+18.9\%}$ | $68.3_{+10.6\%}$ | $58.4_{+29.9\%}$ | $57.4_{+28.5\%}$ | $55.6_{+7.0\%}$ | $56.5_{+10.4\%}$ | $82.1_{+11.3\%}$ | $81.9_{+14.8\%}$ | $86.9_{+14.3\%}$ | $84.1_{+8.3\%}$ |
| | BestIGL | $71.0_{+22.3\%}$ | $73.8_{+19.5\%}$ | $56.3_{+25.3\%}$ | $56.3_{+26.0\%}$ | $72.7_{+39.8\%}$ | $72.8_{+42.5\%}$ | $81.2_{+10.0\%}$ | $81.4_{+14.0\%}$ | $85.8_{+12.9\%}$ | $87.2_{+12.2\%}$ |
| | + ToBE | $73.1_{+25.8\%}$ | $76.9_{+24.5\%}$ | $62.1_{+38.2\%}$ | $61.3_{+37.3\%}$ | $75.8_{+45.7\%}$ | $75.9_{+48.5\%}$ | $85.0_{+15.1\%}$ | $84.5_{+18.5\%}$ | $88.6_{+16.5\%}$ | $89.7_{+15.4\%}$ |
| Macro-F1↑ | Base | 58.7 | 61.4 | 37.5 | 36.2 | 47.3 | 45.1 | 75.3 | 73.2 | 78.0 | 79.8 |
| | + ToBE | $68.7_{+17.1\%}$ | $67.5_{+10.0\%}$ | $57.1_{+52.6\%}$ | $55.8_{+54.3\%}$ | $52.8_{+11.6\%}$ | $52.0_{+15.4\%}$ | $82.6_{+9.7\%}$ | $82.6_{+12.8\%}$ | $87.6_{+12.3\%}$ | $85.2_{+6.8\%}$ |
| | BestIGL | $71.1_{+21.2\%}$ | $73.4_{+19.5\%}$ | $54.3_{+44.8\%}$ | $53.8_{+48.8\%}$ | $72.9_{+53.9\%}$ | $73.7_{+63.6\%}$ | $82.5_{+9.5\%}$ | $82.4_{+12.6\%}$ | $87.7_{+12.4\%}$ | $88.3_{+10.6\%}$ |
| | + ToBE | $72.7_{+23.9\%}$ | $76.0_{+23.9\%}$ | $60.2_{+60.8\%}$ | $59.4_{+64.3\%}$ | $75.3_{+59.2\%}$ | $76.1_{+68.8\%}$ | $85.7_{+13.7\%}$ | $85.1_{+16.2\%}$ | $88.8_{+13.8\%}$ | $89.4_{+12.0\%}$ |
| PerfStd↓ | Base | 28.8 | 31.0 | 38.7 | 39.8 | 36.2 | 38.2 | 26.3 | 28.2 | 23.8 | 21.0 |
| | + ToBE | $18.3_{-36.4\%}$ | $25.4_{-18.1\%}$ | $24.9_{-35.6\%}$ | $33.1_{-17.0\%}$ | $33.3_{-8.1\%}$ | $35.9_{-6.2\%}$ | $19.0_{-27.9\%}$ | $19.5_{-30.9\%}$ | $17.0_{-28.7\%}$ | $19.6_{-6.7\%}$ |
| | BestIGL | $18.9_{-34.4\%}$ | $17.3_{-44.4\%}$ | $28.7_{-25.9\%}$ | $29.7_{-25.3\%}$ | $6.0_{-83.4\%}$ | $9.6_{-75.0\%}$ | $14.4_{-45.4\%}$ | $15.4_{-45.5\%}$ | $11.2_{-53.1\%}$ | $9.7_{-53.8\%}$ |
| | + ToBE | $15.9_{-44.8\%}$ | $14.7_{-52.8\%}$ | $21.9_{-43.4\%}$ | $19.8_{-50.3\%}$ | $4.2_{-88.3\%}$ | $5.6_{-85.3\%}$ | $12.2_{-53.5\%}$ | $12.8_{-54.7\%}$ | $7.4_{-68.9\%}$ | $7.2_{-65.7\%}$ |

