# OpenReview forum: "Class-Imbalanced Graph Learning without Class Rebalancing"
_ICLR.cc/2024/Conference — Submitted to ICLR 2024_

### Official Review · Reviewer_ndZy · 2023-10-28

**Soundness:** 3 good
**Presentation:** 3 good
**Contribution:** 3 good
**Rating:** 6
**Confidence:** 4

**Summary:**

The paper addresses the problem of class imbalance in node classification tasks. Authors introduces ToBE (Topological Balanced augmEntation), a model-agnostic technique. The essence of ToBE is dynamic topological augmentation to identify and rectify nodes that are critically influenced by the identified challenges. Results shows promising improvements including reducing bias and enhancing performance in class-imbalanced node classification, outperforming traditional CR techniques.

**Strengths:**

1. Instead of the conventional class-rebalancing methods, the paper provides a topological viewpoint to address the issue.

2. The paper offers a theoretical understanding of the disparities in graph topology between majority and minority classes, leading to a deeper comprehension of the root causes of the problem.

3. ToBE is model-agnostic and efficient, and can be easily integrated with other existing techniques.

4. Experiments validate the efficacy of ToBE, showcasing its superiority in terms of performance, robustness, and versatility.

**Weaknesses:**

1. The augmentation method introduced in this paper could potentially escalate the computational complexity, presenting challenges for deployment in large-scale scenarios. It might be beneficial to either highlight this as a potential limitation or to delve into a theoretical analysis addressing its implications in expansive applications.

2. As with any data modification technique, augmentation inherently brings the risk of overfitting. It's crucial to recognize this aspect and perhaps consider empirical evaluations or additional experiments to shed light on this concern, suggesting possible mitigation strategies.

3. The manuscript remains silent on the performance of the proposed methodology in multi-class classification environments, as well as its adaptability to tasks other than node classification. Exploring these facets could provide a more comprehensive view of its applicability.

4. While the topological strategy presented is innovative, it seems particularly designed for graph-centric challenges. This specialized focus might limit its direct utility in varied domains, and acknowledging this could provide a more grounded perspective.

**Questions:**

See above

---

> ### Author Response · Authors · 2023-11-15
> **Response to Reviewer ndZy (Part 1)**
>
> **We appreciate for your thoughtful review as well as the time and effort you have taken to evaluate our work. We have carefully considered your comments and criticisms, please see our itemized responses to each of your concerns:**
>
> ## (W1) Computational complexity
>
> **TL;DR: ToBE introduces light computational overhead in practice. We addressed this concern in the "Experiment - Complexity Analysis" section and provided a comprehensive analysis of ToBE's time and space complexity. We will move it to the methodology section to improve the readability.**
>
>
> In short, ToBE$_0$ and ToBE$_1$ exhibits $\mathcal{O}(|\mathcal{V}|C)$ and $\mathcal{O}(|\mathcal{E}|C)$ space/computation complexity, where $|\mathcal{V}|$ is the number of nodes, $|\mathcal{E}|$ is the number of edges, and $C$ is the number of classes, which is typically small in practice. Moreover, since all operations can be executed in parallel in matrix form, the time complexity of ToBE$_0$/ToBE$_1$ can be further reduced to $\mathcal{O}(\frac{|\mathcal{V}|C}{D})$ / $\mathcal{O}(\frac{|\mathcal{E}|C}{D})$, where D is the number of available computing units, often large for modern GPUs.
>
> We provide computational efficiency results of ToBE on real-world graph data in Table 3. Additionally, in Appendix D.2, we discuss strategies for further accelerating ToBE in practical applications.
>
> ## (W2) Risk of overfitting
>
> **TL;DR: Further discussions and experiments show that ToBE can mitigate node and neighbor overfitting induced by static message-passing. ToBE increase the topological structure diversity of minority class nodes without discarding or duplicating instances, which may cause overfitting.**
>
> Great question! Existing studies have shown that the minority class is more susceptible to overfitting due to the limited number of training samples [1, 2]. Properly designed oversampling methods can synthesize more diverse minority samples and mitigate such overfitting [1, 3]. Although ToBE does not directly synthesize samples, it dynamically augments the topological structure to increase the diversity of neighbors for minority class nodes. Therefore, ToBE can mitigate node and neighbor memorization issues [4] (i.e., overfitting) induced by static message-passing.
>
> To empirically show how ToBE can help prevent overfitting, we conducted experiments involving node-replacing and neighbor-replacing:
> 1. **Node replacing**: We replaced the features of minority class training (seen) nodes with features from unseen nodes of the same class. This test evaluates the model's susceptibility to node feature overfitting.
> 2. **Neighbor replacing**: On graph data, message-passing-based GNNs may also overfit the neighborhood of minority class nodes (i.e., neighbor memorization [4]). Towards this, we retain the features of minority training nodes, but replaced their neighbors with unseen nodes. This test evaluates the model's susceptibility to neighbor set overfitting.
>
> Under IR=10, we conducted 5 independent runs, recording the model's accuracy on the training set after each run. Subsequently, we performed 1000 independent node/neighbor-replacing experiments. The results are shown in the tables below.
>
> **Observations**: While GNN can achieve 100% accuracy on the training set, its predictive performance on training nodes drastically declined after node/neighbor replacing, indicating significant overfitting to features and neighbor sets. However, **this performance degradation was substantially mitigated by using ToBE, demonstrating its effectiveness in preventing overfitting.**
>
> **On the Cora Dataset**
>
> | Setting | Base | +ToBE$_0$ | +ToBE$_1$ |
> | ------- | ---- | --------- | --------- |
> | Training Acc | 1.000 | 1.000 | 1.000 |
> | w/ Node Replacing | 0.403±0.203 | 0.755±0.207 | 0.839±0.160 |
> | w/ Neighbor Replacing | 0.762±0.195 | 0.924±0.110 | 0.917±0.110 |
>
> **On the CiteSeer Dataset**
>
> | Setting | Base | +ToBE$_0$ | +ToBE$_1$ |
> | ------- | ---- | --------- | --------- |
> | Training Acc | 1.000 | 1.000 | 1.000 |
> | w/ Node Replacing | 0.600±0.203 | 0.970±0.070 | 0.858±0.243 |
> | w/ Neighbor Replacing | 0.604±0.239 | 0.981±0.055 | 0.842±0.197 |
>
> **On the PubMed Dataset**
>
> | Setting | Base | +ToBE$_0$ | +ToBE$_1$ |
> | ------- | ---- | --------- | --------- |
> | Training Acc | 1.000 | 1.000 | 1.000 |
> | w/ Node Replacing | 0.547±0.367 | 0.934±0.172 | 0.926±0.181 |
> | w/ Neighbor Replacing | 0.449±0.354 | 0.915±0.202 | 0.906±0.211 |

---

> ### Author Response · Authors · 2023-11-15
> **Response to Reviewer ndZy (Part 2)**
>
> ## (W3) Multi-classification and potential future directions
>
> **TL;DR: Our work does indeed focus on multi-class classification scenarios (Appendix C.1). We will incorporate a more in-depth discussions on limitations and possible future directions.**
>
> **Regarding multi-classification**: We want to clarify that our work does indeed focus on multi-class classification scenarios. For further clarification, please refer to Appendix C.1, where the number of classes for each dataset used in the experiments is detailed (Table 4). Additionally, we provide a visualization of class distributions across different settings in Figure 8.
>
> **Regarding tasks beyond node classification**:
> Thank you for your valuable feedback on potential areas for further exploration. We acknowledge the importance of extending ToBE to other imbalanced graph learning tasks. For instance, imbalanced edge prediction is a less-explored yet crucial direction (e.g., for knowledge graph completion). This involves imbalance at both the edge level (presence/absence) and the node/entity level (some entities having more relationships). Adapting the existing CIGL framework to address these challenges is non-trivial. We will incorporate a more in-depth discussion of these possibilities in the manuscript.
>
> ## (W4) From graph to more general domains
>
> **We believe that ToBE has the potential to extend beyond graph-centric challenges and be applied to non-graph tasks.** There are recent works discussing how to extract graph structures from i.i.d. tabular data and facilitate representation learning [5, 6]. Exploring the extension of ToBE from this perspective to aid diverse non-graph structure data learning is an exciting direction. We will incorporate additional discussion on these possibilities in the manuscript.
>
> **Thank you once again for your feedback. We hope we have addressed our concerns, and we welcome any follow-up question or discussion.**
>
> ## References
> [1] Johnson, Justin M., and Taghi M. Khoshgoftaar. "Survey on deep learning with class imbalance." Journal of Big Data 6.1 (2019): 1-54.
> [2] Li, Zeju, Konstantinos Kamnitsas, and Ben Glocker. "Overfitting of neural nets under class imbalance: Analysis and improvements for segmentation." Medical Image Computing and Computer Assisted Intervention–MICCAI 2019: 22nd International Conference, Shenzhen, China, October 13–17, 2019, Proceedings, Part III 22. Springer International Publishing, 2019.
> [3] Gosain, Anjana, and Saanchi Sardana. "Handling class imbalance problem using oversampling techniques: A review." 2017 international conference on advances in computing, communications and informatics (ICACCI). IEEE, 2017.
> [4] Park, Joonhyung, Jaeyun Song, and Eunho Yang. "Graphens: Neighbor-aware ego network synthesis for class-imbalanced node classification." International Conference on Learning Representations. 2021.
> [5] Du, Lun, et al. "TabularNet: A neural network architecture for understanding semantic structures of tabular data." Proceedings of the 27th ACM SIGKDD Conference on Knowledge Discovery & Data Mining. 2021.
> [6] Zhou, Kaixiong, et al. "Table2graph: Transforming tabular data to unified weighted graph." Proceedings of the Thirty-First International Joint Conference on Artificial Intelligence, IJCAI. 2022.

---

> ### Author Response · Authors · 2023-11-21
> **Looking for your response**
>
> **Dear Reviewer ndZy,**
>
> Thank you again for your time and effort in reviewing this submission. This is just a kind reminder that the discussion period is coming to an end. For your convenience, we provide a summary here to make it easier for you to locate our responses to each of your concerns:
>
> - **Part 1:**
>     - Discussions on computational complexity (analysis provided in the paper)
>     - Further analysis and experiments on how ToBE alleviates overfitting
> - **Part 2:**
>     - Clarification on multi-classification, discussions on limitations and furture directions
>     - Discussions on extending ToBE to more general non-graph domains
>
> **Your opinion is important to us.** We would like to know whether our responses address your concerns properly, and we are more than happy to address any remaining points.
>
> Sincerely, Authors

---

> ### Comment · Reviewer_ndZy · 2023-11-22
> **Thank you**
>
> Thank you for your detailed rebuttals. After reading all the rebuttals, including those from other reviewers, I would like to adjust my initial score from 'borderline accept' (6) to 'weak accept' (7). Unfortunately, there isn't a specific selection for 'weak accept,' as the next available option is 'accept' with a score of 8. Therefore, I have decided to maintain my current scores.
>
> To all ACs and SACs, please consider my suggestions as a 'weak accept.' Thank you for your understanding.

---

> ### Author Response · Authors · 2023-11-23
> **Thank you for the positive response!**
>
> **Dear Reviewer ndZy,**
>
> Thank you very much for your response. To thoroughly address your concerns, we have diligently revised the manuscript and uploaded the latest version. Please check the **[the updated PDF](https://openreview.net/pdf?id=bCNYFOaWsy)** with significant changes highlighted in **blue** for your convenience. **We believe this version demonstrates substantial improvements with stronger theoretical analyses, more comprehensive experiments, and related discussions.**
>
> Key revisions include:
> - For W1, we have revised **the end of Section 3** Methodology for better clarity.
> - For W2, we included **Appendix D.3** to provide a detailed discussion on ToBE's ability to mitigate overfitting.
> - For W3 and W4, we incorporated discussions on additional graph learning tasks and the extension of ToBE to broader non-graph learning tasks in **Appendix D.8**.
>
> Furthermore, several other noteworthy improvements have been made, such as (1) a reformed of the theoretical analysis for AMP/DMP, extending it from k=2 to the general case (**Appendix A**); (2) validation of ToBE's effectiveness not only in transductive scenarios but also in aiding GNN generalization to unseen graphs (**Appendix D.4**); (3) further discussion on the advantages and limitations of the two variants of ToBE and practical considerations for their selection (**Appendix D.7**).
>
> **We hope that your concerns have been thoroughly addressed in this version of the paper. If so, we sincerely hope for your support in accepting our paper, and we are truly grateful for your consideration.** We will continue to carefully improve the paper’s organization, making it more accessible for broader groups of audiences to comprehend and appreciate our research.
>
> Thank you once again,
> The Authors

---

### Official Review · Reviewer_UMPW · 2023-10-30

**Soundness:** 2 fair
**Presentation:** 2 fair
**Contribution:** 1 poor
**Rating:** 5
**Confidence:** 5

**Summary:**

This paper addresses the problem of class imbalance graph learning. It first revisits several remaining issues of existing methods. Then, from an orthogonal topological paradigm, they theoretically find two fundamental reasons. After that, they propose a lightweight topological augmentation framework called TOBE to mitigate the class-imbalance bias without class rebalancing. Finally, it conducts some experiments to evaluate the proposed method, showing that that TOBE can sometime outperform state-of-the-art baselines on several tasks across multiple datasets.

**Strengths:**

1.	The authors provide their codes.
2.	It provides some theoretical support for the proposed model.
3.	It tests on several widely-used datasets, and the proposed method can sometimes beat the existing methods.

**Weaknesses:**

1.	The proposed method seems to be meaningless at times. As shown in Table, sometimes the original methods (such as APPNP and GPRGNN) can beat either +ToBE0 or + ToBE1. More over, as shown in Tables 7 and 8, lots of baselines (Vanilla, Reweight, ReNode, and GSMOTE) can sometimes beat either +ToBE0 or + ToBE1.
2.	Some grammatical errors, like 1) groundtruth labels –> “ground-truth”; 2) Coauthor networks-> “co-author”; and 3) “Fig. 5 compares”  “Figure”.

**Questions:**

1.	As shown in Tables 7 and 8, why lots of baselines can sometimes beat both ToBE0 and ToBE1? As such, why the proposed method is useful?
2.	As we can see, the performance of ToBE0 and ToBE1 are unpredictable. So that, how to decide which one should be used for a given method or setting?
3.	“Similar analysis can extend to k ≥ 3.” --- have you ever proved this?
4.	In Table 3, why the “Node” column only has one type of results?
5.	See the weakness in the “*Weaknesses” part.

---

> ### Author Response · Authors · 2023-11-17
> **Response to Reviewer UMPW (Part 1)**
>
> **We appreciate for your thoughtful review as well as the time and effort you have taken to evaluate our work. We have carefully considered your comments and criticisms, please see our itemized responses to each of your concerns:**
>
> ## (W1, Q1) Regarding the effectiveness of ToBE
>
> **TL;DR: Empirically, ToBE is effective in over 98% of the cases (103/105) across all performance metrics (Table 7/8/9). The rare cases that ToBE does not work well are caused by oversmoothing from the multi-step message-passing in APPNP/GPRGNN, and we show that such senario can be prevented by simply using smaller propagaion step in the base model (APPNP/GPRGNN).**
>
> ### **Why is the proposed method useful?**
>
> We note that for a same model, Tables 7 and 8 report Balanced Accuracy and Macro-F1, respectively. Evaluating the effectiveness of ToBE requires a comprehensive consideration of both aspects.
>
> 1. The scenario described in Q1 ("beat both ToBE$_0$ and ToBE$_1$") occurs in only **2 out of 105 experimental conditions** (3 datasets x 7 baselines x 5 GNN backbones). Specifically,
>    - For BAcc: on the PubMed dataset [APPNP, APPNP+ReNode];
>    - For Macro-F1: on the Cora dataset [GPRGNN+Reweight, GPRGNN+ReNode].
> 2. **Jointly considering BAcc and Macro-F1, no baseline can outperform both ToBE$_0$ and ToBE$_1$**, e.g.,
>    - For the PubMed dataset [APPNP, APPNP+ReNode]: Both ToBE$_0$ and ToBE$_1$ boosted their Macro-F1;
>    - Similarly, for the Cora dataset [GPRGNN+Reweight, GPRGNN+ReNode]: Both ToBE$_0$ and ToBE$_1$ boosted their BAcc.
> 3. Our work provides one of the most comprehensive experimental setups in existing literature, covering 2 imbalance types, 4 imbalance ratios, combinations of the proposed method with 5 GNN architectures and 7 baseline methods. We believe that maintaining effectiveness in **over 98% of the cases (103/105) across all performance metrics** in such a comprehensive experimental setup demonstrates the usefulness of ToBE.
>
> ### **Case study on APPNP and GPRGNN.**
>
> ***The reason behind***: We now conduct a deeper analysis on the undesired cases we discussed above. As can be observed (and suggested in W1), these cases are related to APPNP and GPRGNN. We note that **both APPNP and GPRGNN involve multi-step message-passing** based on Personalized PageRank (PPR).  Additionally, due to the introduction of virtual edges, **ToBE-enhanced graphs exhibit improved connectivity**, potentially leading to **over-smoothing issues** when combined with these models. From this perspective, we also observed that:
> 1. Reweight and ReNode only reweight training nodes, so the issue persists.
> 2. Oversampling methods like Resample, GSMOTE, and GENS introduce additional nodes, alleviating over-smoothing. Therefore, ToBE performs better when combined with these methods (as can be observed in Tables 7 and 8).
>
> ***Empirical validation***: Furthermore, we conducted experiments to validate the above issues. As described in Appendix C.2, following the original papers' settings, we set the number of propagation steps K for APPNP/GPRGNN to 10. However, when combined with ToBE, we should reduce K to alleviate over-smoothing. We tested K=10/5/3/1 on the cases where ToBE failed, and the best results in each column are highlighted in bold. Detailed results are listed in the tables below.
>
> We can observe that APPNP/GPRGNN typically achieve the best results with K=10. However, **when combined with ToBE, smaller values of K can cope with the increased graph connectivity, avoiding over-smoothing and outperforming the best results of the original models**. We will include a discussion of over-smoothing in the appendix and point out the caveats of using ToBE in conjunction with multi-step message-passing GNNs.
>
> **PubMed - APPNP - Vanilla**
>
> | Metric      | BAcc           |                |                | F1             |                |                |
> | ----------- | -------------- | -------------- | -------------- | -------------- | -------------- | -------------- |
> | #Prop. Step | Base           | +ToBE$_0$      | +ToBE$_1$      | Base           | +ToBE$_0$      | +ToBE$_1$      |
> | k=10        | **59.30**±0.50 | 55.62±0.31     | 57.82±0.29     | 44.50±0.21     | 44.67±0.07     | 44.59±0.06     |
> | k=5         | 59.07±0.22     | 58.56±0.14     | 59.00±0.51     | 44.70±0.08     | 45.69±0.49     | 44.68±0.15     |
> | k=3         | 59.12±0.09     | 59.85±0.26     | 60.11±0.31     | **45.18±0.15** | 45.76±0.56     | 46.36±1.38     |
> | k=1         | 58.97±0.11     | **59.99**±0.17 | **62.91**±1.91 | 45.01±0.08     | **49.09**±1.42 | **49.40**±4.38 |

---

> ### Author Response · Authors · 2023-11-17
> **Response to Reviewer UMPW (Part 2)**
>
> **PubMed - APPNP - ReNode**
>
> | Metric      | BAcc           |                |                | F1             |                |                |
> | ----------- | -------------- | -------------- | -------------- | -------------- | -------------- | -------------- |
> | #Prop. Step | Base           | +ToBE$_0$      | +ToBE$_1$      | Base           | +ToBE$_0$      | +ToBE$_1$      |
> | k=10        | 72.76±1.37     | 71.54±0.96     | 71.88±0.70     | **71.45**±1.19 | 75.08±1.06     | 75.76±0.84     |
> | k=5         | **73.24**±1.85 | 74.26±1.44     | 73.66±1.00     | 70.09±2.38     | 74.74±1.35     | 73.93±0.89     |
> | k=3         | 72.42±1.77     | 73.76±1.49     | 74.04±1.08     | 69.05±2.13     | **75.58**±1.34 | 74.04±0.99     |
> | k=1         | 68.87±1.33     | **75.19**±1.17 | **76.72**±0.79 | 63.72±1.61     | 75.06±1.19     | **76.65**±0.78 |
>
> **Cora - GPRGNN - Reweight**
>
> | Metric      | BAcc           |                |                | F1             |                |                |
> | ----------- | -------------- | -------------- | -------------- | -------------- | -------------- | -------------- |
> | #Prop. Step | Base           | +ToBE$_0$      | +ToBE$_1$      | Base           | +ToBE$_0$      | +ToBE$_1$      |
> | k=10        | **72.15**±0.57 | 72.90±1.33     | 73.22±0.55     | **71.66**±0.85 | 70.46±0.98     | 71.24±0.49     |
> | k=5         | 71.52±0.17     | **73.90**±1.18 | 74.87±1.06     | 71.27±0.38     | **72.02**±1.22 | **72.90**±0.93 |
> | k=3         | 70.90±0.88     | 72.76±1.00     | **75.16**±1.15 | 70.30±0.89     | 69.72±0.92     | 72.61±0.94     |
> | k=1         | 62.44±0.90     | 71.27±0.66     | 72.46±0.60     | 61.49±1.19     | 68.64±0.59     | 70.30±0.55     |
>
> **Cora - GPRGNN - ReNode**
>
> | Metric      | BAcc           |                |                | F1             |                |                |
> | ----------- | -------------- | -------------- | -------------- | -------------- | -------------- | -------------- |
> | #Prop. Step | Base           | +ToBE$_0$      | +ToBE$_1$      | Base           | +ToBE$_0$      | +ToBE$_1$      |
> | k=10        | **73.38**±0.67 | 73.71±0.57     | 73.93±1.60     | **73.08**±0.66 | 71.52±0.50     | 71.72±1.51     |
> | k=5         | 72.89±1.21     | 73.72±0.93     | 74.32±1.17     | 72.74±1.15     | 72.93±0.71     | 72.12±1.02     |
> | k=3         | 71.77±0.63     | **74.35**±0.92 | **74.55**±1.30 | 71.82±0.66     | **73.25**±0.60 | **73.98**±1.24 |
> | k=1         | 63.47±1.11     | 71.10±0.78     | 72.33±0.90     | 62.85±1.42     | 68.80±0.75     | 70.02±0.79     |
>
> ## (Q2) Choose between ToBE$_0$ and ToBE$_1$
>
> **TL;DR: We recommend using ToBE$_1$ to achieve better classification performance. But in case that computational resources are limited, ToBE$_0$ can serve as a more efficient alternative.**
>
> Thank you for your question, it is crucial for the practical application of ToBE. We will include a detailed discussion in the paper to help readers understand the strengths and limitations of ToBE$_0$ and ToBE$_1$. Here is a brief analysis:
>
> We observe a clear performance gap between ToBE$_0$ and ToBE$_1$, with ToBE$_1$ generally exhibiting better classification performance due to its consideration of local topological structure. In the 15 scenarios presented in Table 1 (the best CIGL score of 3 datasets x 5 GNN backbones):
> 1. ToBE$_1$ outperforms ToBE$_0$ in BAcc/F1 scores significantly in 12/11 scenarios, with an average advantage of 1.692 for F1 in the 11 leading cases.
> 2. In comparison, ToBE$_0$ has a less pronounced advantage in the few cases where it leads, with an average advantage of 0.518 in the 4 leading scenarios.
> 3. Despite the relative performance advantage of ToBE$_1$, both ToBE$_0$ and ToBE$_1$ significantly enhance the best-performing CIGL baseline methods. In the 15 scenarios, ToBE$_0$ provides an average improvement of 4.789/5.848 in BAcc/F1, while ToBE$_1$ brings an average improvement of 5.805/6.950.
>
> But on the other hand, as discussed in the "Experiments - Complexity analysis" section, ToBE$_1$ generally incurs higher time and space complexity compared to ToBE$_0$. Specifically, ToBE$_0$ exhibits linear complexity with respect to the number of nodes, whereas ToBE$_1$'s complexity grows linearly with the number of edges. In real-world graph data, the number of edges is often significantly larger than the number of nodes. Thus ToBE$_0$ is more efficient.

---

> ### Author Response · Authors · 2023-11-17
> **Response to Reviewer UMPW (Part 3)**
>
> ## (Q3) Theoretical analysis for k>3
>
> As mentioned in the paper, our analysis can extend to $k\\ge3$, but the results will look messier. To demonstrate that, we provide a sketch analysis for general $k$ using the same ideas for $k=2$.
>
> Note that the results of both Theorems 1 & 2 depend only on the distributions of $H_{ij}$. Thus, as long as we can calculate the distributions of $H_{ij}$, Theorems 1 & 2 follows. Indeed, the distributions of $H_{ij}$ for general $k$ can still be calculated via the BFS tree counting approach in our proof, i.e.,
> $$
> \\begin{aligned}
> \\mathbb P\\{H_{11}=s\\}={}&\\sum_{a_1+\\cdots+a_k=s}\\binom{n_1-1}{a_1,\\dots,a_k,n_1-1-s}p^{a_1}\\bigg(\\prod_{t=2}^k(1-p)^{a_t(1+a_1+\\cdots+a_{t-2})}(1-(1-p)^{a_{t-1}})^{a_t}\\bigg)(1-p)^{(n_1-1-s)(1+s-a_k)}\\\\
> \\sim{}&\\sum_{a_1+\\cdots+a_k=s}\\frac{n_1^s}{a_1!\\cdots a_k!}p^{a_1}\\bigg(\\prod_{t=2}^k(a_{t-1}p)^{a_t}\\bigg)(1-p)^{n_1(1+s-a_k)}\\\\
> \\to{}&\\mathrm e^{-\\beta_{11}}\\sum_{a_1+\\cdots+a_k=s}\\frac{(\\beta_{11}\\mathrm e^{-\\beta_{11}})^{a_1}}{a_1!}\\bigg(\\prod_{t=2}^{k-1}\\frac{(a_{t-1}\\beta_{11}\\mathrm e^{-\\beta_{11}})^{a_t}}{a_t!}\\bigg)\\frac{(a_{k-1}\\beta_{11})^{a_k}}{a_k!},\\\\
> \\mathbb P\\{H_{12}=s\\}={}&\\sum_{a_1+\\cdots+a_k=s}\\binom{n_2}{a_1,\\dots,a_k,n_2-s}q^{a_1}\\bigg(\\prod_{t=2}^k(1-q)^{a_t}(1-p)^{a_t(a_1+\\cdots+a_{t-2})}(1-(1-p)^{a_{t-1}})^{a_t}\\bigg)(1-q)^{n_2-s}(1-p)^{(n_2-s)(s-a_k)}\\\\
> \\sim{}&\\sum_{a_1+\\cdots+a_k=s}\\frac{n_2^s}{a_1!\\cdots a_k!}q^{a_1}\\bigg(\\prod_{t=2}^k(a_{t-1}p)^{a_t}\\bigg)(1-q)^{n_2}(1-p)^{n_2(s-a_k)}\\\\
> \\to{}&\\mathrm e^{-\\beta_{12}}\\sum_{a_1+\\cdots+a_k=s}\\frac{(\\beta_{12}\\mathrm e^{-\\beta_{22}})^{a_1}}{a_1!}\\bigg(\\prod_{t=2}^{k-1}\\frac{(a_{t-1}\\beta_{22}\\mathrm e^{-\\beta_{22}})^{a_t}}{a_t!}\\bigg)\\frac{(a_{k-1}\\beta_{22})^{a_k}}{a_k!}.
> \\end{aligned}
> $$
> Regarding Theorem 1, note that for any $j=1,\\dots,k$,
> $$
> \\mathrm e^{-\\beta_{11}}\\sum_{a_1=0}^\\infty\\cdots\\sum_{a_k=0}^\\infty a_j \\frac{(\\beta_{11}\\mathrm e^{-\\beta_{11}})^{a_1}}{a_1!}\\bigg(\\prod_{t=2}^{k-1}\\frac{(a_{t-1}\\beta_{11}\\mathrm e^{-\\beta_{11}})^{a_t}}{a_t!}\\bigg)\\frac{(a_{k-1}\\beta_{11})^{a_k}}{a_k!}=\\beta_{11}^j.
> $$
> Thus,
> $$
> \\begin{aligned}
> &\\lim_{n\\to\\infty}\\mathbb E[H_{11}]=\\sum_{s=0}^\\infty s\\cdot\\mathrm e^{-\\beta_{11}}\\sum_{a_1+\\cdots+a_k=s}\\frac{(\\beta_{11}\\mathrm e^{-\\beta_{11}})^{a_1}}{a_1!}\\bigg(\\prod_{t=2}^{k-1}\\frac{(a_{t-1}\\beta_{11}\\mathrm e^{-\\beta_{11}})^{a_t}}{a_t!}\\bigg)\\frac{(a_{k-1}\\beta_{11})^{a_k}}{a_k!}\\\\
> ={}&\\mathrm e^{-\\beta_{11}}\\sum_{a_1=0}^\\infty\\cdots\\sum_{a_k=0}^\\infty(a_1+\\cdots+a_k)\\frac{(\\beta_{11}\\mathrm e^{-\\beta_{11}})^{a_1}}{a_1!}\\bigg(\\prod_{t=2}^{k-1}\\frac{(a_{t-1}\\beta_{11}\\mathrm e^{-\\beta_{11}})^{a_t}}{a_t!}\\bigg)\\frac{(a_{k-1}\\beta_{11})^{a_k}}{a_k!}\\\\
> ={}&\\sum_{j=1}^k\\mathrm e^{-\\beta_{11}}\\sum_{a_1=0}^\\infty\\cdots\\sum_{a_k=0}^\\infty a_j\\frac{(\\beta_{11}\\mathrm e^{-\\beta_{11}})^{a_1}}{a_1!}\\bigg(\\prod_{t=2}^{k-1}\\frac{(a_{t-1}\\beta_{11}\\mathrm e^{-\\beta_{11}})^{a_t}}{a_t!}\\bigg)\\frac{(a_{k-1}\\beta_{11})^{a_k}}{a_k!}\\\\
> ={}&\\sum_{j=1}^k\\beta_{11}^j.
> \\end{aligned}
> $$
> Similarly,
> $$
> \\begin{aligned}
> \\lim_{n\\to\\infty}\\mathbb E[H_{22}]&=\\sum_{j=1}^k\\beta_{22}^{j},\\\\
> \\lim_{n\\to\\infty}\\mathbb E[H_{12}]&=\\sum_{j=1}^k\\beta_{12}\\beta_{22}^{j-1},\\\\
> \\lim_{n\\to\\infty}\\mathbb E[H_{21}]&=\\sum_{j=1}^k\\beta_{21}\\beta_{11}^{j-1}.
> \\end{aligned}
> $$
> Hence,
> $$
> \\lim_{n\\to\\infty}\\frac{\\alpha_1}{\\alpha_2}=\\lim_{n\\to\\infty}\\frac{\\mathbb E[H_{12}]/\\mathbb E[H_{11}]}{\\mathbb E[H_{21}]/\\mathbb E[H_{22}]}=\\frac{\\sum_{j=1}^k\\beta_{12}\\beta_{22}^{j-1}\\big/\\sum_{j=1}^k\\beta_{11}^j}{\\sum_{j=1}^k\\beta_{21}\\beta_{11}^{j-1}\\big/\\sum_{j=1}^k\\beta_{22}^j}=\\bigg(\\rho\\cdot\\frac{\\sum_{j=1}^k(\\rho\\beta)^{j-1}}{\\sum_{j=1}^k\\beta^{j-1}}\\bigg)^2.
> $$
> Regarding Theorem 2, similar analysis applies, but the results get much messier as $k$ grows. For example, for $k=3$, one of the key quantities in the proof of Theorem 2 is
> $$
> \\lim_{n\\to\\infty}\\mathbb E[(1-r_1^{\\text L})^{H_{11}}]=\\mathrm e^{-\\left(1-(1-r_1^{\\text L})\\beta_{11}\\mathrm e^{-\\left(1-(1-r_1^{\\text L})\\beta_{11}\\mathrm e^{-r_1^{\\text L}\\beta_{11}}\\right)\\beta_{11}}\\right)\\beta_{11}}.
> $$
> In general, it has $k$ nested exponentiations, which looks messy. Meanwhile, if we calculate the closed-form expression of $\\lim_{n\\to\\infty}\\frac{\\delta_1}{\\delta_2}$ numerically, then the result for any $k$ will be almost the same as that of $k=2$, i.e.,
> $$
> \\lim_{n\\to\\infty}\\frac{\\delta_1}{\\delta_2}\\approx\\frac{1-r_1^{\\text L}}{1-r_2^{\\text L}}\\mathrm e^{(\\rho-1)\\beta}.
> $$
> That is why we previously chose $k=2$ as a representative case.

---

> ### Author Response · Authors · 2023-11-17
> **Response to Reviewer UMPW (Part 4)**
>
> ## (Q4) The "Node" column in Table 3
>
> As described in Section 3.3, the process of synthesizing virtual nodes for both ToBE$_0$ and ToBE$_1$ is identical: **each class generates one virtual node**. Hence, the "Node" column in Table 3 only presents a single type of result.
>
> ## (W2) Paper writing
>
> We appreciate your meticulous review. We have fixed the errors and will carefully check and revise the entire manuscript.
>
> **Thank you once again for your feedback. We hope we have addressed our concerns, and we welcome any follow-up question or discussion.**

---

> ### Author Response · Authors · 2023-11-21
> **Looking for your response**
>
> **Dear Reviewer UMPW,**
>
> Thank you again for your time and effort in reviewing this submission. This is just a kind reminder that the discussion period is coming to an end. For your convenience, we provide a summary here to make it easier for you to locate our responses to each of your concerns:
> - **Part 1 & 2:**
>     - Discussions regarding the effectiveness of ToBE
>     - Further study and analysis of the critical cases raised in Table 7 & 8
>     - Analysis on choosing between the two ToBE variants
> - **Part 3:**
>     - Sketch analysis for a general $k$
> - **Part 4:**
>     - Clarifications on the "Node" column in Table 3
>
> **Your opinion is important to us.** We would like to know whether our responses address your concerns properly, and we are more than happy to address any remaining points.
>
> Sincerely, Authors

---

> ### Author Response · Authors · 2023-11-23
> **We are looking forward to your reply**
>
> **Dear Reviewer UMPW,**
>
> Thank you again for your invaluable comments, they have greatly helped us to further improve the paper. To thoroughly address your concerns, we have diligently revised the manuscript and uploaded the latest version. Please check the **[the updated PDF](https://openreview.net/pdf?id=bCNYFOaWsy)** with significant changes highlighted in **blue** for your convenience. **We believe this version demonstrates substantial improvements with stronger theoretical analyses, more comprehensive experiments, and related discussions.**
>
> Key revisions include:
> - For W1/Q1, we have included a new section **Appendix D.5** to extend the discussion in our response about combining ToBE with multi-step message-passing GNNs.
> - For W2, we have fixed the typos and carefully checked the rest of the paper.
> - For Q2, we include section **Appendix D.7** to dive deep into the results and give remarks on how to choose between ToBE 0/1 in practice.
> - For Q3, we have reformed our theoretical proof in **Appendix A**, and extend the results to a general $k$.
>
> Furthermore, several other noteworthy improvements have been made, such as (1) validation of ToBE's effectiveness not only in transductive scenarios but also in aiding GNN generalization to unseen graphs (**Appendix D.4**); (2) a comparison with the latest CIGL baselines (**Appendix D.6**); (3) further discussion on the advantages and limitations of the two variants of ToBE and practical considerations for their selection (**Appendix D.7**).
>
> **We hope that your concerns have been thoroughly addressed in this version of the paper. If so, we sincerely hope for your support in accepting our paper, and we are truly grateful for your consideration.** We will continue to carefully improve the paper’s organization, making it more accessible for broader groups of audiences to comprehend and appreciate our research.
>
> Looking forward to your reply and thank you once again,
> The Authors

---

### Official Review · Reviewer_grPF · 2023-11-01

**Soundness:** 1 poor
**Presentation:** 3 good
**Contribution:** 1 poor
**Rating:** 5
**Confidence:** 5

**Summary:**

The study introduces a post-processing module for semi-supervised vertex classification models in the presence of class imbalance. The module aims to mitigate prediction errors and biases caused by class imbalance by incorporating virtual vertices and establishing connections with original vertices exhibiting high prediction errors or low confidence (referred to as high-risk vertices in the paper). Through long-range message propagation, the proposed approach effectively addresses the challenges posed by class imbalance in semi-supervised vertex classification tasks. Experimental results demonstrate its relative improvement over existing models under real-world scenarios characterized by class imbalance.

**Strengths:**

S1. Formulas that accurately describe the AMP and DMP problems were derived, providing precise definitions for these problems.

S2. Extensive experiments consistently show that the proposed post-processing module significantly enhances the learning effectiveness of the current model across multiple metrics and base models.

S3. The writing style is smooth and coherent.

**Weaknesses:**

W1. The lack of comparison with other post-processing modules for class imbalance graph learning, such as the classical Residual Propagation method, undermines the persuasiveness of the proposed method's effectiveness.

W2. The study lacks a comparison between the proposed method and the predictive performance of the base model on balanced data, which diminishes its persuasiveness.

W3. The absence of a comparison with the predictions of the base model on balanced data weakens the persuasiveness of the results.

W4. The use of "relative improvement rate" may not be the most comprehensive measure to evaluate the model's performance, as some base models may inherently perform poorly in addressing highly imbalanced class semi-supervised vertex classification tasks.

W5. The details of how this module collaborates with other base models are not adequately explained, lacking formulas and clear visual representations.

W6. The study's focus is not novel, and the problem scope is narrow. The proposed method has the potential for broader applications, such as imbalanced edge prediction, and should also consider the module's inductive capabilities. Otherwise, solely emphasizing the relative improvement of the existing model's transductive ability may not hold significant practical significance.

**Questions:**

Similar to what was mentioned in the weaknesses:

Q1. How does the effectiveness of this module compare to post-processing modules of other class imbalance graph learning methods?

Q2. How does the performance of this module compare to the base model combined with data balancing during prediction?

Q3. What are the specific formulas and graphical representations illustrating the collaboration between this module and the base model?  Can it be independently developed as a foundational model rather than a post-processing module?

Q4. Can this module be applied to other tasks and demonstrate effectiveness? How does it perform in terms of inductive capabilities?

---

> ### Author Response · Authors · 2023-11-15
> **Response to Reviewer grPF (Part 1)**
>
> **We appreciate for your thoughtful review as well as the time and effort you have taken to evaluate our work. We have carefully considered your comments and criticisms, please see our itemized responses to each of your concerns:**
>
> ## (Q1, Q2, W1) Clarification on TobE's working manner and discussion on potential counterparts
> Thank you for your suggestions! We provide step-by-step clarifications here to address your concerns.
>
> **(W1, Q1) ToBE is not a post-processing module:** We would like to highlight that **ToBE works during the training phase**, thus should be considered as an **in-processing** technique. Specifically,  in each training step, ToBE takes the current model $F$ and original graph $\mathbf{G}$ as input and output augmented graph $\mathbf{G}^*$. Please refer to Algorithm 1 in Appendix B for more details.
>
> **(Q2) We tested ToBE with multiple base models combined with data balancing techniques:**
> As ToBE performs augmentation on the topology structure, it is orthogonal to (and thus can be combined with) other data balancing techniques: they can operate on the augmented graph $\mathbf{G}^*$ for subsequent steps. In the paper we have conducted a comprehensive comparison with relevant data balancing methods, including approaches handling class imbalance by synthesizing new samples (e.g., GraphSMOTE [1], GraphENS [2]) and addressing topological imbalance through modifying loss functions (e.g., ReNode [3]).
>
> > **For example, in Table 1, the row GCN-ReNode corresponds to GCN base model with ReNode [3] data balancing**. The column "Base" report the performance of GCN+ReNode, and the columns "+ToBE$_0$"/"+ToBE$_1$" report the performance after futher incorporating with ToBE. Results show that ToBE boosted the balanced accuracy of GCN+ReNode from 66.60 to 71.37/71.84. Please also refer to Table 7/8/9 in Appendix E for more detailed results.
>
>
> **(Q1, Q2) On potential counterparts:** Other than data augmentation and loss function engineering, existing CIGL methods primarily focus on adversarial generation [4,5], pseudo-labeling [6,7], model refinement [8, 9], and margin adjustment [10], while very few post-processing methods (e.g., data balancing during prediction) exists. We have carefully looked through existing literature and recent surveys [11,12], but faced challenges in finding works that apply *residual propagation* **(Q1)** or *test-time data balancing* **(Q2)** for class-imbalanced node classification. **We would greatly appreciate it if you could provide specific references, and we are more than willing to incorporate relevant experiments and analyses based on your suggestions.**
>
> ## (W2, W3) Test on balanced data
> **We note that CIGL inherently addresses the challenges posed by imbalanced datasets, thus this line of research (e.g., [1-10]) does not consider balanced data as an useful testbed. We provide additional discussions and experiments to illustrate the reasons behind.**
>
> **CIGL degeneratess into normal graph learning on perfectly balanced data**: When data is perfectly balanced (all classes have an equal number of samples), existing CIGL methods tend to degenerate into conventional graph learning processes. For instance, since there are no minority/majority class distinctions, oversampling methods like GraphSMOTE and GraphENS would generate no nodes, while reweighting methods would assign equal weights to each class, reducing them to standard graph learning. Becasue of this, CIGL research works do not conduct experiments on balanced data as this is considered beyond the problem scope.
>
> **Experiments on balanced datasets:** Please see the Table below. It can be observed that on balanced datasets, similar to other CIGL techniques, our method exhibits minimal impact on model training and performance remains consistent. This is attributed to (1) the smaller uncertainty of the base model and (2) the absence of imbalance-aware risk calibration.
>
> | Metric  | BAcc         |              |              | F1           |              |              |
> | ------- | ------------ | ------------ | ------------ | ------------ | ------------ | ------------ |
> | Dataset | Cora         | CiteSeer     | PubMed       | Cora         | CiteSeer     | PubMed       |
> | Vanilla GCN     | 81.67±0.51 | 65.28±0.54 | 77.94±0.27 | 78.89±0.54 | 64.85±0.52 | 76.01±0.14 |
> | GCN+ToBE_0  | 81.88±0.35 | 65.24±0.39 | 77.97±0.15 | 79.56±0.38 | 64.88±0.32 | 76.78±0.36 |
> | GCN+ToBE_1  | 81.90±0.56 | 65.66±0.37 | 77.89±0.32 | 78.97±0.88 | 65.25±0.37 | 76.61±0.27 |

---

> ### Author Response · Authors · 2023-11-15
> **Response to Reviewer grPF (Part 2)**
>
> ## (W4) On the usage of "relative improvement rate"
>
> **The "relative improvement rate" reported in our main results (Table 1) pertains to combinations of base model + CIGL method, rather than the base models themselves that may inherently perform poorly.** This ensures an objective reflection of the performance enhancement attributable to the ToBE method.
>
> > For example, the relative gain for **GCN+ReNode+ToBE** is calculated relative to the performance of **GCN+ReNode**, rather than **GCN** itself (which may perform poorly by itself in CIGL task, as you suggested).
>
> **In Table 2, the "relative improvement rate" was reported for both the best CIGL method and ToBE.** This is can fairly and objectively shows the performance gain brought about by the best CIGL method and ToBE. To better address your concern, we will include a more intuitive metric "performance gain" (Perf(A+ToBE) - Perf(A)), facilitating a clearer understanding of the performance advantage conferred by ToBE. This modification should enhances the transparency and interpretability of our results.
>
> ## (Q3.1, W5) How ToBE works with other CIGL methods
> Thank you for bringing this to our attention. We will include a detailed discussion in the paper.
>
> **In each training epoch, ToBE augments the original graph $\mathbf{G}$ based on the current model $\mathbf{F}$ and returns the augmented graph $\mathbf{G}^*$. Other CIGL methods then operate on $\mathbf{G}^*$ for subsequent steps**, specifically:
> - Loss function engineering methods (Reweight and ReNode) perform the loss computing and backpropagation based on $\mathbf{G}^*$.
> - Data augmentation methods (Resampling, SMOTE, GSMOTE, GENS) will conduct further class-balancing operations on $\mathbf{G}^*$, i.e., they will generate new minority nodes based on $\mathbf{G}^*$.
>
> ## (Q3.2) Can it be independently developed as a foundational model rather than a post-processing module?
> **ToBE is a model-agnostic in-processing module that can be boost any foundation GNN model (e.g., GCN, GAT, GraphSAGE, APPNP, GPRGNN, etc). It is not a post-processing module nor a foundation GNN model itself.**
>
> As described in the paper, ToBE is an in-processing data augmentation technique. It dynamically augments the data during training, with no dependence on the base model used. Our extensive experiments confirmed the versatility of ToBE on various GNN backbones.
>
> ## (Q4.1, W6) ToBE's scope and potential in other tasks
> We appreciate your valuable feedback regarding potential avenues for further research.
>
> **We note that the scope of this study aligns with the existing CIGL research** (e.g., [1] in WSDM'21, [2] in ICLR'22, [10] in ICML'22), **and it has widespread application in real-world scenarios**, e.g., outlier detection, fraudulent user detection, and social spammer detection [12].
>
> **We believe that our work may benefit tasks in other dedicated lines of research, but such extension is non-trivial.** For example, imbalanced edge prediction is an important task especially in domains like knowledge graph completion. However, extending existing CIGL concepts to address this task is non-trivial as it involves multi-facet imbalance: e.g., node-level (quantity diffenrent between different types of entities), edge-level (relation existence/non-existence), and degree-level (certain entities having more relationships). We recognize this as an important future direction and will discuss more possible future directions in the paper.

---

> ### Author Response · Authors · 2023-11-15
> **Response to Reviewer grPF (Part 3)**
>
> ## (Q4.2) How does ToBE perform in terms of inductive capabilities?
>
> **TL;DR: ToBE retifies minority misclassifications, and thus help GNN to learn better representations for minority classes even on unseen graphs. We conducted further experiments and confirm that ToBE can improve GNN's inductive learning capabilities.**
>
> This is an exciting problem. Following your suggestion, we test ToBE's incductive capabilities on the [protein-protein interaction networks (PPI)](https://pytorch-geometric.readthedocs.io/en/latest/generated/torch_geometric.datasets.PPI.html#torch_geometric.datasets.PPI). The task is to predict protein functions on **unseen proteins (graphs)**. The PPI dataset contains multiple distinct protein function targets, forming binary classification tasks with varying class imbalance ratios. This setup serves as a robust testbed for ToBE's capabilities in addressing class imbalance on unseen graphs. We show the results in the table below (PPI-X denotes using the X-th protein function as the target):
>
> **Balance Accuracy on PPI**:
> | Task   | PPI-1 (IR=2.054)   | PPI-2 (IR=3.051)   | PPI-3 (IR=3.835)   | PPI-4 (IR=3.010)   | PPI-5 (IR=6.963)   |
> | ------ | ------------------ | ------------------ | ------------------ | ------------------ | ------------------ |
> | GCN    | 73.52±1.26         | 72.06±3.81         | 69.79±2.82         | 74.61±3.16         | 71.70±3.85         |
> | GCN+ToBE_0 | **74.89**±1.64 (+1.36) | 75.47±2.34 (+3.41) | 70.17±3.20 (+0.38) | **77.64**±2.76 (+3.03) | 74.98±3.98 (+3.28) |
> | GCN+ToBE_1 | 73.97±1.44 (+0.44) | **76.56**±2.82 (+4.49) | **73.43**±3.14 (+3.63) | 76.34±3.13 (+1.73) | **75.12**±3.74 (+3.42) |
>
> **Macro F1 score on PPI**:
> | Task   | PPI-1 (IR=2.054)   | PPI-2 (IR=3.051)   | PPI-3 (IR=3.835)   | PPI-4 (IR=3.010)   | PPI-5 (IR=6.963)   |
> | ------ | ------------------ | ------------------ | ------------------ | ------------------ | ------------------ |
> | GCN    | 73.85±0.34         | 74.30±3.22         | 71.84±1.64         | 75.33±1.77         | 73.18±1.92         |
> | GCN+ToBE_0 | **75.23**±0.80 (+1.39) | 77.07±1.07 (+2.77) | 72.73±2.24 (+0.89) | **76.85**±1.34 (+1.51) | **76.33**±1.49 (+3.15) |
> | GCN+ToBE_1 | 74.86±0.91 (+1.01) | **77.58**±1.54 (+3.28) | **74.90**±1.62 (+3.06) | 76.48±1.40 (+1.15) | 76.28±1.76 (+3.11) |
>
> For each task, we use the first 5 graphs for training, 5 for validation, and test the model on the remaining 10 graphs. Results are averaged over 5 runs with different random seeds. The results show that **ToBE can help GNN to better generalize to unseen graphs**, especially for tasks with high class imbalance ratios (e.g., PPI-5). We attribute this to TOBE's ability to retify minority misclassifications, and thus help GNN to learn better representations for minority classes. A more detailed discussion will be included in the paper.
>
> **Thank you once again for your feedback. We hope we have addressed our concerns, and we welcome any follow-up question or discussion.**
>
> ## References
>
> [1] Tianxiang Zhao, Xiang Zhang, and Suhang Wang. 2021. GraphSMOTE: Imbalanced Node Classification on Graphs with Graph Neural Networks. In WSDM.
> [2] Joonhyung Park, Jaeyun Song, and Eunho Yang. 2022. GraphENS: Neighbor-Aware Ego Network Synthesis for Class-Imbalanced Node Classification. In ICLR.
> [3] Deli Chen, Yankai Lin, Guangxiang Zhao, Xuancheng Ren, Peng Li, Jie Zhou, and Xu Sun. 2021. Topology-Imbalance Learning for Semi-Supervised Node Classification. In NeurIPS.
> [4] Liang Qu, Huaisheng Zhu, Ruiqi Zheng, Yuhui Shi, and Hongzhi Yin. 2021. ImGAGN: Imbalanced Network Embedding via Generative Adversarial Graph Networks. In KDD.
> [5] Yijun Duan, Xin Liu, Adam Jatowt, Hai-tao Yu, Steven Lynden, Kyoung-Sook Kim, and Akiyoshi Matono. 2022. Anonymity Can Help Minority: A Novel Synthetic Data Over-sampling Strategy on Multi-label Graphs. In ECML/PKDD.
> [6] Dawei Zhou, Jingrui He, Hongxia Yang, and Wei Fan. 2018. SPARC: Self-Paced Network Representation for Few-Shot Rare Category Characterization. In KDD.
> [7] Mengting Zhou and Zhiguo Gong. 2023. GraphSR: A Data Augmentation Algorithm for Imbalanced Node Classification. In AAAI.
> [8] Yiyue Qian, Chunhui Zhang, Yiming Zhang, Qianlong Wen, Yanfang Ye, and Chuxu Zhang. 2022. Co-Modality Graph Contrastive Learning for Imbalanced Node Classification. In NeurIPS.
> [9] Min Shi, Yufei Tang, Xingquan Zhu, David Wilson, and Jianxun Liu. 2020. Multi-Class Imbalanced Graph Convolutional Network Learning. In IJCAI.
> [10] Song, Jaeyun, Joonhyung Park, and Eunho Yang. 2022. TAM: topology-aware margin loss for class-imbalanced node classification. in ICLR.
> [11] Ma, Yihong, et al. "Class-Imbalanced Learning on Graphs: A Survey." arXiv preprint arXiv:2304.04300 (2023).
> [12] Liu, Zemin, et al. "A survey of imbalanced learning on graphs: Problems, techniques, and future directions." arXiv preprint arXiv:2308.13821 (2023).

---

> > ### Comment · Reviewer_grPF · 2023-11-22
> >
> > I am very grateful that the author conducted further comparative experiments. I think some of my doubts have been resolved. However, as mentioned earlier, I still have concerns about the innovation and substantial contribution of this paper.
> >
> > Nevertheless, I am willing to raise the current score to 5.

---

> ### Author Response · Authors · 2023-11-21
> **Looking for your response**
>
> **Dear Reviewer grPF,**
>
> Thank you again for your time and effort in reviewing this submission. This is just a kind reminder that the discussion period is coming to an end. For your convenience, we provide a summary here to make it easier for you to locate our responses to each of your concerns:
> - **Part 1:**
>     - Clarification on ToBE's working manner, and discussion on potential counterparts
>     - Results and discussion for balanced data
> - **Part 2:**
>     - Clarification on the calulation of relative gain
>     - Explanation of how ToBE works with other CIGL methods
>     - Clarification on ToBE's ability to work with any fundation GNN model
>     - Discussions regarding ToBE's scope and potential in other tasks
> - **Part 3:**
>     - Further experiments and analysis on ToBE's inductive capability
>
> **Your opinion is important to us.** We would like to know whether our responses address your concerns properly, and we are more than happy to address any remaining points.
>
> Sincerely, Authors

---

> ### Author Response · Authors · 2023-11-23
> **Thank you for your positive response!**
>
> **Dear Reviewer grPF,**
>
> Thank you very much for your response. To thoroughly address your concerns, we have diligently revised the manuscript and uploaded the latest version. Please check the **[the updated PDF](https://openreview.net/pdf?id=bCNYFOaWsy)** with significant changes highlighted in **blue** for your convenience. **We believe this version demonstrates substantial improvements with stronger theoretical analyses, more comprehensive experiments, and related discussions.**
>
> Key revisions include:
> - For W4, we have revised **Tables 1 and 2**, replacing the confusing relative gain with the straightforward absolute gain.
> - For W5/Q3, we have included a detailed description of how ToBE collaborates with other CIGL methods in **Appendix C.2** (Baseline Settings).
> - For W6/Q4, we added results and analysis on the inductive capability of ToBE (**Appendix D.3**) and discussions on potential extension to more tasks in the broader area (**Appendix D.8**).
>
> Furthermore, several other noteworthy improvements have been made, such as (1) a reformed of the theoretical analysis for AMP/DMP, extending it from k=2 to the general case (**Appendix A**); (2) a comparison with the latest CIGL baselines (**Appendix D.6**); (3) further discussion on the advantages and limitations of the two variants of ToBE and practical considerations for their selection (**Appendix D.7**).
>
> **We hope that your concerns have been thoroughly addressed in this version of the paper. If so, we sincerely hope for your support in accepting our paper, and we are truly grateful for your consideration.** We will continue to carefully improve the paper’s organization, making it more accessible for broader groups of audiences to comprehend and appreciate our research.
>
> Thank you once again,
> The Authors

---

### Official Review · Reviewer_4g56 · 2023-11-09

**Soundness:** 2 fair
**Presentation:** 2 fair
**Contribution:** 2 fair
**Rating:** 5
**Confidence:** 4

**Summary:**

This study delves into the challenge of imbalanced node classification on graphs. The authors explore two phenomena in the underlying graph topology that intensify predictive bias due to class imbalance, both theoretically and empirically. They introduce a model called ToBE, designed to alleviate class-imbalance bias without the need for class rebalancing. Through experiments conducted on various datasets, the effectiveness of the proposed model is demonstrated.

**Strengths:**

1. Class imbalance is an important issue in the field of graph imbalance learning, which requires deep investigation.

**Weaknesses:**

1. The AMP and DMP phenomena are studied in many previous works. The AMP is basically the heterophily issue studied in previous heterophily GNN and graph anomaly detection literature. The DMP is basically the information insufﬁcient issue studied in previous topology imbalance literature.

2. I find the theoretical analysis has nothing to do with the model design. In the theoretical analysis, this work only analyzes the relation between the imbalance ratio and the severity of AMP and DMP. However, in the model design, this work directly uses model prediction uncertainty to estimate nodes’ risk of being misclassified and simply claims that this risk of being misclassified is due to AMP/DMP, which is not verified in the theoretical part. From my view, there is not any relationship between these two parts. Even without the theoretical part, the model design part looks self-contained.

3. In the model design, this work claims that, for high-risk nodes, the most possible prediction is unreliable, and instead uses the second possible prediction as the estimated label. Why choose the second possible prediction? Why not the third possible? As this part is critical to model performance, I would expect authors to further clarify this.

4. In Figure 2(b), it seems that there is still a large discrepancy between the performance of the minority class and the majority class, even if the AMP/DMP score is the same. That indicates that there are some other factors that influence the performance discrepancy. I would expect the authors to further clarify this.

5. The existing baselines lack comprehensiveness, and there is a lack of thorough comparison with recent approaches, such as [1, 2, 3, 4, 5].

[1] Imgcl: Revisiting graph contrastive learning on imbalanced node classification. AAAI 2023

[2] Balanced neighbor exploration for semi-supervised node classification on imbalanced graph data. Information Sciences 2023

[3] Graphmixup: Improving class-imbalanced node classification by reinforcement mixup and self-supervised context prediction. ECML-PKDD 2022

[4] Imbalanced node classification beyond homophilic assumption. IJCAI 2023

[5] Tam: Topology-aware margin loss for class-imbalanced node classification. ICML 2022

**Questions:**

Please see the Weaknesses.

**Details Of Ethics Concerns:**

NA.

---

> ### Author Response · Authors · 2023-11-15
> **Response to Reviewer 4g56 (Part 1)**
>
> **We appreciate for your thoughtful review as well as the time and effort you have taken to evaluate our work. We have carefully considered your comments and criticisms, please see our itemized responses to each of your concerns:**
>
> ## (W1) Regarding the concepts of AMP and DMP
>
> **TL;DR: Our novelty is identifying the role that AMP and DMP play in forming biases under class imbalance, which is previously unknown in the class-imbalanced graph learning (CIGL) literature (rather than discovering AMP and DMP themselves).**
>
> As you suggested, and as we discussed in the related works section, AMP and DMP are widely discussed not only in topology imbalance research [1], but also across works concerning graph heterophily [2, 3] and long-distance propagation [4]. The core contribution of our work is that: in the context of class imbalance, we identify the inherent vulnerability of minority nodes to AMP and DMP, and propose a practical framework ToBE to address this issue.
>
>  It's also worth noting that our definitions of AMP and DMP have subtle differences from existing concepts. For example, heterophily focuses on global properties of the graph, but AMP is defined at the node level. Similarly, the influence conflict discussed in [1] primarily targets labeled nodes, whereas AMP/DMP mainly focus on unlabeled nodes. Therefore, to avoid confusion, we named these two factors AMP and DMP. **We will carefully consider alternative names that better emphasize their connection to existing concepts, and we welcome any suggestions you may have on this matter.**
>
> ## (W2) From theoretical results to algorithm design
>
> **TL;DR: Armed with the theoretical results, we investigated AMP/DMP on real-world graphs and found that the impact of AMP/DMP is concentrated on a small fraction of “critical” nodes. This provides us a way to surrogate the infeasible \rho (imbalance ratio) manipulation, and inspired the development of ToBE. Our experimental results confirmed ToBE's ability in alleviating both AMP and DMP (e.g., Figure 6).**
>
> Thank you for your careful and thoughtful review of our paper. As discussed in **Section 2 - "A closer look at AMP & DMP in practice"**, while the imbalance ratio ($\rho$) determines the severity of bias introduced by AMP/DMP, directly altering $\rho$ is not feasible (oversampling methods synthesize new samples based on existing observations, which is fundamentally different from sampling from the underlying distribution). Consequently, we further investigated the distribution of AMP/DMP on real-world graphs and found that the impact of AMP/DMP is concentrated on a small fraction of “critical” nodes. This insight inspired the development of ToBE, and experimental results demonstrate its efficient alleviation of AMP and DMP (e.g., Figure 6).
>
> > In this work, we aimed for a lightweight and broadly applicable algorithm. We acknowledge the need for a more in-depth exploration of ToBE's properties and also more informed algorithm design in future work. Your feedback is greatly appreciated and will consider it in refining our approach and conducting more detailed analyses in subsequent research.
>
> ## (W3) Second- or third-possible prediction?
>
> **TL;DR: We wish to clarify that ToBE computes the likelihood for each class (as indicated in equations (5) and (6)) and samples from the distribution, thus it is not restricted to using only the second possible prediction.**
>
> We apologize for the misunderstanding, which may caused by the illustrative example presented in Figure 5. In practice, if the third possible prediction (say C3) has a probability close to that of the second possible prediction (say C2), they will have similar chances of being selected as the target class. Additionally, we emphasize that the sampling process for each virtual edge is independent. This means that the virtual edges to C2 and C3 may coexist (although it is a rare case due to the discount factor γ). **Such a design helps prevent the accumulation of errors. We will further emphasize this point in the paper (especially in figure 5) to clarify the virtual edge sampling process.**

---

> ### Author Response · Authors · 2023-11-15
> **Response to Reviewer 4g56 (Part 2)**
>
> ## (W4) The other factor behind performance discrepancy
>
> Great question! **TL;DR: The discrepancy between the two curves in Figure 2(b) stems from the poor representation of minority classes in the feature space, which is caused by the "class imbalance" itself.**
>
> Consider a graph with features $\mathbf{X}$ and adjacency matrix $\mathbf{A}$. While AMP/DMP are factors related to the topological structure ($\mathbf{A}$), class imbalance also significantly influences the representation quality of different classes in the feature space ($\mathbf{X}$).  Therefore, a model learn from $\mathbf{X}$ (i.e., traditional i.i.d. imbalance learning) will inherently exhibit poorer generalizability for minority classes [5], and thus minority classes are more sensitive to noises [6, 7]. Additionally, recent research suggests that GNN can be viewed as a Graph Signal Denoising [8, 9] that denoise $\mathbf{X}$ with a low-pass filter based on $\mathbf{A}$. From this perspective, we see that AMP introduces additional noise from dissimilar nodes, and DMP leads to less efficient label propagation/denoising for a specific node. Both factors hinders the graph denoising process, and thus minority class nodes (which are more sensitive to noise) are more prone to the influence of AMP/DMP (i.e., experiencing more performance degradation at the same AMP/DMP level).
>
> > In essence, Figure 2(a) illustrates the proportion of high AMP/DMP nodes for majority/minority classes, while Figure 2(b) demonstrates the sensitivity of majority/minority classes to AMP/DMP. The discussions were initially included, but were omitted due to space constraints. We will take this discussion back in the paper to better illustrate the results.
>
> ## (W5) Comparison with latest baselines
>
> Thank you for the suggestions. **We have included more results comparing ToBE with the suggested CIGL baselines.**
>
> Given the need in our experiments to test ToBE with a comprehensive combination of existing CIGL techniques and GNN backbones, we chose to include model-agnostic and efficient CIGL methods for a thorough evaluation.
>
> Nevertheless, we thoroughly investigated the literature you provided and implemented [BNE](https://github.com/ZonghaiZhu/BNE) [11], [GraphMixup](https://github.com/LirongWu/GraphMixup) [12], and [TAM](https://github.com/Jaeyun-Song/TAM) [14] (the other two papers [10][13] did not provide source code). To ensure a fair comparison, we implemented these methods in a unified experimental framework, employing the same data splitting strategy as in our paper and reporting the averages over five independent runs (i.e., ensuring comparability with the results reported in our paper). The results for Imbalance Ratio=10/20 are as follows:
>
> **Imbalance Ratio = 10**
>
> |Metric|BAcc|||F1|||Disp|||
> |-|-|-|-|-|-|-|-|-|-|
> |Dataset|Cora|CiteSeer|PubMed|Cora|CiteSeer|PubMed|Cora|CiteSeer|PubMed|
> |GCN|0.619|0.376|0.642|0.607|0.281|0.551|0.275|0.299|0.347|
> |BNE|0.651|0.515|0.654|0.674|0.497|0.613|0.204|0.155|0.245|
> |TAM|0.620|0.381|0.671|0.610|0.290|0.607|0.296|0.324|0.308|
> |GraphMixup|0.641|0.508|OOM|0.627|0.473|OOM|0.245|0.252|OOM|
> |ToBE_0|0.655|0.527|**0.686**|0.633|0.516|**0.672**|0.213|**0.139**|**0.092**|
> |ToBE_1|**0.698**|**0.554**|0.676|**0.687**|**0.550**|0.644|**0.185**|**0.139**|0.218|
>
> **Imbalance Ratio = 20**
>
> |Metric|BAcc|||F1|||Disp|||
> |-|-|-|-|-|-|-|-|-|-|
> |Dataset|Cora|CiteSeer|PubMed|Cora|CiteSeer|PubMed|Cora|CiteSeer|PubMed|
> |GCN|0.619|0.376|0.642|0.607|0.281|0.551|0.275|0.299|0.347|
> |BNE|0.651|0.515|0.654|0.674|0.497|0.613|0.204|0.155|0.245|
> |TAM|0.620|0.381|0.671|0.610|0.290|0.607|0.296|0.324|0.308|
> |GraphMixup|0.641|0.508|OOM|0.627|0.473|OOM|0.245|0.252|OOM|
> |ToBE_0|0.655|0.527|**0.686**|0.633|0.516|**0.672**|0.213|**0.139**|**0.092**|
> |ToBE_1|**0.698**|**0.554**|0.676|**0.687**|**0.550**|0.644|**0.185**|**0.139**|0.218|
>
> It can be observed that ToBE still exhibits advantage compared to these methods.
> We also note that GraphMixup uses the dense adjacency matrix for computation, which leads to out-of-memory (OOM) issues on the PubMed dataset (with Tesla V100 32GB GPU).
> We will include relevant discussions in the paper.
>
> > We would like to emphasize that due to the properties of these methods, we did not combine ToBE with them but conducted a direct comparison. BNE preprocesses graph data to obtain independent features and directly uses MLP for prediction, making it incompatible with ToBE and existing GNNs. GraphMixup and TAM rely on invariant graph topology, and the code implementation for GraphMixup does not support the addition of virtual nodes to the original graph.

---

> ### Author Response · Authors · 2023-11-15
> **Response to Reviewer 4g56 (Part 3)**
>
> **Thank you once again for your feedback. We hope we have addressed our concerns, and we welcome any follow-up question or discussion.**
>
> ## References
> [1] Chen, Deli, et al. "Topology-imbalance learning for semi-supervised node classification." Advances in Neural Information Processing Systems 34 (2021): 29885-29897.
> [2] Xin Zheng, Yixin Liu, Shirui Pan, Miao Zhang, Di Jin, and Philip S Yu. Graph neural networks for graphs with heterophily: A survey. arXiv preprint arXiv:2202.07082, 2022.
> [3] Zhe Xu, Yuzhong Chen, Qinghai Zhou, Yuhang Wu, Menghai Pan, Hao Yang, and Hanghang Tong. Node classification beyond homophily: Towards a general solution. In Proceedings of the 29th ACM SIGKDD Conference on Knowledge Discovery and Data Mining, pp. 2862–2873, 2023.
> [4] Gasteiger, Johannes, Aleksandar Bojchevski, and Stephan Günnemann. "Predict then Propagate: Graph Neural Networks meet Personalized PageRank." International Conference on Learning Representations. 2018.
> [5] Johnson, Justin M., and Taghi M. Khoshgoftaar. "Survey on deep learning with class imbalance." Journal of Big Data 6.1 (2019): 1-54.
> [6] Seiffert, Chris, et al. "An empirical study of the classification performance of learners on imbalanced and noisy software quality data." Information Sciences 259 (2014): 571-595.
> [7] Napierala, Krystyna, and Jerzy Stefanowski. "Types of minority class examples and their influence on learning classifiers from imbalanced data." Journal of Intelligent Information Systems 46 (2016): 563-597.
> [8] Ma, Yao, et al. "A unified view on graph neural networks as graph signal denoising." Proceedings of the 30th ACM International Conference on Information & Knowledge Management. 2021.
> [9] Nt, Hoang, and Takanori Maehara. "Revisiting graph neural networks: All we have is low-pass filters." arXiv preprint arXiv:1905.09550 (2019).
> [10] Imgcl: Revisiting graph contrastive learning on imbalanced node classification. AAAI 2023
> [11] Balanced neighbor exploration for semi-supervised node classification on imbalanced graph data. Information Sciences 2023
> [12] Graphmixup: Improving class-imbalanced node classification by reinforcement mixup and self-supervised context prediction. ECML-PKDD 2022
> [13] Imbalanced node classification beyond homophilic assumption. IJCAI 2023
> [14] Tam: Topology-aware margin loss for class-imbalanced node classification. ICML 2022

---

> ### Author Response · Authors · 2023-11-21
> **Looking for your response**
>
> **Dear Reviewer 4g56,**
>
> Thank you again for your time and effort in reviewing this submission. This is just a kind reminder that the discussion period is coming to an end. For your convenience, we provide a summary here to make it easier for you to locate our responses to each of your concerns:
> - **Part 1:**
>     - Our novelty in discovering the role of AMP//DMP under class imbalance, and how AMP/DMP related to existing concepts
>     - How do our theoretical results motivate the algorithm design
>     - Clarification on the second/third-possible prediction
> - **Part 2:**
>     - Reasons behind the performance discrepancy in Figure 2(b)
>     - Further comparison between ToBE and the latest baselines
>
> **Your opinion is important to us.** We would like to know whether our responses address your concerns properly, and we are more than happy to address any remaining points.
>
> Sincerely, Authors

---

> ### Author Response · Authors · 2023-11-23
> **We are looking forward to your reply**
>
> **Dear Reviewer 4g56,**
>
> Thank you again for your invaluable comments, they have greatly helped us to further improve the paper. To thoroughly address your concerns, we have diligently revised the manuscript and uploaded the latest version. Please check the **[the updated PDF](https://openreview.net/pdf?id=bCNYFOaWsy)** with significant changes highlighted in **blue** for your convenience. **We believe this version demonstrates substantial improvements with stronger theoretical analyses, more comprehensive experiments, and related discussions.**
>
> Key revisions include:
> - For W1, we have carefully revised the **related work section** to emphasize on the relationship between AMP/DMP and existing concepts.
> - For W2 and W4, we rewrote the ending paragraph of **Section 2** (theoretical analysis) to better illustrate the factors behind the performance disparity in **Figure 2(b)** and the motivation of our algorithm design.
> - For W3, we have added a footnote for **Figure 5** to improve clarity and prevent further confusion.
> - For W5, we include setup details, results, and analysis regarding the comparison with the latest baselines in **Appendix D.6**.
>
> Furthermore, several other noteworthy improvements have been made, such as (1) a reformed theoretical analysis for AMP/DMP, extending it from k=2 to the general case (**Appendix A**); (2) validation of ToBE's effectiveness not only in transductive scenarios but also in aiding GNN generalization to unseen graphs (**Appendix D.4**); (3) further discussion on the advantages and limitations of the two variants of ToBE and practical considerations for their selection (**Appendix D.7**).
>
> **We hope that your concerns have been thoroughly addressed in this version of the paper. If so, we sincerely hope for your support in accepting our paper, and we are truly grateful for your consideration.** We will continue to carefully improve the paper’s organization, making it more accessible for broader groups of audiences to comprehend and appreciate our research.
>
> Looking forward to your reply and thank you once again,
> The Authors

---

### Meta-Review · Area_Chair_xfHU · 2023-12-06

**Metareview:**

The paper provides a technique named ToBE aimed to handle class imbalance in the context of node classification. This problem of class imbalance is well motivated. The method provided in the paper is motivated by theoretical analysis, and is tested on established datasets. Another notable advantage of the method is it being model agnostic.
This being said, the reviews point out concerns regarding the analysis not being thorough enough: The comparison to the baselines was mentioned as unclear (resulting e.g. in the comment by UMPW related to weakness 1), and the design choices may deserve a more thorough justification (see comments by 4g56). The authors provided a response that may alleviate these concerns to some effect, but the magnitude of the required change seems to me to be too large to justify accepting the paper in its current form, rather than have it undergo a new set of reviews. Concluding, the paper has potential but it requires a revision before it can be accepted.

**Justification For Why Not Higher Score:**

major issues regarding the depth of the analysis.

**Justification For Why Not Lower Score:**

n/a

---

### Decision · Program_Chairs · 2024-01-16

Reject